# Stable mid-infrared polarization imaging based on quasi-2D tellurium at room temperature

Lei Tong[1,5], Xinyu Huang[1,5], Peng Wang[2,5], Lei Ye[1✉], Meng Peng[1,2], Licong An[3], Qiaodong Sun[1], Yong Zhang[1], Guoming Yang[1], Zheng Li[1], Fang Zhong[2], Fang Wang[2], Yixiu Wang[3], Maithilee Motlag[3], Wenzhuo Wu[3✉], Gary J. Cheng[3✉] & Weida Hu[2,4✉]

Next-generation polarized mid-infrared imaging systems generally requires miniaturization, integration, flexibility, good workability at room temperature and in severe environments, etc. Emerging two-dimensional materials provide another route to meet these demands, due to the ease of integrating on complex structures, their native in-plane anisotropy crystal structure for high polarization photosensitivity, and strong quantum confinement for excellent photodetecting performances at room temperature. However, polarized infrared imaging under scattering based on 2D materials has yet to be realized. Here we report the systematic investigation of polarized infrared imaging for a designed target obscured by scattering media using an anisotropic tellurium photodetector. Broadband sensitive photoresponse is realized at room temperature, with excellent stability without degradation under ambient atmospheric conditions. Significantly, a large anisotropic ratio of tellurium ensures polarized imaging in a scattering environment, with the degree of linear polarization over 0.8, opening up possibilities for developing next-generation polarized mid-infrared imaging technology.

[1] School of Optical and Electronic Information and Wuhan National Laboratory for Optoelectronics, Huazhong University of Science and Technology, Wuhan 430074, China. [2] State Key Laboratory of Infrared Physics, Shanghai Institute of Technical Physics Chinese Academy of Sciences, Shanghai 200083, China. [3] School of Industrial Engineering and Birck Nanotechnology Centre, Purdue University, West Lafayette, IN 47907, USA. [4] Hangzhou Institute for Advanced Study, University of Chinese Academy of Sciences, Hangzhou 310024, China. [5]These authors contributed equally: Lei Tong, Xinyu Huang, Peng Wang. ✉email: leiye@hust.edu.cn; wenzhuowu@purdue.edu; gjcheng@purdue.edu; wdhu@mail.sitp.ac.cn

Polarized infrared imaging systems (PIRIS) are of great importance for the development of astronomy, biological diagnosis, national security, and industrial applications[1–4], which require precise recognition of a target object in the presence of clutter. At present, the trend of modernizing polarized infrared imaging applications increasingly requires convenience, miniaturization, good workability at room temperature and severe environments, and larger pixel density for future devices[5–7]. Recently, the emerging researches into two-dimensional (2D) materials have potential to provide another routine for next-generation photodetectors. The inherent characteristics of strong light-matter interaction and low dark current in 2D materials can provide high photoresponsivity and low noise at room temperature[8–14]. The lack of dangling bonds on the surface of 2D materials can reduce surface-induced performance degradation, facilitating easy integration with other materials or substrates[15]. More importantly, some 2D materials with intrinsic in-plane anisotropy structure exhibit the potential to realize polarization photodetection without polarization filters[16–18]. Hence, it is a feasible approach to develop state-of-the-art PIRIS devices by implementing in-plane anisotropic 2D materials. Although many 2D materials have been used to successfully fabricate polarization photodetectors, there are still some limitations of these 2D materials which hinder their practical applications, such as instability under ambient conditions, narrow photoresponse range, and insufficient polarized sensitivity[9,19–23]. Besides, there are reports of 2D material mid-infrared (MIR) polarization photodetectors[24,25], but the works about polarization imaging under scattering based on 2D materials have not yet been realized, addressing the feasibility of 2D materials-based photodetectors to further achieve contrast-enhanced polarized infrared imaging.

Here, we have systematically investigated polarized infrared imaging applications based on quasi-2D tellurium (Te) photodetector. The narrow bandgap of quasi-2D Te provides strong light-matter interactions in the spectral range from visible to MIR (~3.0 μm), and the high carrier mobility and small effective carrier mass provide the potential to achieve high photo-detecting performance[26–28]. Furthermore, the anisotropic crystal structure of quasi-2D Te gives rise to strong anisotropic photoresponse, explained by the energy band structure in theory[27,29–31]. Detailed Raman spectrum characterizations have been performed to verify the high quality of quasi-2D Te, carrier mobility measured by field-effect transistor (FET)-based devices can reach up to ~$9 \times 10^2 \, cm^2 \, V^{-1} \, s^{-1}$ at room temperature, which is at the forefront in the field of 2D materials. The wide spectral photoresponse of the Te photodetector is confirmed from 0.52 to 3.0 μm, and the most sensitive photoresponsivity ($R$) under 1.06 μm illumination is ~$1.36 \times 10^3 \, AW^{-1}$ with corresponding measured detectivity ($D_M^*$) of ~$1.15 \times 10^{10}$ Jones. The photoresponsivity also remains higher than ~$3.53 \times 10^2 \, AW^{-1}$ and the measured detectivity higher than ~$3.01 \times 10^9$ Jones, when illuminated by the incident light with a wavelength of 3.0 μm. Due to the unique asymmetric crystal structure, the polarized photoresponse is also demonstrated, with the photoresponsivity anisotropic ratio of ~8 under the incident light with a wavelength of 2.3 μm, which is sufficient for potential polarized imaging. Importantly, polarized infrared imaging tests under scattering environments have also been carried out for the Te-based photodetectors, without the integration of complex polarization filters. Under strongly scattering environment, achieving target photoelectric imaging is hard for 2H-phase MoTe$_2$-based photodetectors which have no polarized photoresponse, however, the quasi-2D Te-based photodetector reported here can realize imaging for the target with the degree of linear polarization (DoLP) over 0.8 at the

wavelength of 2.3 μm. Furthermore, we have also implemented polarized imaging for incident light over a wide wavelength range in more detail. Significantly, the feasibility of implementing Te for polarized imaging proved in this work is at the first stage, for the next step, the integration for authentic facilities in the future requires efforts upon achieving the design and fabrication of quasi-2D Te-based area-array, which worth more investigations.

## Results

**Anisotropic characterization of Te.** The quasi-2D Te crystals (Supplementary Note 1, Supplementary Fig. 1a) are synthesized through a solution method[27,32], the crystal quality is verified through Raman spectrum tests (Supplementary Fig. 1b–h). The thickness of Te flake measured by atomic force microscope (AFM) is ~27.5 nm (Supplementary Fig. 1i–j), and its thicker thickness is beneficial to reach stronger absorption, leading to higher photocurrent, but the thickness should not be too thick, otherwise the overlarge dark current will lead to poor photo-detectivity and other performance. Besides, the thicker Te flake possesses narrower bandgap[8], which is beneficial to extend the detectable bandwidth to longer infrared wavelength. Quasi-2D Te consists of a helical chain in the [0001] direction, with adjacent helical chains stacked together through a non-covalent weak bonding[26–28,32], forming an anisotropic crystal structure, as shown in Fig. 1a. To investigate its electrical properties in more detail, we have fabricated a Te-based FET device with five parallel electrodes marked by numbers from 1 to 5 as shown in the schematic of the device in Fig. 1b, resulting in different channel lengths between electrodes, and the channel length between two electrodes is measured through optical images as shown in Fig. 1c. Figure 1d shows the transfer curves for the devices with different channel lengths at room temperature, with the drain bias fixed at 1.0 V and the gate bias is varied from −10.0 V to 10.0 V. The on–off ratio is slightly decreased in shorter channel length, which is due to slightly weaker gate control over the drain current. High mobility of ~$9 \times 10^2 \, cm^2 \, V^{-1} \, s^{-1}$ is calculated from the transfer curves, which verifies the potential to achieve high photodetecting performance (more detailed analysis in Supplementary Notes 2 and 3, Supplementary Table 1, Supplementary Figs. 2 and 3, 25). The anisotropic crystal structure of Te causes a certain degree of intrinsic anisotropy for the electrical performances of Te-based FET (optical image is shown in Supplementary Fig. 1k), with the electrical conductivity anisotropic ratio (along $x$- and $y$-axis) of ~2.24 under gate bias of 0 V (inset of Fig. 1e). From the transfer curves (Fig. 1e), we have calculated the on–off ratio and field-effect mobility for both $x$-axis and $y$-axis directions respectively, as shown in Fig. 1f, similarly, the field-effect mobility anisotropic ratio is ~1.69 (Fig. 1g). The polarized Raman spectra of Te is measured as shown in Fig. 1h–k. Based on the results, the intensities of both $E_1$-$TO$ peak and $A_1$ peak show two-lobe shape, while the intensity of $E_2$ peak shows a four-lobe shape (Fig. 1i–k). To more intuitively present the high anisotropy in Raman spectra, we have defined Raman anisotropic ratio as the maximum intensity of Raman peak over the minimum, producing Raman anisotropic ratios of ~13, ~4, and ~12.5 for $E_1$-$TO$, $A_1$, and $E_2$ mode, respectively, in experiments. In theory, quasi-2D Te crystal belongs to $D_3^4$ space group, the anisotropic Raman intensity can be fitted by Raman tensor[33,34]:

$$R(E_1 - TO) = \begin{bmatrix} a & 0 & 0 \\ 0 & b & c \\ 0 & c & 0 \end{bmatrix}, \quad (1)$$

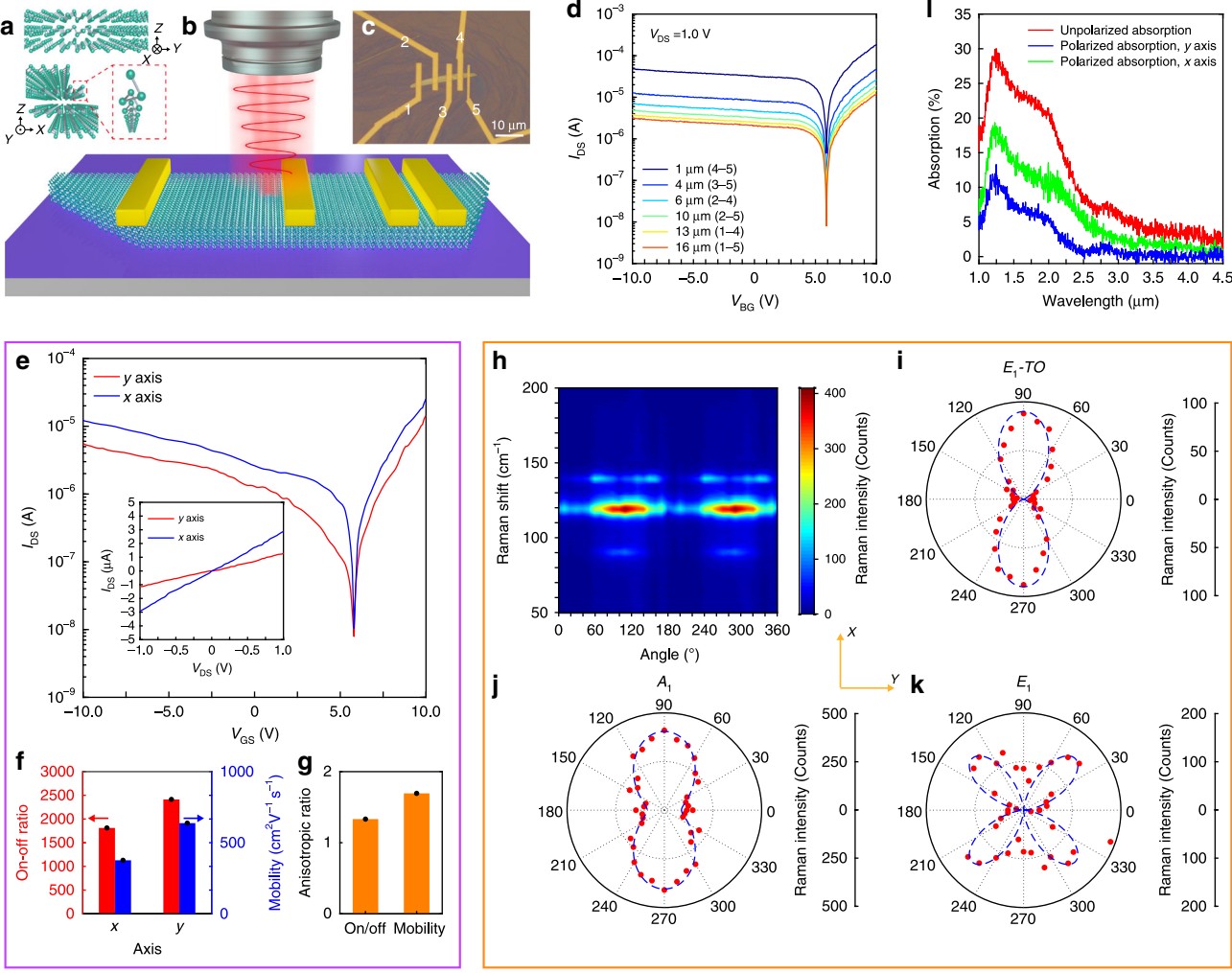

**Fig. 1 Anisotropic characterization of Te nanoflake. a** Schematic of the crystal structure of Te with a helical chain formed by three adjacent tellurium atoms. **b** Schematic of the device structure. **c** Optical image of the device, the distance between the five electrodes are 4, 3, 2, and 1 μm, respectively. **d** Transfer curve at different channel lengths, the selected electrodes are marked in the legends, the drain bias is 1.0 V. **e** Anisotropic transfer curve along x- and y-axis, the gate bias is from −10.0 to 10.0 V, the drain bias is fixed at 1.0 V. The inset shows the drain current under the drain bias from −1.0 to 1.0 V and the gate bias of 0 V, the current along x-axis is ~2.24 times of the current along y-axis. **f** Calculated on–off ratio and field-effect mobility based on the transfer curve in (**e**) along x- and y-axis. **g** Anisotropic ratio of on/off ratio and mobility based on the data in (**e**). **h** Polarized Raman intensity mapping of tellurium nanoflake as a function of wave number and incident angle, **i–k** polar plot of $E_1$-TO, $A_1$, and $E_2$ mode intensity. $E_1$-TO, $A_1$ mode are two-lobe shape, and $E_2$ mode is four-lobe shape. The red spots are experimental results, and the blue dashed lines are fitted results. The anisotropic ratios are ~13, ~4, and ~12.5 for $E_1$-TO, $A_1$, and $E_2$ mode, respectively. **l** Absorption spectra of tellurium nanoflake, the red line is the unpolarized absorption spectra of tellurium nanoflake, with a peak located at ~1.25 μm, and the green and blue lines are polarized absorption spectra of Te nanoflake, with the incident light polarization direction along x- and y-axis, respectively. From 1.25 to 3.0 μm, the absorption along x-axis is higher than that along y-axis, and the anisotropic ratio at 1.25 and 2.3 μm are ~1.8 and ~8.0, respectively.

$$R(A_1) = \begin{bmatrix} d & 0 & 0 \\ 0 & e & 0 \\ 0 & 0 & f \end{bmatrix}, \quad (2)$$

$$R(E_2) = \begin{bmatrix} 0 & g & h \\ g & 0 & 0 \\ h & 0 & 0 \end{bmatrix}, \quad (3)$$

where $a$, $b$, $c$, $d$, $e$, $f$, $g$, and $h$ are Raman tensor elements. The polarized Raman intensity for the three modes are presented as follows:

$$I_{E_1-\text{TO},\parallel} = |a\sin^2\theta|^2, \quad (4)$$

$$I_{A_1,\parallel} = |d\sin^2\theta + f\cos^2\theta|^2, \quad (5)$$

$$I_{E_2,\parallel} = |2h\sin\theta\cos\theta|^2. \quad (6)$$

where $\theta$ is the angle between the longer side of the crystal and the incident light polarization direction (simulation details in Supplementary Note 4, Supplementary Fig. 4). Based on the Raman tensor analysis, the polarized Raman spectra can be fitted precisely, and the x-axis of crystal orientation can be determined along the direction which possesses the maximum intensity of $E_1$-TO peak[27,33]. The measured absorption of Te is also anisotropic, as shown in Fig. 1. The absorption characters from NIR to MIR range is observed under nonpolarized illumination (red curve), with a maximum absorption of ~28% located near 1.25 μm and

absorption edge extended to more than 3.0 μm. We have also measured the polarized absorption spectra with polarized illumination along the y- and x-axis direction, respectively. From the results, the absorption along the x-axis (green curve) is higher than that along the y-axis (blue curve) for the measured wavelength range, which is relevant to the anisotropic band structure of the Te Brillouin zone (Supplementary Fig. 5). The anisotropic ratio of absorption increases with the increase of absorption wavelength (Supplementary Fig. 6)[27,29]. Notably, in the range of absorption wavelength from 2.5 to 3.0 μm, the y-axis direction absorption is very low. Therefore, the absorption anisotropic ratio can reach up to ~10 for the absorption range from 2.3 to 3.0 μm, promising excellent potential in polarized MIR photodetectors.

**Broadband stable optoelectronic performance of Te**. The photoresponse performances are measured between the electrode 2 and 4 in Fig. 1c, with the device channel length of ~6 μm and channel width of ~2.5 μm, located at the center of the Te flake. The gate bias is fixed at 0 V, the drain bias is varied from −1.0 to 1.0 V, and the incident lights are unpolarized with the output power from 0 to 6.0 mW. As shown in Fig. 2, the photoresponse wavelength can cover the range from 0.52 to 3.0 μm. Figures 2a, b display the photocurrent of Te-based photodetector illuminated under 2.3 and 3.0 μm at 300 K temperature, respectively, showing its feasibility for photodetecting at NIR and MIR range. (All photocurrent curves at 300 K temperature are included in Supplementary Fig. 7a-j.) The corresponding photocurrents are increasing with the increase of incident laser power, indicating the high responses for MIR photodetection. To characterize the device performances under different illumination powers and wavelengths, we have summarized the major parameters of photoresponsivity ($R$) and measured detectivity ($D_M^*$) of the photodetector at room temperature by using the pseudo-color mapping, as shown in Fig. 2e, f (scatter diagrams are included in Supplementary Fig. 7k-n). The drain bias is fixed at 1.0 V, and the gate bias is fixed at 0 V, the frequency of the noise current is chosen to be 1 kHz, the corresponding noise current density is $4.55 \times 10^{-11}\,\mathrm{A}\sqrt{\mathrm{Hz}^{-1}}$ at room temperature, and the calculation details are described in the Supplementary Notes 5. The lower illumination power always leads to higher $R$ and $D_M^*$, and the evolution of $R$ and $D_M^*$ under fixed laser power is in accordance with absorption results. Due to effective absorption of Te at 3.0 μm, high $R$ of ~$3.54 \times 10^2\,\mathrm{AW}^{-1}$ and $D_M^*$ of ~$3.01 \times 10^9$ Jones are realized under the illumination at the wavelength of 3.0 μm, which is competitive among various 2D materials-based photodetectors (Supplementary Table 2, Supplementary Fig. 26)[24–26,35–43]. The thick Te flake (Supplementary Fig. 1i-j) also contributes to such high performance due to its stronger absorption character, as the transmitted light to the substrate is limited. The high performance is also related to a large photoconductive gain, which is assisted by the fast carrier transition time due to the high mobility of Te and the short channel length of ~6.0 μm. In addition, the defects located inside the Te crystal can contribute to the photoconductive gain through shallow states of photo-induced carriers at the band tail[44,45], which is a fast process to provide a large gain. The existence of photogain is revealed from the EQE values in Supplementary Fig. 8a-c. Besides, we have also measured the transfer curves under 1.06 μm illumination and 1.0 V drain bias at room temperature, as shown in Fig. 2g, the dark state is shown in Fig. 2h, and the net photocurrent is summarized in Fig. 2i, the maximum net photocurrent is realized at the neutral point with ~5.9 V gate bias, also suggesting that the photoresponses of the device are not dominated by the trap-induced photogating effect. To further show the origin of photogain, we have measured the photocurrent

distribution in the Te channel with another device (Fig. 2j), the photocurrent is generated at the channel/electrode interface instead of the whole channel, excluding the photogating as a dominant factor for the high photogain. The response time is measured at 1.0 V drain bias, 0 V gate bias, and 6.0 mW laser output power for several selected wavelengths, as summarized in Supplementary Fig. 9, the rise time is at the range of 48.7–62.7 μs, and the fall time is at the range of 62.7–78.0 μs. The carrier lifetime is extracted from transient photoluminescence decay curves of Te, as shown in Supplementary Fig. 10. Based on carrier lifetime and transit time, a photogain ($G$) at the level of ~$2.3 \times 10^3$ can be calculated, indicating the high performance in the Te-based device. Besides, to make it more convenient to compare the device performances with previous works, the calculated detectivity ($D_C^*$) is also summarized in Supplementary Fig. 11. It is still worth noting that the calculated detectivity ($D_C^*$) can always overestimate the actual detectivity of 2D materials-based devices, as the dark current is lower than the actual noise current. Moreover, we have also investigated the device performances under low-temperature conditions, to more completely analyze its photodetecting performances. The drain bias is also varied from −1.0 to 1.0 V, and the gate bias is fixed at 0 V. At lower temperature (77 K), the photodetecting performances can be further enhanced due to lower dark current, as shown in Fig. 2c, d (all photocurrent curves are summarized in Supplementary Fig. 12a-j). The dependences of $R$ and $D_M^*$ on the wavelength and laser power of the incident light, are summarized in Fig. 2k, l, respectively (scatter diagrams are included in Supplementary Fig. 12k-n, external quantum efficiency ($EQE$) is shown in Supplementary Fig. 8d-f). Here the results show similar trends to those measured at 300 K (Fig. 2e, f). At 77 K, a higher $R$ can reach up to ~$8.36 \times 10^2\,\mathrm{AW}^{-1}$ and $D_M^*$ is at a higher level of ~$9.04 \times 10^9$ Jones, under the illumination wavelength of 3.0 μm. Moreover, device stability is very important for 2D materials-based MIR photodetectors. Although some 2D materials-based photodetectors are showing high MIR photoresponse, they cannot realize long-lasting stable photodetecting under ambient environment, such as easily oxidized black phosphorus and black phosphorus-arsenic. Here, the Te photodetector reveals high stability under ambient conditions, and the device has been measured after 3 months, exhibiting unnoticeable degradation of performance (Fig. 2m and Supplementary Fig. 13). The corresponding $R$ and $D_M^*$ in Fig. 2n, o are calculated from the shaded rectangle region in Fig. 2m, which also indicate the stable high performance of the device.

**Polarized photoresponses of Te**. The polarized photoresponses of the device have also been investigated under linear polarized illuminations. The drain bias is varied from −1.0 to 1.0 V, the gate bias is fixed at 0 V, the laser power is fixed at 6.0 mW, the laser polarization direction is fixed and initially parallel to the longer side of the device (y-axis which is defined as 0°), and the device is rotated with a step of 10° to measure its polarized photoresponse. The polarized photocurrents under the illuminations with the wavelengths of 0.52, 0.637, 0.785, 1.55, and 2.3 μm are presented in Supplementary Fig. 14, and the corresponding line shape curves are included in the Supplementary Figs. 15–19. To better illustrate the photoresponse anisotropy, here we define the net polarization current $\Delta I_p$:

$$\Delta I_p = I_p(\theta) - I_{p,min}(\theta) = I_p(\theta) - I_p(0°). \tag{7}$$

The net polarization photocurrents under the illuminations at 0.52, 1.55, and 2.3 μm are shown in Fig. 3a–c, respectively, and the results measured with other wavelengths are shown in Supplementary Fig. 20. Strong polarization photodetection is

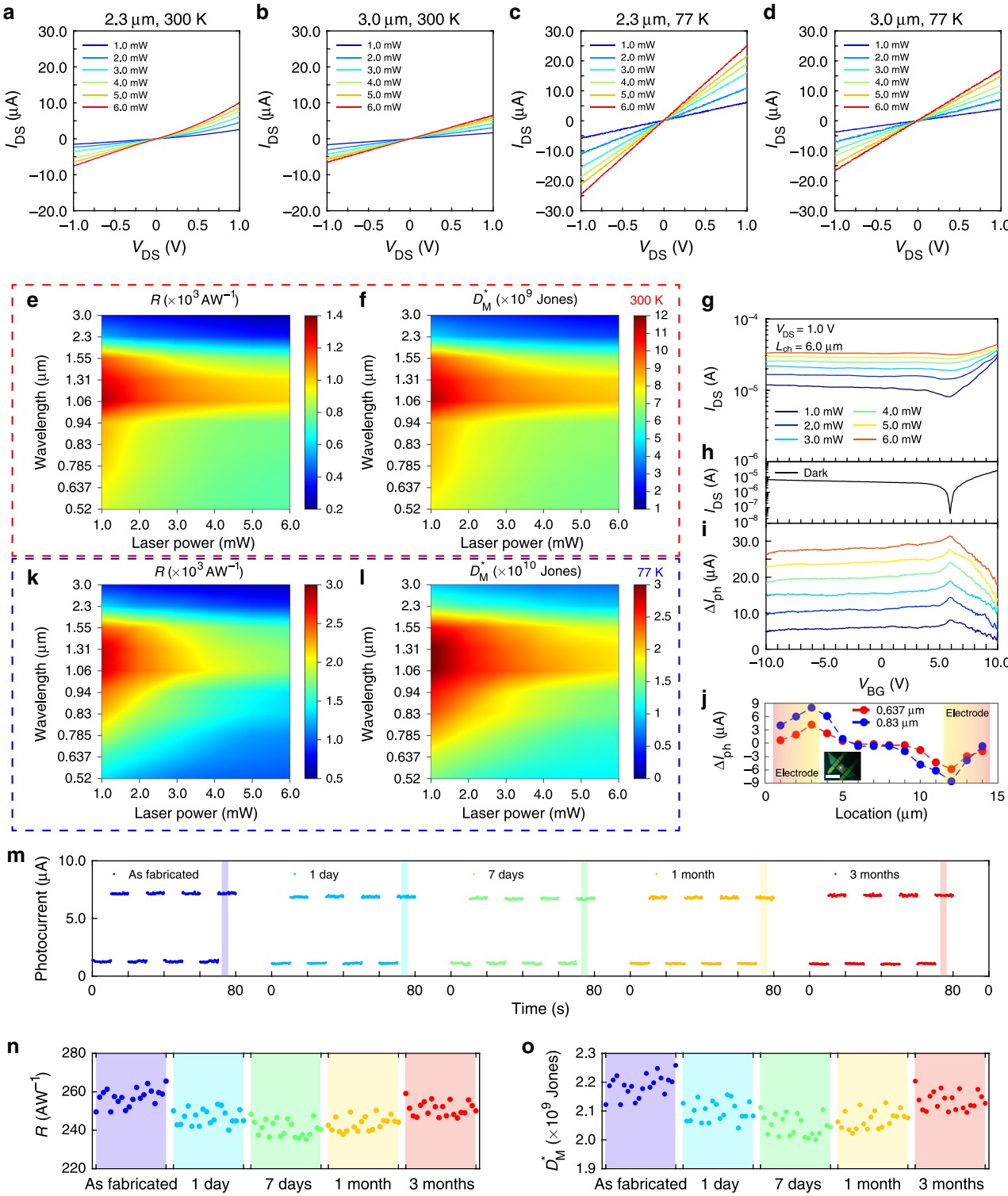

clearly observed to demonstrate that the Te photodetector can identify the slight differences between different polarized illuminations. The photocurrents under 1.0 V drain bias for different incident polarization angles are shown in Fig. 3d–f, corresponding to the incident wavelengths of 0.52, 1.55, and 2.3 μm, respectively. Even at 0° and 180° polarization direction where the absorption is relatively weaker, obvious photoresponse is still observed, which also ensures the high performances of the device for polarization applications. The visual polarized photoresponsivities are presented in polar diagrams, as shown in

Fig. 3g–i. The largest $R$ is $\sim 3.60 \times 10^2 \, \mathrm{AW^{-1}}$ under 0.52 μm illumination, $\sim 4.56 \times 10^2 \, \mathrm{AW^{-1}}$ under 1.55 μm illumination, and $\sim 3.03 \times 10^2 \, \mathrm{\mu AW^{-1}}$ under 2.3 μm illumination, respectively. Based on the results, the anisotropic ratios can reach up to $\sim 2.55$ and $\sim 2.39$, for the visible (0.52 μm) and NIR (1.55 μm) photoresponsivities. Importantly, a higher photoresponsivity anisotropic ratio of $\sim 7.58$ is realized under 2.3 μm illumination for Te-based photodetector, which is sufficient to achieve polarized imaging applications without the assistance of extrinsic modulation. The anisotropic ratio is also compared with the

**Fig. 2 Unpolarized broadband optoelectronic responses for Te nanoflake device. a, b** Photocurrent for the infrared range: 2.3 and 3.0 μm illumination under 300 K temperature, respectively, as a function of laser power and drain bias. **c, d** Photocurrent for the infrared range: 2.3 and 3.0 μm illumination under 77 K temperature, respectively, as a function of laser power and drain bias. **e, f** Pseudo-color mapping figures for $R$ and $D_M^*$, as a function of laser power and wavelength at room temperature (300 K). The drain bias is fixed at 1.0 V, and the gate bias is 0 V, the results are in accordance with the absorption characters of Te and indicate the high performance of Te nanoflake for NIR and MIR range. **g–i** Photoresponses under different gate biases. The drain bias is 1.0 V, and the incident wavelength is 3.0 μm. Panels (**g**) and (**h**) show the transfer curves under different incident powers and the dark state, and (**i**) shows the net photocurrents under different illumination powers based on current data in (**g**) and (**h**). The maximum photocurrent is achieved at the neutral point of ∼5.9 V gate bias for all the incident powers. **j** The photocurrent distribution in the Te channel under 0.637 and 0.83 μm illumination at 1.0 V drain bias and 0 V gate bias, with the photocurrent generated at the channel/electrode interface. The inset shows the image of the Te device with laser spot size of ∼3 μm, scale bar, 10 μm. **k, l** Pseudo-color mapping figures for $R$ and $D_M^*$, as a function of laser power and wavelength at 77 K. The drain bias is fixed at 1.0 V, and the gate bias is 0 V, much higher device performance is achieved under low temperature. **m** Stability verification, the photocurrent and dark current are measured at 1.0 V drain bias and 0 V gate bias, the selected illumination wavelength is 3.0 μm, with the laser power of 6.0 mW. The performance at five selected point-in-time: as fabricated, 1 day later, 7 days later, 1 month later, 3 months later, are not degraded, which indicates that the Te-based device is very stable at ambient conditions for a long time. **n, o** $R$ and $D_M^*$ for the measured range marked by shaded rectangles in (**m**), the high performance of Te is preserved for a long time.

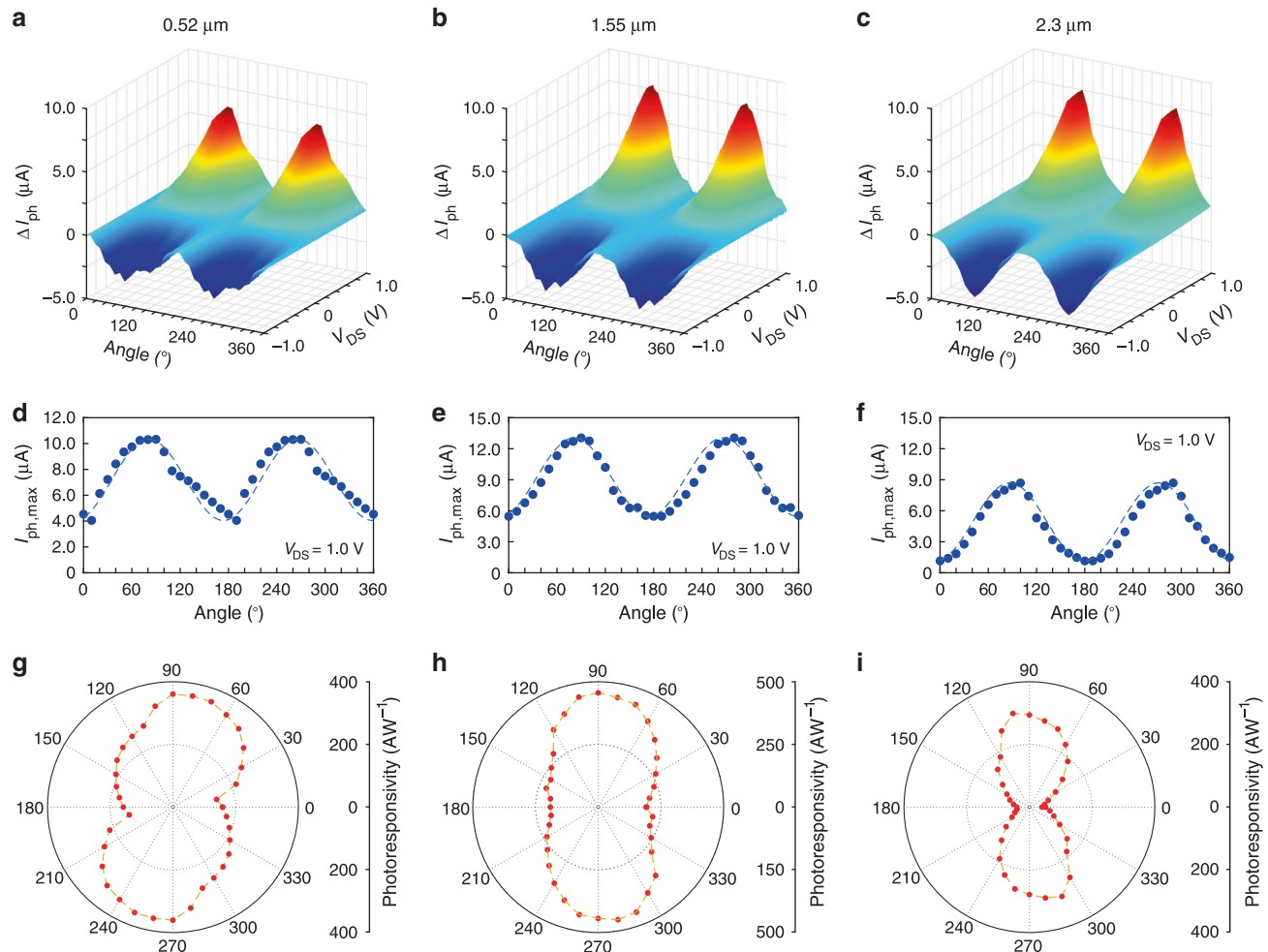

**Fig. 3 Polarized optoelectronic responses for Te nanoflake devices. a–c** The net polarized photocurrent $\Delta I_p$ for the incident wavelengths of 0.52, 1.55, and 2.3 μm at room temperature, under the incident power of 6.0 mW, respectively. **d–f** Polarized photocurrent under 1.0 V drain bias and 0 V gate bias for the incident wavelengths of 0.52, 1.55, and 2.3 μm, respectively. The dots are experimental data, and the dashed lines are fitted curves based on sine curve. **g–i** Polar diagram of the polarized photoresponsivity under 1.0 V drain bias and 0 V gate bias for the incident wavelengths of 0.52, 1.55, and 2.3 μm, respectively, and the anisotropic ratio are 2.55, 2.39, and 7.58 for the three wavelengths, respectively.

widely researched 2D materials such as black phosphorous, WTe$_2$, ReS$_2$, ReSe$_2$, etc. (Supplementary Table 3, Supplementary Fig. 27)[9,17,21,23,26,38,43,46–50], which is showing satisfactory polarization properties of Te. Some previous works have already demonstrated the optical anisotropy of Te, which also verify the anisotropic performances of Te, in accordance with our results[26,27]. Moreover, although black phosphorus and black phosphorus-arsenic may realize similar or higher anisotropic ratio[49,50], stable Te with a strong anisotropic ratio is more appealing for future polarization applications when taking the easy oxidization nature of these materials at ambient conditions into account.

**Polarized imaging applications of Te**. High photoresponsivity and detectivity are basic requirements for realistic imaging applications. For the Te photodetector, the unpolarized imaging results can be acquired under the illuminations with the wavelengths from 1.06 to 2.3 μm illuminations, as shown in Supplementary Fig. 21a. Under linear polarized illumination with the wavelength of 1.55 and 2.3 μm, the obvious polarization distinction is also observed in Supplementary Fig. 21b-c, when the device is under different orientation, to confirm the strongly polarized photosensitivity of the device. Based on Supplementary Fig. 21, the polarization extinction ratio of Te-based device under 2.3 μm illumination is reaching ~8, which is acceptable in the field of 2D materials. Based on absorption results, it can reach even higher extinction ratio at a longer wavelength, making it a potential material for polarization applications in the MIR range. Under stronger optical scattering conditions, such as foggy and cloudy weather, colloid and emulsion media, the incident light is severely scattered to obstruct imaging based on non-polarizing facilities[26,52]. To test the performance in scattering conditions, we have carried out imaging tests for the Te photodetector, without the integration of a complicated polarization filter, and the scattering media is placed on the surface of the target. Since no polarization filter is needed, the fabricated device can be used directly to get polarized images, while the traditional method by rotating the polarization filter is not applicable. According to the imaging method as shown the schematic in the inset of Fig. 4a, the scattering signal illuminated onto the photodetector is approximately linear polarized (light scattering mechanism in Supplementary Note 6), the polarized photocurrents along different orientations are collected by rotating the photodetector. The signal processing procedure calculates the DoLP (Fig. 4b), defined by Eqs. (8)–(11)

$$\text{DoLP} = \frac{\sqrt{S_1^2 + S_2^2}}{S_0}, \tag{8}$$

$$S_0 = I_{0°}(x, y) + I_{90°}(x, y), \tag{9}$$

$$S_1 = I_{0°}(x, y) - I_{90°}(x, y), \tag{10}$$

$$S_2 = I_{45°}(x, y) - I_{135°}(x, y), \tag{11}$$

where $I_{\theta}(x, y)$ refers to the polarized intensity detected at $\theta$ angle (more details in Supplementary Note 7)[6,51,52]. For the conventional division-of-focal-plane polarimeter (DoFP) structures, it requires at least four pixels to acquire the first three components of the Stokes vector to calculate one DoLP data (Supplementary Fig. 22)[5,6,52], which means that to get a $m \times n$ DoLP polarization image, it requires a $2m \times 2n$ device matrix. On the contrary, only a single-pixel Te photodetector is adequate to calculate a DoLP data, which means that it requires only a $m \times n$ device matrix to get a $m \times n$ DoLP polarization image. It suggests that the Te-based device needs fewer pixels to realize the same resolution for DoLP, making it easier to further promote the imaging resolution. To better demonstrate the advantage of polarized imaging based on anisotropic Te flakes, a non-polarization 2H-phase MoTe$_2$ photodetector is used to image the HUST and SITP target sheltered by the scattering media. Figure 4c shows the imaging results for the MoTe$_2$ photodetector with hardly distinguishable DoLP. However, the imaging results for the Te photodetector show much better contrast for the target after calculating the DoLP, as shown in Fig. 4d, e. At 2.3 μm illumination, the larger absorption anisotropic ratio can realize clearer polarization imaging, comparing the results with 1.55 μm illumination (Fig. 4f), demonstrating that the Te photodetector is promising for polarization imaging applications.

## Discussion

Imaging applications based on the DoFP structure are applicable to all regions from visible to LWIR, which has attracted significant attention. At present, most DoFP structures require a layer of polarization filter and a layer of photo-sensing array[5], making it requires four single-point devices with different polarization filtering directions to detect polarized signals along four different directions for the calculation of one pixel in DoLP. Theoretically, wire grid polarizers with matured fabrication technology can also reach a high polarization extinction ratio as only light polarized parallel to the grid can pass through the polarizer. As for 2D materials including Te, their intrinsic polarized photosensitivity can provide another route to realize polarization functionalities. For example, the intrinsic polarized photosensitivity of Te promotes polarization imaging without polarization filters. The photocurrent in one single-point device along different polarization orientations can be collected by rotating the Te device, for the calculation of DoLP. However, it is worth noting that the Te-based devices and many other 2D materials-based polarization devices are still at the infancy with some existing challenges at present. Currently, the fabrication of large-scale device arrays is a great challenge in the field of 2D materials. With the development of matured methods for producing large films, the Te photodetector array could be one promising option to realize the practical polarization imaging. This work serves as the first step toward the development of polarization imaging arrays based on 2D materials, which is of great significance.

It is also worth noting that a large photoresponse anisotropy of ~8 is realized under 2.3 μm illumination, and the anisotropic ratio can be further enhanced for longer wavelength according to the absorption results, promising Te as an ideal material for NIR and MIR polarized imaging devices to reach a higher DoLP. As one of the 2D materials, Te can be integrated with on-chip structures easily, such as waveguides, photonic crystals, and metasurfaces, to further enhance optoelectronic responses and increase the photoresponse anisotropy, which is worth more investigation. 2D Te is also applicable for flexible devices with polarization sensitivity, promising the high level of integration and flexibility[53]. In addition, most DoFP structure lacks the capability to detect circularly polarized signals, which is the fourth element of a Stokes vector:

$$S_3 = I_{RH}(x, y) - I_{LH}(x, y), \tag{12}$$

RH means right-handed light and LH means left-handed light, thus, to integrate Te with chiral meta-structures and make full use of its strong spin–orbit coupling, may be a promising path to fully understand the imaging of circularly polarized signals [54–57].

In conclusion, we have systematically studied the photo-detecting properties of quasi-2D tellurium photodetectors and used the photodetector to successfully realize polarized infrared imaging under scattering. High photoresponsivity and detectivity were achieved for illumination with the wavelengths from 0.52 to 3.0 μm. Due to its anisotropic crystal structure, the Te photodetector shows a photoresponsivity anisotropic ratio reaching up to ~8 under 2.3 μm illumination and can be even higher than ~10 under the MIR range illumination. Based on the anisotropic photoresponses, polarized imaging capability is preserved for quasi-2D Te-based photodetector under scattering environments, with the DoLP reaches over 0.8 at the wavelength of 2.3 μm. As a rising star in the field of 2D materials, after overcoming the challenge of controllable grown of large-scale tellurium film, Te photodetector is promising for the next-generation imaging facilities, which is significant for the development of national security, remote sensing, biological diagnosis, and optical engineering.

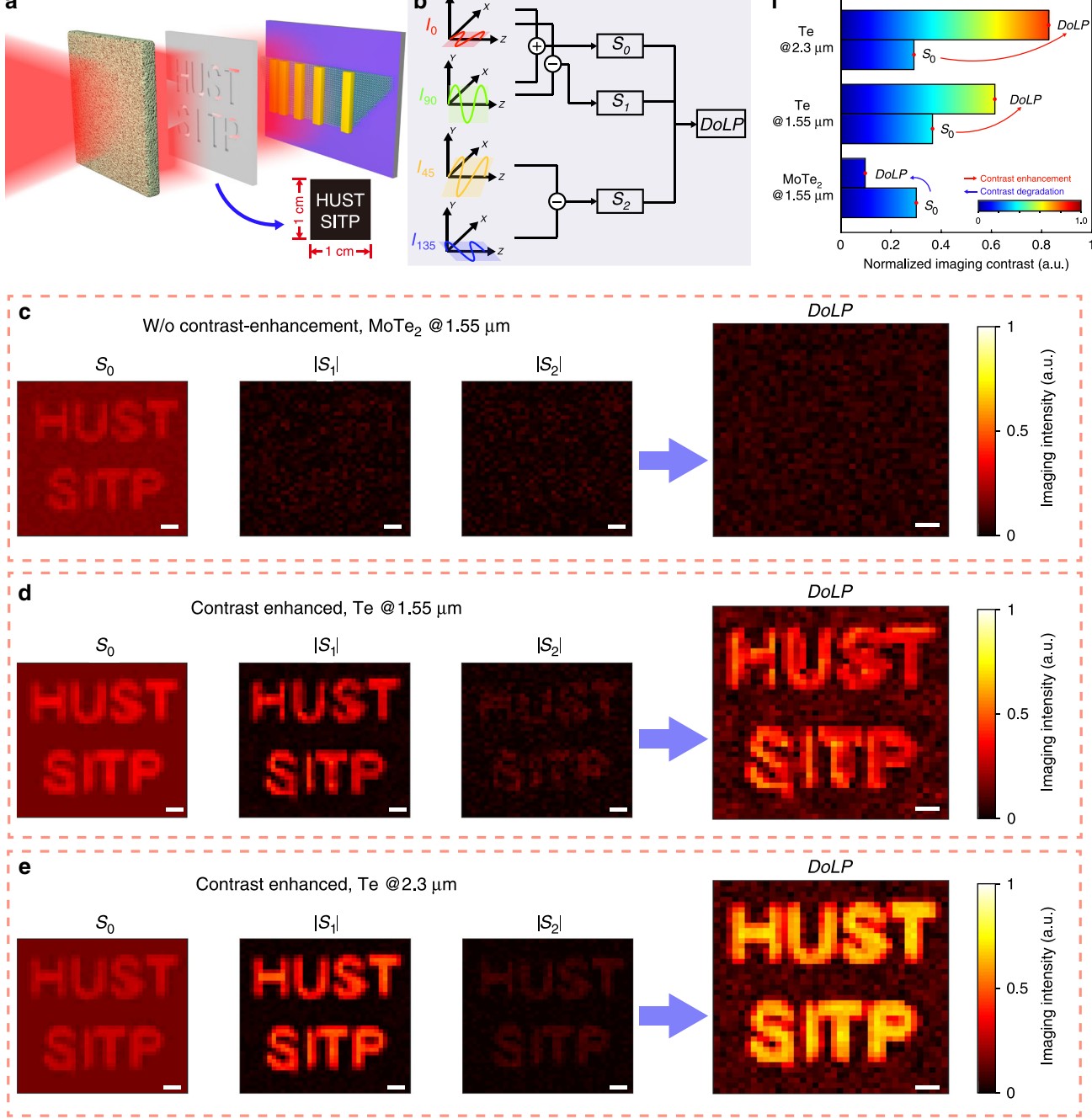

**Fig. 4 Polarization imaging applications for Te nanoflake device. a** Schematic of the imaging target, with a scattering media above the target. The inset is the schematic of the text pattern HUST and SITP. **b** Schematic of the polarization imaging mechanism for the calculation of DoLP. **c** Imaging DoLP results for the nonpolarized 2H-MoTe$_2$ device under 1.55 μm illumination for comparison. The illumination power is fixed at 1.0 mW. The DoLP is almost 0, indicating that MoTe$_2$ device shows no polarization detection ability, which is not applicable for polarized imaging under scattering environments. **d, e** Imaging DoLP results for the Te device under 1.55 and 2.3 μm illumination, respectively. The imaging contrast is strong for the target due to its intrinsic excellent polarization detection ability. Scale bars in (**c**)–(**e**), 1 mm. **f** Normalized imaging contrast of S$_0$ and DoLP. For MoTe$_2$, the contrast becomes lower due to its isotropic photoresponse along any direction, and for Te device, the contrast is higher after calculating DoLP, and DoLP enhancement under 2.3 μm illumination is higher than 1.55 μm illumination due to stronger absorption anisotropy.

## Methods

**Solution synthesis method.** The crystals are fabricated through a solution synthesis method based on ref. [27].

**Raman and AFM measurements.** Raman spectra is measured by LABRAM ARAMIS system (Horiba), the incident laser is 532 nm Nd:YAG laser, the scattering signal is reflected to the detector with 1/10 attenuation.

The polarized Raman spectra are measured by Alpha 300R confocal Raman imaging system (WITec), with 532 nm laser. The polarization directions of the

incident laser and the scattering signal are modulated to be parallel, then the sample is rotated in a step of 10° to measure the polarized Raman signal.

AFM data are measured by using Digital TNSTRUments/Veeco Dimension 3000 system in the tapping mode of 0.5 Hz.

**Polarized Raman calculations.** The polarized Raman intensity is calculated through Raman tensor analysis, details are included in the Supplementary Materials. Based on Raman tensor analysis, the direction with the highest $E_1$-TO and $A_1$

intensity can be determined to be the x-direction. Based on the calculation results, in Fig. 1d, we have defined the x-direction to be 0° and rotated the pattern.

**Absorption measurements**. The absorption spectra is measured at room temperature with confocal microscopes. The broadband laser is focused on the sample. At first, the total reflection $r_{total}$ on silver mirror is measured to confirm the profile of the laser source. Then the reflected signal on a bare silicon wafer $r_{SiO_2}$ is collected, finally, the reflected signal on the tellurium sample $r_{Te}$ is collected. Based on the following equations:

$$r_{SiO_2} = r_{total} - \alpha_{SiO_2}, \tag{12}$$

$$r_{Te} = r_{total} - \alpha_{Te} - \alpha_{SiO_2}, \tag{13}$$

where $\alpha_{SiO_2}$ is the absorption of the silicon wafer, $\alpha_{Te}$ is the absorption of the Te sample, and the absorption rate of the Te sample can be calculated:

$$A_{Te} = \frac{r_{SiO_2} - r_{Te}}{r_{total}}. \tag{14}$$

For polarized absorption measurements, a half-wave plate is added before the light source, the polarization direction is modulated to be along (y-direction, 0°) and perpendicular (x-direction, 90°) to the longer side of tellurium sample, the polarized absorption spectra is calculated by using the same method mentioned above.

**Device fabrication**. The device is fabricated by electron beam lithography (FEI Quanta 650 SEM and Raith Elphy Plus), vacuum thermal evaporation and lift-off process. The metal electrodes consist of 20 nm Ni and 130 nm Au.

**Electronic and optoelectronic measurements**. The unpolarized current data are measured by using a probe station system (SEMISHARE SE-6), a semiconductor characterization system (KEYSIGHT B1500A) and a home-made photocurrent mapping station at atmospheric pressure conditions. The photocurrent is measured in the range of 0.52–3.0 μm. In the range of 0.52–1.55 μm, the selected single-wavelength laser sources include 0.52 μm (Thorlabs, LP520-SF15), 0.637 μm (Thorlabs, LP637-SF70), 0.785 μm (Thorlabs, LP785-SF20), 0.83 μm (Thorlabs, LP830-SF30), 0.94 μm (Thorlabs, LP940-SF30), 1.06 μm (Thorlabs, LP1060-FC), 1.31 μm (Thorlabs, LPSC1310-FC), and 1.55 μm (Thorlabs, LPSC1550-FC). For 2.3 and 3.0 μm illumination, the output laser wavelength is acquired by filtering a supercontinuum broadband white laser source. The photocurrent from 0.52 to 0.94 μm are collected through the semiconductor characterization system, the laser is directly illuminated on the device without focusing, the diameter of the laser spot is determined through optical images, which is 1000 μm in radius. The photocurrent from 1.06 to 3.0 μm are collected through the home-made photocurrent mapping station, the laser is focused on the sample by a ×20 objective, so the radius is much smaller at 50 μm. In addition, the output laser power illuminated on the device throughout the mapping station is also decreased due to losses through the light path and lenses. The realistic light power illuminated on the device location is calibrated by using power meters (Thorlabs, S130C, S132C, S175C), the light power density is similar for photocurrent measurements (0.32, 0.64, 0.95, 1.27, 1.59, and 1.91 nW μm$^{-2}$ for 1.0, 2.0, 3.0, 4.0, 5.0, and 6.0 mW output power of the laser source, respectively).

**Polarized photocurrent measurements**. The polarized photocurrent is measured by using a home-made photocurrent mapping station, we have selected five wavelengths in our measurements based on the operating wavelength range of the half-wave plate, 0.52 μm (Thorlabs, LP520-SF15), 0.637 μm (Thorlabs, LP637-SF70), 0.785 μm (Thorlabs, LP785-SF20), 1.55 μm (Thorlabs, LPSC1550-FC), and 2.3 μm (filtered through a supercontinuum broadband white laser source). The laser spot is focused on the sample by a ×20 objective with the laser spot of 50 μm in radius from optical images collected through CCD camera, a half-wave plate is placed between the objective and our sample, the laser polarization direction is fixed, and initially parallel to the y-axis of our device, then the polarized photocurrent is acquired by rotating the device with a step of 10°.

**Response speed measurements**. The chopped laser is controlled by the laser diode/temperature controller (Thorlabs ITC4020). The photocurrent is measured by an oscilloscope for response speed analysis.

**Noise spectral density measurements**. The noise spectral density is measured by using a spectrum analyzer (SR770).

**Imaging application measurements**. The imaging measurements are carried out through a home-made imaging system. The light source is Fianium/NKT Photonics LLFT Contrast supercontinuum white laser source, the output wavelength is between 1.0 and 2.4 μm. For unpolarized imaging tests, we have selected four wavelengths of 1.06, 1.550, 2.0, and 2.3 μm. For polarized imaging tests, we have measured 1.55 and 2.3 μm by adding a half-wave plate between the target and the device to acquire linear polarized illumination, the laser polarization direction is fixed, and initially parallel to the y-axis of our device, the polarized imaging results are collected by rotating the device with a step of 10°. For scattering tests, the half-wave plate is removed from the light path, photocurrent at different orientations is also measured by rotating the device. One resistance is cascaded with the device, and the photocurrent changes are measured through changes of bias for the series resistance. The device location is controlled by two x-axis and y-axis step motor, by changing the device location, photocurrent signal is acquired for each pixel. Details for the measurements in Fig. 4 are also provided in the Supplementary Information.

## Data availability
The datasets for this study are available from the corresponding author on reasonable request.

## Code availability
The simulation codes for this study are available from the corresponding authors on reasonable request.

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

## Acknowledgements

The authors acknowledge funding by the National Natural Science Foundation of China (No. 61704061, 61974050, 61725505, 11734016), Natural Science Foundation of USA (No. 1538360, 1636101, 1741100). The synthesis of tellurene materials is supported by the National Science Foundation of USA under Grant CMMI-1762698. W.Z.W. and G.J.C. acknowledge the EFC-Future of Manufacturing grant from College of Engineering, Purdue University. The tellurene materials synthesized in this project are only for the fundamental studies reported here in this work. Program of Shanghai Subject Chief Scientist (No. 19XD1404100), Key Research Project of Frontier Science of CAS (No. QYZDB-SSW-JSC031). The authors acknowledge the assistance from Prof. Juejun Hu and Yifei Zhang at MIT during the absorption measurement.

## Author contributions

L.Y., G.J.C., and W.H. conceived the project and designed the experiments; Y.W. and W.W. grew the samples; L.T., X.H., and P.W. performed device fabrication and characterization; L.A. performed absorption measurements; L.T. analyzed the data; L.T. and X.H. prepared the paper; M.P., L.A., Q.S., Y.Z., G.Y., Z.L., F.Z., and F.W. provided many critical comments and analyses.

## Competing interests

The authors declare no competing interests.
