## [Peer Review File · Nature Communications]

Reviewers' comments:

Reviewer #1 (Remarks to the Author):

The authors present a quasi-2D tellurium-based detector for detection in the visible to 3um light. As the authors suggest this is a fast-moving area of research and the content of the paper is mostly informative / appropriate. They also show polarization dependent imaging by rotating their detector, which to the best of my knowledge is the first time that this has been shown (previous papers have just claimed that this is possible without demonstrating). However, to be considered for publication the article would need to address several major issues:

1. The reported detectivity at 3 um at room temperature is very high/impressive. It is higher than that reported for state-of-the-art commercial III-V/II-VI detectors at 3um at room temperature. This is more remarkable given the lower absorption in the Te detector compared to commercial detectors (which absorb close to 100% of MIR light) due to the Te optical anisotropy, the nm-scale of the Te flake and transmission into the Si/SiO₂ substrate. As such, it is likely that readers will challenge this measurement. To support their claim, I would recommend the Authors provide a full suite of data/plots used to extract the champion detectivity of $\sim 2.83 \times 10^{11}$ Jones @ 3um @ room temperature. This includes:

Details of the gate bias used for the champion detectivity measurement, is this at 0V bias? Was the detectivity calculated for incident light that is unpolarized or polarized? Is FigS37B measured with a bandwidth of 1Hz at the same gate bias as the responsivity measurement? More description of the champion detector itself, what is the channel length? can the authors provide a clear picture of the champion device?

2. The authors place a large emphasis on the polarization sensitivity of Te, stating in the manuscript that the reported Te optical anisotropy "is much stronger than widely researched materials such as black phosphorous, WTe₂, ReS₂, ReSe₂". This claim seems to be unfounded, there are several existing papers which show equivalent and even higher optical anisotropy on black phosphorus and perhaps black phosphorus-arsenic. See for example papers from the groups of Yaqiong Xu or Ali Javey among others. It should also be made clear that optical anisotropy has already been demonstrated for Te based photodetectors.

3. The last figure of the manuscript aims to showcase the intrinsic advantages of the Te detector in taking polarization images. However, to take the image the detector still must be mechanically rotated. It is not clearly explained how this provides a resolution advantage over a conventional polarization resolved camera (i.e. a standard MIR camera with a rotating polarizer)? In terms of camera complexity, it seems it would be simpler to rotate an isolated polarizer rather than having to rotate the entire detector FPA and supporting circuitry? can the polarization extinction ratio of the Te detector compete with standard wire grid polarizers?

Minor comments:

1. The statement "At lower temperature (77K), the photodetecting performances can be further enhanced due to higher net photocurrent for the device" is a bit confusing. Typically, improvements at lower temperatures arise due to lower noise. Can the authors clarify this or ideally provide data to explain?

2. The language / English used in some areas is difficult to follow particularly in the technical parts.

3. The Authors might want to clarify what they mean by "mid infrared". Some scientific fields classify the mid infrared as starting at 3 or 5 um and hence most of the measurements presented in this manuscript would not be considered mid infrared.

4. The title of the manuscript insinuates that high-resolution polarization imaging has been

demonstrated. The authors should include a scale bar in figure 4. It may instead be more accurate to explain within the text that this paper serves as a first step towards high resolution imaging and remove the claim from the title.

5. The absorption spectrum in Figure 1 should be converted to a real % value rather than arbitrary units. This plot should also be extended beyond 3um so the band edge can be confirmed.

6. There are 42 supplementary figures, most of which are not crucial to the key claims of the paper. In addition, as highlighted above some of the exciting claims of this paper are missing supporting data / information so the authors may want to consider condensing some of these plots.

Reviewer #2 (Remarks to the Author):

The authors have reported a quasi-2D tellurium infrared photodetector with polarization-dependent photoresponse. They have done systematical research and highlighted two aspects, high responsivity at mid-infrared range and high anisotropic photoresponse. The paper is informative and very well written. However, I have doubts about the novelty of this report, the perspective of this material as photodetector and the way the author calculated the detectivity. I will detail my doubts as follows.

#1, Novelty.

I notice a paper in *ACS Nano* 2018, 12, 7, 7253-7263, which has presented a similar work with this paper. That paper also shows wide photodetection range to 3 micrometre and polarization-dependent photoresponse by using tellurium with the same fabricating method. Compared with ACS Nano paper, this work reported higher detectivity, higher anisotropic and better data quality, but this does not justify the novelty from my point of view.

#2 Perspective of tellurium as photodetector

From the material side, people have known that bulk Tellurium has photoresponse for a long time. After the rise of nanomaterials, tellurium nanowires have also been intensively researched. However this material has not been used as candidate for infrared detection, because there are better solutions, for example, III-V compounds. The author may argue with the quasi-2D structure. However, from the structure side, in comparison with bulk tellurium, hydrothermal quasi-2D products in this paper may be even harder to integrate into photodetector arrays, as it is nearly impossible to form uniform film with hydrothermal method. The author mentioned one application of polarized imaging system, however, there are more concerns that I want to address below.

#2.1. Commercial imaging system can hardly rely on hydrothermal products, as it cannot produce uniform film as I mentioned above. Also because they simply have more defects in the crystals. These defects will generate more dark current and give rise to photogating effect.

#2.2. This 'photogating effect' has favoured photocurrent generation in this paper and is the main reason for the high responsivity. This mechanism has been confirmed in the paper. However, note that until now, there is no commercial imaging system based on this effect.

#2.3. The commercial polarization imaging system needs more anisotropic ratio than the value of 8 in this paper. In fact current value by wires are more than 20 [*Optics express*, 2010, 18(18): 19087- 19094]. Using anisotropic materials for polarization imaging system is, on the contrary, less feasible than the wires solution, as fabricating detector arrays with different material orientations is probably a nightmare.

#3. The detectivity and responsivity calculation.

#3.1. In the part of 'method 6', the author addressed the laser diameter as 'determined through optical images', which is quite confusing. If the beam is a focused beam by 20x objective as mentioned in 'method 7', the responsivity calculation can be totally wrong as the power should be the total power (as the beam is smaller than device area) instead of the one in the paper which is calculated by multiplying power density with area. Sadly, this is likely true as the responsivity

numbers in the 'polarization measurement' by 'method 7' are $3.6e2$ A/W at 520 nm and $3.03e2$ A/W at 2300 nm. These are very similar with $1.36e3$ A/W at 1060 nm in photocurrent measurement, which implies their similar measurement methods (both by focused beam), as focused and unfocused beam should have resulted in larger differences in the responsivity numbers. I hope the author could supply more details to prove I am wrong.

#3.2. However, even if the responsivity is right, I still have the following concerns.

#3.2.1. Why there is a plateau in the range of 0.1 to 1 Hz in Figure S36? The spectra should follow $1/f$ if it is $1/f$ noise as claimed in the paper.

#3.2.2. Why the noise spectra is very noisy, for example, ranging from $1e-26$ to $1e-21$ at 10 Hz?

#3.2.3. Even with the unknown plateau and large uncertainty in the noise spectra, there is still a clear trend of noise spectra density in Figure S36. By linearly fitting the curve, one could extract the noise spectra density $S_{n²}$ (with unit of $A^{²}/Hz$) with values of $8e-23$, $8e-24$ at 1 Hz and 10 Hz, respectively. Based on these values, the detectivity could be calculated precisely by using

$$D^* = R \cdot \sqrt{A/S_{n²}} \quad (C1)$$

Where R , A and $S_{n²}$ are responsivity, area in unit $cm^{²}$ and noise spectra density in unit $A/Hz^{^{1/2}}$. For responsivity value of $3.54e2$ A/W and area of 15 square micrometres, the detectivity should be $1.53e10$ Jones at 1Hz and $4.85e10$ Jones at 10 Hz. Both of these values are one order of magnitude smaller than the reported value of $2.83e11$ Jones in the paper.

The authors were one-step close to reach the precise equation of detectivity. However, they finally chose other equations (equation S18, S19) and did not specify where these equations are from. In comparison with equation C1, they actually replaced noise spectra density (in unit of $A^{²}/Hz$) with $2qI_{dark}$ in their equation S18. Therefore, it is not surprised that they have never specified the essential frequency when reporting detectivity.

I notice in 2D photodetection field, many people use approximate equations instead of the standard one. However, these approximate equations may have special preconditions, which in other cases may not stand.

Besides the above three major concerns, I have other concerns as follows.

#4. For the anisotropic electrical measurement, why did the author choose X and Y channel with different lengths as shown in Figure S10? In principle, they should be at the same lengths so their properties could be compared.

#5. Mistakes in SI, for example:

#5.1. In Note 2, line 4, the author indicated 'note 4', but it should be note 3.

#5.2. Reference 10 in SI should be about $Bi_{₂O_{₂}Se$ instead of GeP.

#5.3. Reference 37 in SI has unrecognized symbols.

Reviewer #3 (Remarks to the Author):

Review of "Stable high resolution mid infrared polarization imaging based on quasi-2D tellurium at room temperature"

The manuscript entitled "Stable high resolution mid infrared polarization imaging based on quasi-2D tellurium at room temperature" describes the use of quasi-2D Te (27nm thick) for high contrast and polarized mid-IR imaging. The manuscript has an excellent detail and analysis of the properties of the photodetectors and the device performance is of interest.

The authors state "Besides, we are aware of no report about mid infrared polarization imaging based on 2D materials at all" while there are no reports detailing imaging, there are reports of 2D material mid infrared polarization photodetectors (<https://doi.org/10.1021/acs.nanolett.6b01977>, <https://www.nature.com/articles/s41467-017-01978-3>) which should be cited within the

manuscript.

Based on this I believe the novelty of the paper lies in the ability to A) form 2D Te nanosheets that are air stable and fabricate a photodetector from them; and B) use these nanosheets specifically for photodetector based imaging. In this context, specifically case B) I believe the manuscript demonstrates suitable potential impact on the field, detailed analysis and methodology, to be suitable for publication after the following revisions:

1. The authors state that the quasi-2D Te nanosheets were grown by a solution phase method as described in their reference 25. Looking at the thickness of the nanosheet (Figure S4) this is approximately 27nm. In reference 25 the authors describe the fabrication route as capable of producing 2D nanosheets down to 3nm thickness. Why have the authors chosen to use 27nm thick Te in this study? Was any comparison done to thinner sheets? The novelty on the paper rests on the lack of MIR polarization imaging from 2D materials therefore the use of such a thick Te nanosheet needs to be addressed.
2. Comparing the material to other 2D MIR photodetectors such as Black Phosphorous, (which are sub 3nm thick), how does the device performance of the 27nm Te compare? Can this performance difference be attributed to the extra thickness of material increasing the amount of photons absorbed?
3. Table S1-S3 – it is hard to compare mobility values with vastly different materials thicknesses – can the authors add a column showing 2D material thickness for the measured mobility as this would provide context to the measured values – particularly reference 17 and 21
4. Figure S14 – do the band structure calculations assume a quasi-2D material?, can 26nm thick be treated as bulk for electronic purposes?
5. Can the authors comment on how many devices they characterized? If there are more than 1 are these all the same thickness? If there was only 1 - can the authors comment on how repeatable the data is?

Also, minor text/formatting changes are recommended.

- Figure 3. the degree symbol in D, E, and F is superscript formatted – it shouldn't be
- Figure 4, The scale bar in A (that applies to C D and E) is difficult to read, this should be enlarged or added to C, D, and E.
- Figure S4 – needs a scale bar or axis; can the authors do roughness analysis/provide a profile along the long axis of the flake

Reviewer 1:

The authors present a quasi-2D tellurium-based detector for detection in the visible to 3 μ m light. As the authors suggest this is a fast-moving area of research and the content of the paper is mostly informative / appropriate. They also show polarization dependent imaging by rotating their detector, which to the best of my knowledge is the first time that this has been shown (previous papers have just claimed that this is possible without demonstrating). However, to be considered for publication the article would need to address several major issues:

1. The reported detectivity at 3 μ m at room temperature is very high / impressive. It is higher than that reported for state-of-the-art commercial III-V/II-VI detectors at 3 μ m at room temperature. This is more remarkable given the lower absorption in the Te detector compared to commercial detectors (which absorb close to 100% of MIR light) due to the Te optical anisotropy, the nm-scale of the Te flake and transmission into the Si/SiO₂ substrate. As such, it is likely that readers will challenge this measurement. To support their claim, I would recommend the Authors provide a full suite of data/plots used to extract the champion detectivity of $\sim 2.83 \times 10^{11}$ Jones @ 3 μ m @ room temperature. This includes:

Details of the gate bias used for the champion detectivity measurement, is this at 0V bias? Was the detectivity calculated for incident light that is unpolarized or polarized? Is Fig. S37B measured with a bandwidth of 1Hz at the same gate bias as the responsivity measurement? More description of the champion detector itself, what is the channel length? can the authors provide a clear picture of the champion device?

Answer: Thanks for the reviewer's valuable comments. Indeed, the absorption of Te is below 100% ($\sim 56\%$, according to the result from Te absorption spectra), but the thickness of Te flake is ~ 27.5 nm, which can still provide effective absorption up to 3 μ m and limit transmission of light into Si/SiO₂ substrate to realize high performances, according to previous other works about 2D materials-based photodetectors (**Small** 13, 1700894 (2017); **ACS Nano** 12, 7253–7263 (2018); **Small methods** 2, 1700294 (2018);

Nat. Photon. 12, 601-607 (2018)). In addition, the high crystal quality of Te flake will ensure high photocurrent, assisted by a certain degree of photogain originated from high electron mobility and narrow channel width.

Based on the reviewer's good comments, we have provided more discussions about the experimental details in the revised manuscript and Supplementary information, to support our photodetecting measurements. For all photocurrent measurements, the gate bias is fixed at 0 V. The detectivity is measured under unpolarized incident light. The dark current in Figure S37B is also measured under 0V gate bias and 100 Hz bandwidth. Based on the last comment of reviewer 1 and 3rd comment of reviewer 2, we have revised the figures in the Supplementary Information, the noise density spectra of the Te device is re-measured, which is more concrete to characterize the device performances at dark state, and Figure S37B is removed from our revised Supplementary Information. The photodetecting performance under unpolarized illumination is measured for the device as shown in Fig. 1A, which is a clearer picture of the device provided in revised manuscript, and the electrode 2 and 4 in Fig. 1A are selected for the measurements, so the channel length of the device is $\sim 6 \mu\text{m}$, and the channel width is $\sim 2.5 \mu\text{m}$.

The revised parts in the manuscript and supplementary information are listed below:

1. Page 5, line 1-7:

The thickness of Te flake measured by atomic force microscope (AFM) is $\sim 27.5 \text{ nm}$ (Figure S1D), and its thicker thickness is beneficial to reach stronger absorption, leading to higher photocurrent, but the thickness should not be too thick, otherwise

the overlarge dark current will lead to poor photo-detectivity and other performance. Besides, the thicker Te flake possesses narrower bandgap⁸, which is beneficial to extend the detectable bandwidth to longer infrared wavelength.

2. Page 5, line 9-14:

To investigate its electrical properties in more detail, we have fabricated a Te-based FET device with five parallel electrodes marked by numbers from 1 to 5 as shown in the schematic of the device in Figure 1A, resulting in different channel lengths between electrodes, and the channel length between two electrodes is measured through optical images as shown in the right inset of Figure 1A.

3. Page 7, line 6-9:

The photoresponse performances are measured between the electrode 2 and 4 in Figure 1A, respectively, with the device channel length of $\sim 6 \mu\text{m}$ and channel width of $\sim 2.5 \mu\text{m}$, located at the center of the Te flake. The gate bias is fixed at 0 V, the drain bias is varied from -1.0 V to 1.0 V, and the incident lights are unpolarized.

4. Page 7, line 22-24:

The drain bias is fixed at 1.0 V and the gate bias is fixed at 0 V, the calculation details are described in the Supplementary Information notes 5.

5. Page 8, line 27-28:

The drain bias is also varied from -1.0 V to 1.0 V, and the gate bias is fixed at 0 V.

6. Page 9, line 17-18:

The drain bias is varied from -1.0 V to 1.0 V, the gate bias is fixed at 0 V,

7. Page 10, line 3-5:

Even at 0° and 180° polarization directions where the absorption are relatively weaker, obvious photoresponse is still observed, which also ensures the high performances of the device for polarization applications.

8. Page 24, line 8-9:

the drain current under the drain bias from -1.0 V to 1.0 V and the gate bias of 0 V,

9. Supplementary information, page 4, line 2-3:

Based on transfer curves in Figure 1B, here we focus on the curve for the longest channel length of $16 \mu\text{m}$, the drain bias is fixed at 1.0 V,

10. Supplementary information, page 18, line 3-5:

(C) Maximum on current at 10.0 V gate bias for different channel length, the drain bias is 1.0 V.

11. Supplementary information, page 23, line 5-6, line 8, line 10:

the gate bias is 0 V.

12. Supplementary information, page 29, line 5:

The gate bias is fixed at 0 V.

13. Supplementary information, page 40, line 2-7:

Figure S24 (A) Noise spectra density of the tellurium device. The drain bias is fixed at 0 V (red curve) and 1.0 V (blue curve), and the gate bias is 0 V. The noise is dominated by $1/f$ noise. (B)-(C) Dark current under drain bias from -1.0 V to 1.0 V at room temperature and 77 K, respectively, the gate bias is 0 V. It's also proved that the dark current is lower than the noise current, which leads to the overestimated calculated detectivity (D_C^*) comparing with the measured detectivity (D_M^*).

2. The authors place a large emphasis on the polarization sensitivity of Te, stating in the manuscript that the reported Te optical anisotropy “is much stronger than widely researched materials such as black phosphorous, WTe_2 , ReS_2 , $ReSe_2$ ”. This claim seems to be unfounded, there are several existing papers which show equivalent and even higher optical anisotropy on black phosphorus and perhaps black phosphorus-arsenic. See for example papers from the groups of Yaqiong Xu or Ali Javey among others. It should also be made clear that optical anisotropy has already been demonstrated for Te based photodetectors.

Answer: Thanks for the reviewer’s pointing out our inaccurate description about that the optical anisotropy of Te is at the forefront among various 2D materials. We have corrected this description in the revised manuscript. Indeed, Te cannot show distinct superiority of optical anisotropy in absorption spectrum compared with BP, although it shows stronger anisotropy than that of BP in Raman spectrum based on previous works (Table S3, supplementary information). But, different from Te with very high stability in the ambient conditions, BP or black phosphorus-arsenic are unstable in the

ambient conditions, leading to great obstruction in practical applications. It can also be proved by the stability test of Te-based photodetector in Figure 2, and this promises the subsequent long-time polarization imaging under ambient conditions.

In addition, thanks for the reviewer's good comments. Based on previous works about Te-based devices (Yaqiong Xu, *Nat. Electron.*, 2018, 1, 228 or Ali Javey, *ACS Nano*, 2018, 12, 7253.), the optical anisotropy has already been demonstrated in these work. In the revised manuscript, we have made it clear that optical anisotropy has already been demonstrated for Te based photodetectors. Noting that for Yaqiong Xu, *et al.*'s work, they have studied the anisotropy of Raman spectrum, and for Ali Javey, *et al.*'s work, they have studied the anisotropy of absorption spectrum and photocurrent. In our work, the anisotropic performance is the fundamental for polarization imaging applications as shown in Figure 4, which should be discussed at first before we refer to the polarization imaging parts in our opinion, so we have systematically studied the polarization sensitivity of Te, including in Raman spectrum, absorption spectrum, electrical properties, and optoelectronic response.

The revised parts in the manuscript are listed below:

1. Page 2, line 18-20:

Significantly, the Te photodetector has a large anisotropic ratio of ~ 8 for $2.3 \mu\text{m}$, ensuring high-contrast polarized imaging in scattering environment.

2. Page 9, line 8-11:

Although some 2D materials-based photodetectors are showing high MIR photoresponse, they cannot realize long lasting stable photodetecting under ambient environment, such as easily oxidized black phosphorus and black phosphorus-arsenic.

3. Page 10, line 10-14:

Importantly, a much higher photoresponsivity anisotropic ratio of ~ 7.58 realized under $2.3 \mu\text{m}$ illumination for Te-based photodetector is at the forefront, according to widely researched materials such as black phosphorous, WTe_2 , ReS_2 , ReSe_2 , etc. (Table S3, Figure S28)^{9, 17, 21, 23, 26, 38, 43, 46-48},

4. Page 10, line 15-17:

Some previous works have already demonstrated the optical anisotropy of Te, which

also verify the anisotropic performances of Te, in accordance with our results^{26, 27}.

5. Page 10, line 19-20:

Moreover, although black phosphorus and black phosphorus-arsenic may realize similar or higher anisotropic ratio, these materials can be oxidized easily in ambient conditions, thus stable Te with strong anisotropic ratio is more appealing for future polarization applications.

3. The last figure of the manuscript aims to showcase the intrinsic advantages of the Te detector in taking polarization images. However, to take the image the detector still must be mechanically rotated. It is not clearly explained how this provides a resolution advantage over a conventional polarization resolved camera (i.e. a standard MIR camera with a rotating polarizer)? In terms of camera complexity, it seems it would be simpler to rotate an isolated polarizer rather than having to rotate the entire detector FPA and supporting circuitry? Can the polarization extinction ratio of the Te detector compete with standard wire grid polarizers?

Answer: Thanks for the reviewer's significant comments. For a standard MIR camera with separated polarizers above sensing pixels, there're two main methods reported to achieve polarization images. The one is by rotating the detector, the other one is by rotating the polarizer, and these two methods are equivalent. For conventional polarization resolved camera, the imaging method is by fabricating optical gratings with different polarization direction onto each sensing pixel, and each pixel can only detect optical signals linearly polarized along one direction, which is determined by the optical gratings. Thus, at least four different optical gratings are required to get one DoLP data, which means that to get a $m \times n$ DoLP polarization image, it requires a $2m \times 2n$ device array (Figure S20 in the revised Supplementary information). So this method requires more complicated optical grating system to achieve polarization imaging, and optical crosstalk in the optical grating system is inevitable to limit the extinction ratio (Optics express, 2010, 18(18): 19087- 19094).

In our work, the Te-based photodetector can perform high contrast MIR polarization imaging under ambient at room temperature, without the help from

polarizer or optical grating, which is different from traditional polarization imaging devices, making the device fabrication easier. Indeed, there is no significant resolution advantage, compared with a standard MIR camera with a rotating polarizer. But the Te-based photodetector can achieve MIR polarization imaging without polarizer or optical grating system, leading to easier integration manufacturing process. Especially, it will promise this manufacturing process advantage for polarization imaging after realizing Te array photodetectors.

As described above, the method by rotating the entire detector FPA and supporting circuitry and the method by rotating an isolated polarizer are equivalent, which can both achieve polarization imaging. It actually can be simpler to rotate an isolated polarizer rather than having to rotate the entire detector FPA and supporting circuitry, while rotating the entire detector FPA and supporting circuitry is also not difficult to achieve by automation engineering. Moreover, the Te-based photodetector has no polarizer or optical grating system in our work, so by rotating the device is the only method to realize polarization imaging.

Based on the measured results about polarization photodetecting performances of Te-based device, the polarization extinction ratio is ~ 8 at illumination wavelength of $2.3 \mu\text{m}$, ~ 12 at illumination wavelength of $3.0 \mu\text{m}$, and even higher for longer wavelength, which is comparable with standard wire grid polarizers (Optics express, 2010, 18(18): 19087- 19094).

We have revised the manuscript carefully to explain the advantages of our Te polarization imaging device more clearly, and the revised parts in the manuscript are listed below:

1. Page 10, line 27-30; page 11, line 1-6:

Theoretically, wire grid polarizers can reach extremely high polarization extinction ratio as only light polarized parallel to the grid can pass through the polarizer, but for practical applications, the fabrication is rather complicated, the optical crosstalk is inevitable, and the transmitted light can scatter between the wire grid and the below device⁴⁹, making the polarization extinction ratio degraded to a level of $\sim 10^1$. Based on Figure S22, the polarization extinction ratio of Te-based device under $2.3 \mu\text{m}$

illumination is reaching ~ 8 , which is comparable with standard wire grid polarizers. Based on absorption results, it can reach even higher extinction ratio at longer wavelength, making it a potential material for polarization applications in the MIR range.

2. Page 11, line 11-13:

Since no polarization filter is needed, the device fabrication can be easier, and to get polarized images, traditional method by rotating the polarization filter is not applicable.

3. Page 11, line 25-29; page 12, line 1-3:

For the conventional division-of-focal-plane polarimeter (DoFP) structures, it requires at least four pixels to acquire Stokes vector components (Figure S23)^{5, 6, 50}, which means that to get a $m \times n$ DoLP polarization image, it requires at least $2m \times 2n$ device matrix, making it more complicated to enhance the imaging resolution. On the contrary, only a single pixel Te photodetector is adequate to calculate DoLP, which means that it requires only a $m \times n$ device matrix to get a $m \times n$ DoLP polarization image, making it easier to further promote the imaging resolution.

4. Page 12, line 22-26:

In other words, at least four single point devices are needed to detect polarized signals along different directions to achieve only one pixel in DoLP, which impedes the realization of miniaturization and larger pixel density. Besides, the detection direction is strictly limited by the polarization filter.

5. Page 13, line 6-12:

It's also worth noting that large photoresponse anisotropy of ~ 8 is realized under $2.3 \mu\text{m}$ illumination, and the anisotropic ratio can be further enhanced for longer wavelength according to the absorption results, promising Te as an ideal material for NIR and MIR polarized imaging devices. As one of 2D materials, Te can be integrated with on-chip structures easily, such as waveguides, photonic crystals and metasurfaces, to further enhance optoelectronic responses and increase the photoresponse anisotropy, which is worth more investigation.

6. Page 13, line 24-27:

Due to its anisotropic crystal structure, the Te photodetector shows a photoresponsivity anisotropic ratio reaching up to ~8 under 2.3 μm illumination and can be even higher than ~10 under the MIR range illumination.

Minor comments:

1. The statement “ At lower temperature (77K), the photodetecting performances can be further enhanced due to higher net photocurrent for the device” is a bit confusing. Typically, improvements at lower temperatures arise due to lower noise. Can the authors clarify this or ideally provide data to explain?

Answer: Thanks for the reviewer’s valuable comments. The statement here is really confusing. At lower temperature (77 K), the dark current is much lower comparing with that of at room temperature (Fig. S21C-D in the revised Supplementary information), leading to lower noise. We have revised the manuscript to make these statements clearer.

The revised parts are listed below:

1. Page 8, line 28-29:

At lower temperature (77K), the photodetecting performances can be further enhanced due to lower dark current, as shown in Figure 2C-2D

2. Supplementary information, page 16, line 3-4:

Under 77 K temperature, the dark current can be much lower comparing with that of at room temperature (Figure S24C), which is beneficial to realize higher detectivity.

2. The language / English used in some areas is difficult to follow particularly in the technical parts.

Answer: Thanks for the reviewer’s helpful comments. We have revised the whole manuscript thoroughly and carefully in language to make the discussions clearer. The revised parts are listed below:

1. Page 2, line 18-20:

Significantly, the Te photodetector has a large anisotropic ratio of ~8 for 2.3 μm ,

ensuring high-contrast polarized imaging in scattering environment.

2. Page 3, line 5-6:

At present, most PIRISs comprise three-dimensional semiconductor-based photodetectors integrated with complex polarization filters.

3. Page 3, line 18-19:

Hence, it's a feasible approach to develop state-of-the-art PIRIS devices by implementing in-plane anisotropic 2D materials.

4. Page 4, line 16-18:

Importantly, polarized infrared imaging tests under scattering environments have also been carried out for the Te-based photodetectors for the first time, without the integration of complex polarization filters.

5. Page 4, line 23-26:

Significantly, upon achieving design and fabrication of quasi-2D tellurium-based area-array, this photodetecting system is promising to be used into authentic facilities in future, which is worth more investigations.

6. Page 5, line 7-9:

Quasi-2D tellurium consists of a helical chain in the [0001] direction, with adjacent helical chains stacked together through a non-covalent bonding^{26-28, 32}, forming a strongly anisotropic crystal structure, as shown in the left inset of Figure 1A.

7. Page 5, line 10-14:

To investigate its electrical properties in more detail, we have fabricated a Te-based FET device with five parallel electrodes marked by numbers from 1 to 5 as shown in the schematic of the device in Figure 1A, resulting in different channel lengths between electrodes, and the channel length between two electrodes is measured through optical images as shown in the right inset of Figure 1A.

8. Page 6, line 19-24:

The measured absorption of Te is also anisotropic, as shown in Figure 1E. Strong absorption characters from NIR to MIR range is achieved under nonpolarized illumination (red curve), with a maximum absorption of ~55% located near 1.2 μm and absorption edge extended to more than 3.0 μm . We have also measured the

polarized absorption spectra with polarized illumination along the y axis and x axis direction, respectively.

9. Page 7, line 6-9:

The photoresponse performances are measured between the electrode 2 and 4 in Figure 1A, respectively, with the device channel length of $\sim 6 \mu\text{m}$ and channel width of $\sim 2.5 \mu\text{m}$, located at the center of the Te flake. The gate bias is fixed at 0 V, the drain bias is varied from -1.0 V to 1.0 V, and the incident lights are unpolarized.

10. Page 7, line 11-14:

Figure 2A-2B display the photocurrent of Te-based photodetector illuminated under $2.3 \mu\text{m}$ and $3.0 \mu\text{m}$ at 300 K temperature, respectively, showing its high photoresponse at NIR and MIR range (All photocurrent curves at 300 K temperature are included in Figure S7A-J).

11. Page 7, line 21-24:

Here we focus on the D_C^* to make it more convenient to compare the device performances with previous works. The drain bias is fixed at 1.0 V and the gate bias is fixed at 0 V, the calculation details are described in the Supplementary Information notes 5.

12. Page 8, line 1-12:

The thick Te flake (Figure S1D) also contributes to such high performances, as the transmitted light to the substrate is limited, leading to stronger absorption characters. The device also exhibits large EQE as shown in Figure 2G, which is originated from effective photon absorption and a large photoconductive gain. The large photoconductive gain is a result of ultrafast carrier transition time due to high mobility of Te and short channel length of $6.0 \mu\text{m}$, in addition, the high crystal quality with few defects can contribute to the photoconductive gain through shallow trapping of photo-induced carriers at band tail states^{44, 45}. We have also measured the net photocurrent under various gate bias for $1.06 \mu\text{m}$ illumination and 1.0 V drain bias at room temperature, the maximum net photocurrent is realized at the neutral point with ~ 5.9 V gate bias, verifying that there is no obvious photogating behavior for the device (Figure S9).

13. Page 8, line 15-25:

The fast response speed can also exclude the photogating effect. The carrier lifetime is extracted from transient photoluminescence decay curves of Te, as shown in Figure S11, based on carrier lifetime and transit time, the photogain (G) can be calculated, which is at the level of $\sim 10^3$. It's still worth noting that the calculated detectivity (D_C^*) can always overestimate the actual detectivity of 2D materials-based devices, as the dark current is lower than the actual noise current. As a result, we have also analyzed the measured detectivity (D_M^*) in the Supplementary Information Figure S12, the frequency is chosen to be 1 kHz based on the response speed measurements, the drain bias is at 1.0 V and the gate bias is 0 V. The highest D_M^* is 1.15×10^{10} Jones at $1.06 \mu\text{m}$, and the highest D_M^* is 3.01×10^9 Jones for $3.0 \mu\text{m}$.

14. Page 8, line 30; Page 9, line 1-2:

The R , D_C^* and EQE as function of the wavelength and laser power of the incident light, are summarized in Figure 2H-2J, respectively

15. Page 9, line 8-11:

Although some 2D materials-based photodetectors are showing high MIR photoresponse, they cannot realize long lasting stable photodetecting under ambient environment, such as easily oxidized black phosphorus and black phosphorus-arsenic.

16. Page 9, line 13-15:

The corresponding R and D_C^* in Figure 2L-2M, are calculated from the shaded rectangle region in Figure 2K, which also indicate the stable high performance of the device.

17. Page 10, line 3-5:

Even at 0° and 180° polarization directions where the absorption are relatively weaker, obvious photoresponse is still observed, which also ensures the high performances of the device for polarization applications.

18. Page 12, line 3-5:

To better demonstrate the advantage of polarized imaging based on anisotropic Te flakes, a non-polarization 2H-phase MoTe_2 photodetector is used to image the "HUST" and "SITP" target sheltered by the scattering media.

19. Page 12, line 22-26:

In other words, at least four single point devices are needed to detect polarized signals along different directions to achieve only one pixel in DoLP, which impedes the realization of miniaturization and larger pixel density. Besides, the detection direction is strictly limited by the polarization filter.

20. Page 13, line 1-4:

Currently, fabrication of large-scale device arrays is a great challenge in the field of 2D materials. To overcome this problem, one applicable method is by using targeted transfer method. However, this method is still very complicated, if matured grown method for large film can be developed,

21. Page 13, line 6-12:

It's also worth noting that large photoresponse anisotropy of ~ 8 is realized under $2.3 \mu\text{m}$ illumination, and the anisotropic ratio can be further enhanced for longer wavelength according to the absorption results, promising Te as an ideal material for NIR and MIR polarized imaging devices. As one of 2D materials, Te can be integrated with on-chip structures easily, such as waveguides, photonic crystals and metasurfaces, to further enhance optoelectronic responses and increase the photoresponse anisotropy, which is worth more investigation.

22. Page 13, line 13-16:

which is the fourth element of a Stokes vector:

$$S_3 = I_{RH}(x, y) - I_{LH}(x, y), \quad (12)$$

RH means right-handed light and LH means left-handed light,

23. Supplementary information, page 4, line 8-12:

For the longest channel length, drain current under different gate bias is also measured in Figure S2A and S2B, which is in accordance with transfer curves in Figure 1B. The maximum on current under gate bias of 10.0 V is summarized in Figure S2C, which is decreasing with longer channel length based on transfer curves in Figure 1B.

24. Supplementary information, page 9, line 11-13:

based on photoresponse and noise current results, here we define the detectivity extracted from this method to be the measured detectivity (D_M^*):

25. Supplementary information, page 9 line 28; page 10, line 1:

Then implement equation S20 into S18, the detectivity can be calculated by equation S21, here we refer this detectivity as the calculated detectivity (D_C^*):

3. The authors might want to clarify what they mean by “mid infrared”. Some scientific fields classify the mid infrared as starting at 3 or 5 μm and hence most of the measurements presented in this manuscript would not be considered mid infrared.

Answer: Thanks for the reviewer’s good comments. Yes, the mid infrared (MIR) is defined to start at different wavelength based on different applications. For most works, MIR is classified to start at 3 or 5 μm . In our work, we have measured the unpolarized photoresponse at 3 μm , which can reach the MIR range. But for polarized photodetecting and imaging applications, we have measured the photoresponse at 2.3 μm due to equipment limitations. But according to the absorption results, the Te flakes are capable to realize photoresponses to more than 3 μm , and the anisotropic ratio of absorption is higher than ~ 10 in the MIR range for Te, so we believe that Te can even reach higher performance for polarized imaging applications in the MIR range. In the revised manuscript, we have revised some statements for the experiments to make it more concrete for the measured wavelength range, and clarified our discussions about “mid infrared”.

The revised parts in the manuscript are listed below:

1. Page 2, line 9-13:

However, high-contrast polarized infrared imaging based on 2D materials has not yet been realized to confirm such feasibility. Here we systematically report the experimental demonstration of high-contrast polarized infrared imaging of a designed target obscured by scattering media based on anisotropic quasi-2D tellurium (Te) photodetector.

2. Page 2, line 18-20:

Significantly, the Te photodetector has a large anisotropic ratio of ~ 8 for 2.3 μm , ensuring high-contrast polarized imaging in scattering environment.

3. Page 4, line 14-18:

The anisotropic ratio of photoresponsivity is ~ 8 under the incident light with wavelength of $2.3 \mu\text{m}$, which is sufficient for potential high contrast polarized imaging. Importantly, polarized infrared imaging tests under scattering environments have also been carried out for the Te-based photodetectors for the first time, without the integration of complex polarization filters.

4. Page 6, line 29-24:

The measured absorption of Te is also anisotropic, as shown in Figure 1E. Strong absorption characters from NIR to MIR range is achieved under nonpolarized illumination (red curve), with a maximum absorption of $\sim 55\%$ located near $1.2 \mu\text{m}$ and absorption edge extended to more than $3.0 \mu\text{m}$. We have also measured the polarized absorption spectra with polarized illumination along the y axis and x axis direction, respectively.

5. Page 7, line 11-14:

Figure 2A-2B display the photocurrent of Te-based photodetector illuminated under $2.3 \mu\text{m}$ and $3.0 \mu\text{m}$ at 300 K temperature, respectively, showing its high photoresponse at NIR and MIR range (All photocurrent curves at 300 K temperature are included in Figure S7A-J).

6. Page 9, line 8-11:

Although some 2D materials-based photodetectors are showing high MIR photoresponse, they cannot realize long lasting stable photodetecting under ambient environment, such as easily oxidized black phosphorus and black phosphorus-arsenic.

7. Page 10, line 17-20:

Moreover, although black phosphorus and black phosphorus-arsenic may realize similar or higher anisotropic ratio, these materials can be oxidized easily in ambient conditions, thus stable Te with strong anisotropic ratio is more appealing for future polarization applications.

8. Page 10, line 27-30; page 11, line 1-6:

Theoretically, wire grid polarizers can reach extremely high polarization extinction ratio as only light polarized parallel to the grid can pass through the polarizer, but for practical applications, the fabrication is rather complicated, the optical crosstalk is

inevitable, and the transmitted light can scatter between the wire grid and the below device⁴⁹, making the polarization extinction ratio degraded to a level of $\sim 10^1$. Based on Figure S22, the polarization extinction ratio of Te-based device under $2.3 \mu\text{m}$ illumination is reaching ~ 8 , which is comparable with standard wire grid polarizers. Based on absorption results, it can reach even higher extinction ratio at longer wavelength, making it a potential material for polarization applications in the MIR range.

9. Page 11, line 11-13:

Since no polarization filter is needed, the device fabrication can be easier, and to get polarized images, traditional method by rotating the polarization filter is not applicable.

10. Page 13, line 6-12:

It's also worth noting that large photoresponse anisotropy of ~ 8 is realized under $2.3 \mu\text{m}$ illumination, and the anisotropic ratio can be further enhanced for longer wavelength according to the absorption results, promising Te as an ideal material for NIR and MIR polarized imaging devices. As one of 2D materials, Te can be integrated with on-chip structures easily, such as waveguides, photonic crystals and metasurfaces, to further enhance optoelectronic responses and increase the photoresponse anisotropy, which is worth more investigation.

11. Page 13, line 24-27:

Due to its anisotropic crystal structure, the Te photodetector shows a photoresponsivity anisotropic ratio reaching up to ~ 8 under $2.3 \mu\text{m}$ illumination and can be even higher than ~ 10 under the MIR range illumination.

4. The title of the manuscript insinuates that high-resolution polarization imaging has been demonstrated. The authors should include a scale bar in figure 4. It may instead be more accurate to explain within the text that this paper serves as a first step towards high resolution imaging and remove the claim from the title.

Answer: Thanks for the reviewer's significant comments. We have added scale bars in Figure 4 to make the imaging experiments more concrete. And we have corrected

the title to be “Stable **high contrast** mid infrared polarization imaging based on quasi-2D tellurium at room temperature”. In the revised manuscript, we have explained that our work is the first step towards high resolution polarized imaging in the field of 2D materials. The revised parts in the manuscript are listed below:

1. Title:

Stable high **contrast** mid infrared polarization imaging based on quasi-2D tellurium at room temperature

2. Page 10, line 27-30; page 11, line 1-6:

Theoretically, wire grid polarizers can reach extremely high polarization extinction ratio as only light polarized parallel to the grid can pass through the polarizer, but for practical applications, the fabrication is rather complicated, the optical crosstalk is inevitable, and the transmitted light can scatter between the wire grid and the below device, making the polarization extinction ratio degraded to a level of $\sim 10^1$.⁴⁷ Based on Figure S22, the polarization extinction ratio of Te-based device under $2.3 \mu\text{m}$ illumination is reaching ~ 8 , which is comparable with standard wire grid polarizers. Based on absorption results, it can reach even higher extinction ratio at longer wavelength, making it a potential material for polarization applications in the MIR range.

3. Page 11, line 11-13:

Since no polarization filter is needed, the device fabrication can be easier, and to get polarized images, traditional method by rotating the polarization filter is not applicable.

4. Page 11, line 25-29; page 12, line 1-3:

For the conventional division-of-focal-plane polarimeter (DoFP) structures, it requires at least four pixels to acquire Stokes vector components (Figure S23)^{5, 6, 50}, which means that to get a $m \times n$ DoLP polarization image, it requires at least $2m \times 2n$ device matrix, making it more complicated to enhance the imaging resolution. On the contrary, only a single pixel Te photodetector is adequate to calculate DoLP, which means that it requires only a $m \times n$ device matrix to get a $m \times n$ DoLP polarization image, making it easier to further promote the imaging

resolution.

5. Page 12, line 12-14:

Combined with the highly anisotropic performance of Te, this work may serve as the first step towards high resolution imaging based on 2D materials, which is of great significance.

6. Page 12, line 22-26:

In other words, at least four single point devices are needed to detect polarized signals along different directions to achieve only one pixel in DoLP, which impedes the realization of miniaturization and larger pixel density. Besides, the detection direction is strictly limited by the polarization filter.

7. Page 29, Figure 4C-E:

8. Page 29, line 10:

Scale bar in (C)-(E), 1 mm.

5. The absorption spectrum in Figure 1 should be converted to a real % value rather than arbitrary units. This plot should also be extended beyond 3 μm so the band edge

can be confirmed.

Answer: Thanks for the reviewer's valuable comments. We have re-measured the absorption spectrum of Te flake, and the revised absorption result is shown as Figure 1E in the revised manuscript. The revised parts in the manuscript are listed below:

1. Page 6, line 19-24:

The measured absorption of Te is also anisotropic, as shown in Figure 1E. Strong absorption characters from NIR to MIR range is achieved under nonpolarized illumination (red curve), with a maximum absorption of ~55% located near 1.2 μm and absorption edge extended to more than 3.0 μm . We have also measured the polarized absorption spectra with polarized illumination along the y axis and x axis direction, respectively.

2. Page 24, Fig. 1E:

6. There are 42 supplementary figures, most of which are not crucial to the key claims of the paper. In addition, as highlighted above some of the exciting claims of this paper are missing supporting data / information so the authors may want to consider condensing some of these plots.

Answer: Thanks for the reviewer's valuable comments. We have checked the supplementary information carefully, and rearranged the supplementary information.

The revised parts in the supplementary information are listed below:

1. Supplementary information, page 3, line 9-19:

Atomic force microscope (AFM) is also measured for the sample in Figure S1A, as shown in Figure S1D(i)-(iii). Figure S1D(ii) and S1D(iii) show the height distribution along two perpendicular directions marked by white and red dashed lines in Figure S1D(i), the thickness of Te is ~ 27.5 nm. We have chosen thick sample because the absorption edge can be redshifted for longer wavelength, which is beneficial for MIR photodetecting and imaging, and the absorption can be stronger for thick samples, transmitted light into the substrate can also be limited. To verify anisotropic electronic characters of Te, we have fabricated another device with two perpendicular pairs of electrodes, the channel lengths are $10 \mu\text{m}$ for both directions to ensure precise comparison in Figure 1C, the optical image of this device is shown in Figure S1E.

2. Supplementary information, page 4, line 2-3:

Based on transfer curves in Figure 1B, here we focus on the curve for the longest channel length of $16 \mu\text{m}$, the drain bias is fixed at 1.0 V,

3. Supplementary information, page 4, line 8-12:

For the longest channel length, drain current under different gate bias is also measured in Figure S2A and S2B, which is in accordance with transfer curves in Figure 1B. The maximum on current under gate bias of 10.0 V is summarized in Figure S2C, which is decreasing with longer channel length based on transfer curves in Figure 1B.

4. Supplementary information, page 9, line 11-13:

based on photoresponse and noise current results, here we define the detectivity extracted from this method to be the measured detectivity (D_M^*):

5. Supplementary information, page 9, line 18-25:

The noise spectra density of the device is measured as shown in Figure S24A, the noise at 0 V drain bias is lower than that of at 1.0 V drain bias, and the noise is dominated by flicker ($1/f$) noise which is originated from fluctuations of local electronic states. For our photocurrent measurements, the sampling frequency is much higher than 1 Hz, so the noise in our device should be flicker ($1/f$) noise, which is analogous to previous works about the 2D materials-based devices⁶. In the field of 2D materials-based devices, the noise current can be replaced by equation S20 to make it

simpler for the calculation of detectivity:

6. Supplementary information, page 9, line 28; page 10, line 1:

Then implement equation S20 into S18, the detectivity can be calculated by equation S21, here we refer this detectivity as the calculated detectivity (D_C^*):

7. Supplementary information, page 10, line 3-4:

Based on equation S21, it's noted that the working bandwidth is removed, so in most works, the bandwidth can be ignored to extract D_C^* data

8. Supplementary information, page 10, line 6-13:

The dark current can lead to overestimated detectivity comparing with the noise density spectra, to accurately characterize the device performances, here we have also extracted the measured detectivity D_M^* data based on equation S18. The drain bias is at 1.0 V, the bandwidth $\Delta f = 1000\text{Hz}$, the corresponding noise current density is $4.55 \times 10^{-11} \text{ AHz}^{1/2}$. As shown in Figure S10, the measured detectivity D_M^* is about 1-2 magnitude lower than the calculated detectivity D_C^* , the highest D_M^* is 1.15×10^{10} Jones at $1.06 \mu\text{m}$, and the highest D_M^* is 3.01×10^9 Jones for $3.0 \mu\text{m}$, which is still excellent for practical applications.

9. Supplementary information, page 10, line 14-16:

External quantum efficiency (EQE) can be calculated through equation S22, which measures the ratio of photogenerated carriers number over incident photon numbers in one second.

10. Supplementary information, page 10, line 20-28; page 11, line 1-3:

Photogain characterizes the ratio between the photo carrier's life time (τ_l) and transit time (τ_T) through the channel, which can be calculated from equation S23 and S24.

$$\tau_T = \frac{L^2}{\mu V_{DS}}, \quad (\text{S23})$$

$$G = \frac{\tau_l}{\tau_T}, \quad (\text{S24})$$

where L is the channel length, μ is the mobility and V_{DS} is the drain bias. From transient photoluminescence decay curve in Figure S11, the carrier's life time $\tau_l = \text{%%}$, then the photogain is calculated at the level of 10^5 . For our high quality Te, the high mobility and narrow channel length is beneficial for fast transit time

$\tau_T = 4 \times 10^{-10}$ s, and the long carrier's life time τ_l is a result of shallow trap at few defects due to band tail states in Te, so the photogain is high, which is not a result of photogating as we have discussed in the manuscript.

11. Supplementary information, page 11, line 5-19:

5.1 Calculation example

Here we have performed the photocurrent measurements under the illumination with the wavelength of 3.0 μm at room temperature for example to calculate the above figure of merits. From Figure 2B, under 1.0 mW laser illumination, the net photocurrent at 1.0 V drain bias is 1.6883 μA . The radius of the laser spot is 1000 μm , and the channel length between electrode 2 and 4 in Figure 1A, is $\sim 6 \mu\text{m}$, and the channel width is $\sim 2.5 \mu\text{m}$. Then the photoresponsivity is calculated by following:

$$R = \frac{I_{ph}}{PA} = \frac{1.6883 \times 10^{-6} \text{ A}}{1.0 \times 10^{-3} \text{ W} \times \frac{6 \times 2.5}{\pi \times 1000^2}} = 353.5967 \text{ A/W}. \quad (\text{S25})$$

The dark current is $I_{dark} = 1.066 \mu\text{A}$, then the calculated detectivity of 3.0 μm illumination is

$$\begin{aligned} D_C^* &= \sqrt{\frac{A}{2qI_{dark}}} R = \sqrt{\frac{6 \times 10^{-4} \times 2.5 \times 10^{-4} \text{ cm}^2}{2 \times 1.6 \times 10^{-19} \text{ C} \times 1.066 \times 10^{-6} \text{ A}}} \times 353.596 \text{ A/W} \\ &= 2.3448 \times 10^{11} \text{ cm W}^{-1} \text{ Hz}^{1/2}. \end{aligned} \quad (\text{S26})$$

The EQE of 3.0 μm illumination is

$$EQE = \frac{hc}{q\lambda} R = \frac{6.626 \times 10^{-34} \text{ Js} \times 3.0 \times 10^8 \text{ ms}^{-1}}{1.6 \times 10^{-19} \text{ C} \times 3 \times 10^{-6} \text{ m}} \times 353.5967 \text{ A/W} = 146.4332$$

(S27)

12. Supplementary information, page 16, line 3-8:

Under 77 K temperature, the dark current can be much lower comparing with that of at room temperature (Figure S23C), which is beneficial to realize higher detectivity. In addition, the net photocurrent is defined by equation S38:

$$I_{ph \text{ net}} = I_{ph} - I_{dark}, \quad (\text{S38})$$

which is also increased, comparing with the room temperature case. As a result, the photoresponsivity, detectivity and photogain are much higher at low temperature.

13. Supplementary information, page 17, Figure S1:

14. Supplementary information, page 18, Figure S2:

15. Supplementary information, page 19, Figure S3:

16. Supplementary information, page 20: Figure S4:

17. Supplementary information, page 23, Figure S7:

18. Supplementary information, page 24, Figure S8:

19. Supplementary information, page 25, Figure S9:

Figure S9 Net photocurrent under different gate bias. The drain bias is 1.0 V, and the illumination wavelength is 1.06 μm. The top panel shows the net photocurrent under different illumination power, and the bottom panel shows the transfer curve at dark state. The maximum photocurrent is achieved when the drain current at dark state is lowest at ~5.9 V gate bias.

20. Supplementary information, page 26, Figure S10:

21. Supplementary information, page 27, Figure S11:

Figure S11 Transient photoluminescence decay curves of Te. Excitation by a 450-nm laser and probing with 3500 nm. Based on the carrier lifetime and transit time, the photogain can be calculated to be $\sim 10^3$.

22. Supplementary information, page 28, Figure S12:

Figure S12 Measured detectivity (D_M^*) based on the noise density spectra at room temperature (300 K). the drain bias is at 1.0 V, and the bandwidth is $\Delta f = 1000\text{Hz}$. The noise current density is $4.55 \times 10^{-11} \text{ AHz}^{1/2}$. For most works in the field of 2D materials-based devices, the detectivity can be overestimated by implementing dark current into calculations.

23. Supplementary information, page 29, Figure S13:

24. Supplementary information, page 38, Figure S22:

25. Supplementary information, page 40, Figure S24:

Figure S24 (A) Noise spectra density of the tellurium device. The drain bias is fixed at 0 V (red curve) and 1.0 V (blue curve), and the gate bias is 0 V. The noise is dominated by $1/f$ noise. (B)-(C) Dark current under drain bias from -1.0 V to 1.0 V at room temperature and 77 K, respectively, the gate bias is 0 V. It's also proved that the dark current is lower than the noise current, which leads to the overestimated calculated detectivity (D_C^*) comparing with the measured detectivity (D_M^*).

Reviewer 2:

The authors have reported a quasi-2D tellurium infrared photodetector with polarization-dependent photoresponse. They have done systematical research and highlighted two aspects, high responsivity at mid-infrared range and high anisotropic photoresponse. The paper is informative and very well written. However, I have doubts about the novelty of this report, the perspective of this material as photodetector and the way the author calculated the detectivity. I will detail my doubts as follows.

#1, Novelty.

I notice a paper in ACS Nano 2018, 12, 7, 7253-7263, which has presented a similar work with this paper. That paper also shows wide photodetection range to 3 micrometre and polarization-dependent photoresponse by using tellurium with the same fabricating method. Compared with ACS Nano paper, this work reported higher detectivity, higher anisotropic and better data quality, but this does not justify the novelty from my point of view.

Answer: Thanks for the reviewer's good comments. The previous work from Ali Javey et al. emphasized that the Al₂O₃ optical cavity can enhance the photodetecting performance of Te-based photodetector. But for our work, the most important novelty is realizing high contrast polarization imaging applications based on our device under scattering environment, for the first time. Without of any polarizer or optical grating system, one device is adequate to calculate one DoLP data based on the Te photodetector, while traditional device with optical grating system requires at least four pixels to calculate one DoLP data. We have emphasized the advantages of polarization imaging based on our device.

The revised parts in the manuscript are listed below:

1. Page 2, line 9-13:

However, high-contrast polarized infrared imaging based on 2D materials has not yet been realized to confirm such feasibility. Here we systematically report the experimental demonstration of high-contrast polarized infrared imaging of a designed

target obscured by scattering media based on anisotropic quasi-2D tellurium (Te) photodetector.

2. Page 3, line 5-6:

At present, most PIRISs comprise three-dimensional semiconductor-based photodetectors integrated with complex polarization filters.

3. Page 3, line 18-19:

Hence, it's a feasible approach to develop state-of-the-art PIRIS devices by implementing in-plane anisotropic 2D materials.

4. Page 4, line 14-18:

The anisotropic ratio of photoresponsivity is ~ 8 under the incident light with wavelength of $2.3 \mu\text{m}$, which is sufficient for potential high contrast polarized imaging. Importantly, polarized infrared imaging tests under scattering environments have also been carried out for the Te-based photodetectors for the first time, without the integration of complex polarization filters.

5. Page 4, line 22-26:

Furthermore, we have also implemented polarized imaging for incident light over a wide wavelength range in more detail. Significantly, upon achieving design and fabrication of quasi-2D tellurium-based area-array, this photodetecting system is promising to be used into authentic facilities in future, which is worth more investigations.

6. Page 10, line 27-30; page 11, line 1-6:

Theoretically, wire grid polarizers can reach extremely high polarization extinction ratio as only light polarized parallel to the grid can pass through the polarizer, but for practical applications, the fabrication is rather complicated, the optical crosstalk is inevitable, and the transmitted light can scatter between the wire grid and the below device, making the polarization extinction ratio degraded to a level of $\sim 10^1$.⁴⁷ Based on Figure S22, the polarization extinction ratio of Te-based device under $2.3 \mu\text{m}$ illumination is reaching ~ 8 , which is comparable with standard wire grid polarizers. Based on absorption results, it can reach even higher extinction ratio at longer wavelength, making it a potential material for polarization applications in the MIR

range.

10. Page 11, line 11-13:

Since no polarization filter is needed, the device fabrication can be easier, and to get polarized images, traditional method by rotating the polarization filter is not applicable.

11. Page 11, line 25-29; page 12, line 1-3:

For the conventional division-of-focal-plane polarimeter (DoFP) structures, it requires at least four pixels to acquire Stokes vector components (Figure S23)^{5, 6, 48}, which means that to get a $m \times n$ DoLP polarization image, it requires at least $2m \times 2n$ device matrix, making it more complicated to enhance the imaging resolution. On the contrary, only a single pixel Te photodetector is adequate to calculate DoLP, which means that it requires only a $m \times n$ device matrix to get a $m \times n$ DoLP polarization image, making it easier to further promote the imaging resolution.

12. Page 12, line 12-14:

Combined with the highly anisotropic performance of Te, this work may serve as the first step towards high resolution imaging based on 2D materials, which is of great significance.

12. Page 12, line 22-26:

In other words, at least four single point devices are needed to detect polarized signals along different directions to achieve only one pixel in DoLP, which impedes the realization of miniaturization and larger pixel density. Besides, the detection direction is strictly limited by the polarization filter.

13. Page 13, line 6-12:

It's also worth noting that large photoresponse anisotropy of ~ 8 is realized under $2.3 \mu\text{m}$ illumination, and the anisotropic ratio can be further enhanced for longer wavelength according to the absorption results, promising Te as an ideal material for NIR and MIR polarized imaging devices. As one of 2D materials, Te can be integrated with on-chip structures easily, such as waveguides, photonic crystals and metasurfaces, to further enhance optoelectronic responses and increase the

photoresponse anisotropy, which is worth more investigation.

#2 Perspective of tellurium as photodetector

From the material side, people have known that bulk Tellurium has photoresponse for a long time. After the rise of nanomaterials, tellurium nanowires have also been intensively researched. However, this material has not been used as candidate for infrared detection, because there are better solutions, for example, III-V compounds. The author may argue with the quasi-2D structure. However, from the structure side, in comparison with bulk tellurium, hydrothermal quasi-2D products in this paper may be even harder to integrate into photodetector arrays, as it is nearly impossible to form uniform film with hydrothermal method. The author mentioned one application of polarized imaging system, however, there are more concerns that I want to address below.

Answer: Thanks for the reviewer's valuable comments. III-V compounds and many other 3D materials have been widely used for infrared detection, but these materials suffer from large noise at room temperature, thus requiring large cooling facilities to enhance detectivity. Recent years have seen many progresses in 2D materials-based photodetection applications, the ultrathin nature of 2D materials can further enhance photon-electron interactions due to confined geometry, making it easier to realize high photodetecting performances at room temperature, overcoming the shortcoming of required cooling facilities. Besides, 2D materials can be implemented with high flexibility, and integrated with nanostructures, such as metasurfaces, waveguides and photonic crystals, to further enhance photodetecting performances. Thus, 2D materials are promising for next generation photodetecting applications. As one kind of 2D materials, photodetecting applications based on quasi-2D Te is also worth to be investigated, and high infrared detecting performances can be achieved at room temperature based on recent works and our experiments¹⁻².

Indeed, the fabrication of photodetector arrays based on 2D materials is one challenge at present, and many groups have tried to find a way to integrate 2D materials into photodetector arrays³. For hydrothermal synthesis method, in Wu et

al.'s work², high quality, large flakes (more than 40 μm in size) of Te can be synthesized maturely, they have also carried out large-scale transfer to assemble thin 2D tellurene film through a Langmuir-Blodgett (LB) process, and a networked continuous thin film can be fabricated through ink-jet printing, making it possible to fabricate device arrays. In our work, we have demonstrated high contrast polarized imaging based on Te-based device in the field of 2D materials, under strong scattering environment. This opens up new possibilities for developing new polarized mid infrared imaging technology. Based on previous huge amount works about 2D materials-based photodetectors, it will promote more and more methods to try to achieve 2D materials with large scale for photodetector arrays. Here, our work is the first step toward photodetector arrays, which is very significant.

References:

1. Amani, M. et al. Solution-synthesized high-mobility tellurium nanoflakes for short-wave infrared photodetectors. *ACS Nano* 12, 7253-7263 (2018).
2. Wang, Y. et al. Field-effect transistors made from solution-grown two-dimensional tellurene. *Nat. Electron.* 1, 228-236 (2018).
3. J. Lee et al. Direct synthesis of a self assembled $\text{WSe}_2/\text{MoS}_2$ heterostructure array and its optoelectrical properties. *Adv. Mater.* 1904194 (2019).

The revised parts in the manuscript are listed below:

1. Page 13, line 1-4:

Currently, fabrication of large-scale device arrays is a great challenge in the field of 2D materials. To overcome this problem, one applicable method is by using targeted transfer method. However, this method is still very complicated, if matured grown method for large film can be developed,

#2.1. Commercial imaging system can hardly rely on hydrothermal products, as it cannot produce uniform film as I mentioned above. Also because they simply have more defects in the crystals. These defects will generate more dark current and give rise to photogating effect.

Answer: Thanks for the reviewer's good comments. Only relying on hydrothermal synthetic method, is very difficult to directly produce uniform film. In our work, we have used high crystal quality Te flakes (not films, with the size of more than 10 μm) to fabricate photodetectors for infrared polarization imaging, and based on the previous work¹ from Prof. Wu (he is the corresponding author in our work), the provided Te flakes show few defects nature based on high-angle annular dark-field scanning transmission electron microscopy (HAADF-STEM) results.

Generally, the numerous existing defects will trap the photo-induced carriers, leading to slow photoresponse speed and photogating behavior. But for photodetecting measurements, the dark current is slightly high due to the relative thick Te flake of our device. And the device shows a high response speed (~ 50 to $60 \mu\text{s}$) according to the time-resolved photoresponse measurement, differing from photogating induced slow response speed. Moreover, we have also measured the transfer curve of the Te device with and without illuminations to extract the maximum net photocurrent, the neutral point of the transfer curve is not shifted under on or off illumination², and as shown in Figure S9, the maximum photocurrent is achieved at the neutral point with a gate bias of $\sim 5.9 \text{ V}$, which also suggests no photogating behavior in our device^{3,4}. These indicate that no obvious photogating behaviors for the Te flakes.

Moreover, our work focuses on using quasi-2D Te flake-based photodetector to realize stable high contrast infrared polarization imaging under strong scattering environments, promising the first step towards high resolution polarized imaging in the field of 2D materials. This may attract more and more attentions to develop scientific solutions to produce uniform film based on 2D materials for commercial imaging systems in the future.

References:

1. Wang, Y. *et al.* Field-effect transistors made from solution-grown two-dimensional tellurene. *Nat. Electron.* 1, 228-236 (2018).
2. Long, M. *et al.* Room temperature high-detectivity mid-infrared photodetectors based on black arsenic phosphorus. *Sci. Adv.* 3, e1700589 (2017).

3. Guo, Q. et al. Black phosphorus mid-infrared photodetectors with high gain. *Nano Lett.* **16**, 4648-4655 (2016).
4. H. Fang, W. Hu. Photogating in low dimensional photodetectors. *Adv. Mater.* **4**, 170023 (2017).

The revised parts in the manuscript are listed below:

1. Page 8, line 3-12:

The device also exhibits large EQE as shown in Figure 2G, which is originated from effective photon absorption and a large photoconductive gain. The large photoconductive gain is a result of ultrafast carrier transition time due to high mobility of Te and short channel length of 6.0 μm , in addition, the high crystal quality with few defects can contribute to the photoconductive gain through shallow trapping of photo-induced carriers at band tail states^{44, 45}. We have also measured the net photocurrent under various gate bias for 1.06 μm illumination and 1.0 V drain bias at room temperature, the maximum net photocurrent is realized at the neutral point with ~ 5.9 V gate bias, verifying that there is no obvious photogating behavior for the device (Figure S9).

2. Page 8, line 12-19:

The response time is measured at 1.0 V drain bias, 0 V gate bias and 6.0 mW laser output power for several selected wavelengths, as summarized in Figure S10, the rise time is at the range of 48.7 μs to 62.7 μs , and the fall time is at the range of 62.7 μs to 78.0 μs . The fast response speed can also exclude the photogating effect. The carrier lifetime is extracted from transient photoluminescence decay curves of Te, as shown in Figure S11, based on carrier lifetime and transit time, the photogain (G) can be calculated, which is at the level of $\sim 10^3$.

3. Supplementary Materials, page 25:

Figure S9 Photocurrent under different gate bias. The drain bias is 1.0 V, and the illumination wavelength is 1.06 μm . The top panel shows the photocurrent under different illumination power, and the bottom panel shows the transfer curve at dark state. The maximum photocurrent is achieved when the drain current at dark state is lowest at ~ 5.9 V gate bias.

#2.2. This 'photogating effect' has favoured photocurrent generation in this paper and is the main reason for the high responsivity. This mechanism has been confirmed in the paper. However, note that until now, there is no commercial imaging system based on this effect.

Answer: Thanks for the reviewer's comments. We have not actually discussed the photocurrent generation mechanism in detail in the previous manuscript, and the reviewer pointed out the good comment about the photocurrent generation mechanism. So, we have described detailedly the mechanism in the revised manuscript.

The photogating effect is one common mechanism to enhance the photocurrent for photodetecting, due to the deep trapping of photo-induced carriers at defect centers in semiconductors. Therefore, the photogating effect will lead to the slow photoresponse speed and the shift of the transfer curve under illumination¹. However, in our work, the Te-based device shows fast photoresponse speed, and the neutral point of the transfer curve is not shifted under on or off illumination¹, the maximum

photocurrent is obtained at the neutral point of ~ 5.9 V gate bias, as shown in Figure S9, suggesting no obvious photogating behavior. In addition, the high photoresponsivity should be attributed to the high photoconductive gain due to the large carrier life time and short transit time. The high crystal quality Te flake is of the shallow trapping of photo-induced carriers at band tails dominates contributes to the long carrier life time^{1, 2}, and the high mobility, and narrow channel width of our device lead to fast transit time.

References:

1. Furchi, M. M. *et al.* Mechanisms of photoconductivity in atomically thin MoS₂. *Nano Lett.* **14**, 6165-6170 (2014).
2. Fu, Q. *et al.* Ultrasensitive 2D Bi₂O₂Se phototransistors on silicon substrate. *Adv. Mater.* **31**, 1804945 (2018).

The revised parts in the manuscript are listed below:

1. Page 8, line 3-12:

The device also exhibits large EQE as shown in Figure 2G, which is originated from effective photon absorption and a large photoconductive gain. The large photoconductive gain is a result of ultrafast carrier transition time due to high mobility of Te and short channel length of $6.0 \mu\text{m}$, in addition, the high crystal quality with few defects can contribute to the photoconductive gain through shallow trapping of photo-induced carriers at band tail states^{44, 45}. We have also measured the net photocurrent under various gate bias for $1.06 \mu\text{m}$ illumination and 1.0 V drain bias at room temperature, the maximum net photocurrent is realized at the neutral point with ~ 5.9 V gate bias, verifying that there is no obvious photogating behavior for the device (Figure S9).

2. Page 8, line 12-19:

The response time is measured at 1.0 V drain bias, 0 V gate bias and 6.0 mW laser output power for several selected wavelengths, as summarized in Figure S10, the rise time is at the range of $48.7 \mu\text{s}$ to $62.7 \mu\text{s}$, and the fall time is at the range of $62.7 \mu\text{s}$ to $78.0 \mu\text{s}$. The fast response speed can also exclude the photogating effect. The carrier lifetime is extracted from transient photoluminescence decay curves of Te, as shown

in Figure S11, based on carrier lifetime and transit time, the photogain (G) can be calculated, which is at the level of $\sim 10^3$.

3. Supplementary Materials, page 25:

Figure S9 Photocurrent under different gate bias. The drain bias is 1.0 V, and the illumination wavelength is 1.06 μm . The top panel shows the photocurrent under different illumination power, and the bottom panel shows the transfer curve at dark state. The maximum photocurrent is achieved when the drain current at dark state is lowest at ~ 5.9 V gate bias.

#2.3. The commercial polarization imaging system needs more anisotropic ratio than the value of 8 in this paper. In fact current value by wires are more than 20 [Optics express, 2010, 18(18): 19087- 19094]. Using anisotropic materials for polarization imaging system is, on the contrary, less feasible than the wires solution, as fabricating detector arrays with different material orientations is probably a nightmare.

Answer: Thanks for the reviewer's valuable comments. We have referred to the mentioned reference carefully, the wires are functioned as polarization filters (optical gratings). In theory, wire grids can realize a high anisotropic ratio at the level of 10^4 - 10^5 , since only light with the polarization direction parallel to the grid direction can be transmitted. However, for realistic applications, the anisotropic ratio can be degraded severely to a level of 10^1 , due to inevitable optical crosstalk through the

wire grid and light scattering between the wire grid and the sensing layer¹. The anisotropic ratio of wire grids is basically equivalent with that of Te flake, and the anisotropic ratio of Te-based device can reach ~ 8 under $2.3 \mu\text{m}$ illumination, but for longer wavelength, its anisotropic ratio can be even higher.

It is undeniable that the engineering problems of fabricating photodetector array based on anisotropic 2D materials are great challenge for the development of new generation photodetecting imaging at present. But for 2D materials, their intrinsic anisotropic nature promises higher integration and easier fabrication without polarizers or optical grating system. Moreover, based on the mechanism of DoLP detection discussed above, our Te-based device is also promising to enhance imaging resolution, compared with traditional polarization imaging system. In other ways, 2D materials also show their potential applications in flexible photodetectors, and they can be integrated easily with photonic crystals, metasurfaces to further enhance optoelectronic responses and the anisotropic ratio for polarization imaging. For our work, we have demonstrated the first step towards high resolution polarized imaging in the field of 2D materials, though it is far from commercialization. From the above statements, after matured methods for direct uniform film growth can be developed, and engineering problems of fabricating photodetector array based on anisotropic materials can be settled, this opens up new possibilities for developing new polarized mid infrared imaging technology.

References:

1. Gruev, V., Perkins, R. & York, T. CCD polarization imaging sensor with aluminum nanowire optical filters. *Opt. Express* **18**, 19087-19094 (2010).

The revised parts in the manuscript for the point that we have not implemented polarization filters for polarization imaging applications based on Te are listed below:

1. Page 10, line 27-30; page 11, line 1-6:

Theoretically, wire grid polarizers can reach extremely high polarization extinction ratio as only light polarized parallel to the grid can pass through the polarizer, but for practical applications, the fabrication is rather complicated, the optical crosstalk is inevitable, and the transmitted light can scatter between the wire grid and the below

device⁴⁹, making the polarization extinction ratio degraded to a level of $\sim 10^1$. Based on Figure S22, the polarization extinction ratio of Te-based device under $2.3 \mu\text{m}$ illumination is reaching ~ 8 , which is comparable with standard wire grid polarizers. Based on absorption results, it can reach even higher extinction ratio at longer wavelength, making it a potential material for polarization applications in the MIR range.

2. Page 11, line 11-13:

Since no polarization filter is needed, the device fabrication can be easier, and to get polarized images, traditional method by rotating the polarization filter is not applicable.

3. Page 11, line 25-29; page 12, line 1-3:

For the conventional division-of-focal-plane polarimeter (DoFP) structures, it requires at least four pixels to acquire Stokes vector components (Figure S23)^{5, 6, 48}, which means that to get a $m \times n$ DoLP polarization image, it requires at least $2m \times 2n$ device matrix, making it more complicated to enhance the imaging resolution. On the contrary, only a single pixel Te photodetector is adequate to calculate DoLP, which means that it requires only a $m \times n$ device matrix to get a $m \times n$ DoLP polarization image, making it easier to further promote the imaging resolution.

4. Page 12, line 12-14:

Combined with the highly anisotropic performance of Te, this work may serve as the first step towards high resolution imaging based on 2D materials.

5. Page 12, line 22-26:

In other words, at least four single point devices are needed to detect polarized signals along different directions to achieve only one pixel in DoLP, which impedes the realization of miniaturization and larger pixel density. Besides, the detection direction is strictly limited by the polarization filter.

6. Page 13, line 1-4:

Currently, fabrication of large-scale device arrays is a great challenge in the field of 2D materials. To overcome this problem, one applicable method is by using targeted

transfer method. However, this method is still very complicated, if matured grown method for large film can be developed,

7. Page 13, line 6-12:

It's also worth noting that large photoresponse anisotropy of ~ 8 is realized under $2.3 \mu\text{m}$ illumination, and the anisotropic ratio can be further enhanced for longer wavelength according to the absorption results, promising Te as an ideal material for NIR and MIR polarized imaging devices. As one of 2D materials, Te can be integrated with on-chip structures easily, such as waveguides, photonic crystals and metasurfaces, to further enhance optoelectronic responses and increase the photoresponse anisotropy, which is worth more investigation.

8. Page 13, line 24-27:

Due to its anisotropic crystal structure, the Te photodetector shows a photoresponsivity anisotropic ratio reaching up to ~ 8 under $2.3 \mu\text{m}$ illumination and can be even higher than ~ 10 under the MIR range illumination.

#3. The detectivity and responsivity calculation.

#3.1. In the part of 'method 6', the author addressed the laser diameter as 'determined through optical images', which is quite confusing. If the beam is a focused beam by 20x objective as mentioned in 'method 7', the responsivity calculation can be totally wrong as the power should be the total power (as the beam is smaller than device area) instead of the one in the paper which is calculated by multiplying power density with area. Sadly, this is likely true as the responsivity numbers in the 'polarization measurement' by 'method 7' are $3.6 \times 10^2 \text{ A/W}$ at 520 nm and $3.03 \times 10^2 \text{ A/W}$ at 2300 nm. These are very similar with $1.36 \times 10^3 \text{ A/W}$ at 1060 nm in photocurrent measurement, which implies their similar measurement methods (both by focused beam), as focused and unfocused beam should have resulted in larger differences in the responsivity numbers. I hope the author could supply more details to prove I am wrong.

Answer: Thanks for the reviewer's valuable comments. For photocurrent measurements, more details have been included in the method 6 and method 7 to

make it clearer.

For the unpolarized photocurrent measurement, the experiments are carried out by probe station with a semiconductor characterization system for 0.52 μm to 0.94 μm illumination, the light source is directly illuminated on the sample without focusing, and the light spot radius is 1000 μm from optical images. For other wavelengths, the experiments are carried out by a home-made photocurrent mapping station, the laser is focused through a $\times 20$ objective, and the laser spot radius is 50 μm from optical images, which is comparable with the device channel area. Although there are two different light spot radius, the laser output power illuminated on the device for method 7 is much lower than that for method 6, due to the higher power losses through the home-made photocurrent mapping station. Before our calculation, we have ~~checked~~ calibrated the laser output power illuminated on the device location using a power meter, and the effective power density illuminated on the device is similar for both method 6 and method 7, which is 0.32 $\text{nW}/\mu\text{m}^2$, 0.64 $\text{nW}/\mu\text{m}^2$, 0.95 $\text{nW}/\mu\text{m}^2$, 1.27 $\text{nW}/\mu\text{m}^2$, 1.59 $\text{nW}/\mu\text{m}^2$, 1.91 $\text{nW}/\mu\text{m}^2$ for 1.0 mW, 2.0 mW, 3.0 mW, 4.0 mW, 5.0 mW and 6.0 mW illumination, respectively.

We have revised the methods part to make our discussions clearer. The revised parts in the revised manuscript are listed below:

1. Page 15, line 28-29; page 16, line 1-11:

The photocurrent from 0.52 μm to 0.94 μm are collected through semiconductor characterization system, the laser is directly illuminated on the device without focusing, the diameter of the laser spot is determined through optical images, which is 1000 μm in radius. The photocurrent from 1.06 μm to 3.0 μm are collected through home-made photocurrent mapping station, the laser is focused on the sample by a $\times 20$ objective, so the radius is much smaller at 50 μm . In addition, the output laser power illuminated on the device throughout the mapping station is also decreased due to losses through the light path and lenses. The realistic light power illuminated on the device location is calibrated by using power meters (Thorlabs, S130C, S132C, S175C), the light power density is similar for photocurrent measurements (0.32 $\text{nW}/\mu\text{m}^2$, 0.64 $\text{nW}/\mu\text{m}^2$, 0.95 $\text{nW}/\mu\text{m}^2$, 1.27 $\text{nW}/\mu\text{m}^2$, 1.59 $\text{nW}/\mu\text{m}^2$, 1.91 $\text{nW}/\mu\text{m}^2$ for

1.0 mW, 2.0 mW, 3.0 mW, 4.0 mW, 5.0 mW and 6.0 mW output power of the laser source, respectively).

2. Page 16, line 14-20:

The polarized photocurrent is measured by using a home-made photocurrent mapping station, we have selected five wavelengths in our measurements based on the operating wavelength range of the half-wave plate, 0.52 μm (Thorlabs, LP520-SF15), 0.637 μm (Thorlabs, LP637-SF70), 0.785 μm (Thorlabs, LP785-SF20), 1.55 μm (Thorlabs, LPSC1550-FC) and 2.3 μm (filtered through a supercontinuum broad band white laser source). The laser spot is focused on the sample by a $\times 20$ objective with the laser spot of 50 μm in radius from optical images collected through CCD camera,

#3.2. However, even if the responsivity is right, I still have the following concerns.

#3.2.1. Why there is a plateau in the range of 0.1 to 1 Hz in Figure S36? The spectra should follow $1/f$ if it is $1/f$ noise as claimed in the paper.

Answer: Thanks for the reviewer's significant comments. The noise spectra should be discussed in more detail to make our explanations clearer. Based on the previous works about 2D materials-based devices, there exists two kinds of noises. One is generation-recombination (g-r) noise, which is caused by the fluctuations of carrier density. The g-r noise is flat in noise spectra density, and always exists for low frequency. The other noise is $1/f$ noise, which is originated from fluctuations of local electronic states. For most cases, noise of 2D materials-based device should be dominated by $1/f$ noise.

However, for our device, we have noted that the noise spectra is very noisy, we think that the connections between our device and the equipment may have a great influence during our measurements, so it's still hard to determine whether the origin of the plateau in the range of 0.1 to 1 Hz is really from g-r noise. Besides, based on the photoresponse speed tests, the rise and fall time is at the level of $\sim 50 \mu\text{s}$ (Figure S9 in the revised Supplementary information), it's necessary to extend the noise spectra to higher frequencies to make our experimental results more convincing. So, we have re-measured new noise spectra for the same device in the revised manuscript

and discussed the origin of noise based on the new noise spectra. From the result, the noise is determined by 1/f noise under both 0 V and 1.0 V drain bias from 0.1 Hz to 1000 Hz. It's also worth noting that all the measurements are carried out under a frequency higher than 1 Hz, so we think that the device characters are dominated by 1/f noise. We have revised the statements about the noise spectra in the revised manuscript and Supplementary information, which are listed below:

1. Page 7, line 21-22:

Here we focus on the D_C^* to make it more convenient to compare the device performances with previous works.

2. Page 8, line 15-25:

The fast response speed can also exclude the photogating effect. The carrier lifetime is extracted from transient photoluminescence decay curves of Te, as shown in Figure S11, based on carrier lifetime and transit time, the photogain (G) can be calculated, which is at the level of $\sim 10^3$. It's still worth noting that the calculated detectivity (D_C^*) can always overestimate the actual detectivity of 2D materials-based devices, as the dark current is lower than the actual noise current. As a result, we have also analyzed the measured detectivity (D_M^*) in the Supplementary Information Figure S12, the frequency is chosen to be 1 kHz based on the response speed measurements, the drain bias is at 1.0 V and the gate bias is 0 V. The highest D_M^* is 1.15×10^{10} Jones at $1.06 \mu\text{m}$, and the highest D_M^* is 3.01×10^9 Jones for $3.0 \mu\text{m}$.

3. Supplementary information, page 9, line 11-13:

based on photoresponse and noise current results, here we define the detectivity extracted from this method to be the measured detectivity (D_M^*):

4. Supplementary information, page 9, line 18-25:

The noise spectra density of the device is measured as shown in Figure S24A, the noise at 0 V drain bias is lower than that of at 1.0 V drain bias, and the noise is dominated by flicker (1/f) noise which is originated from fluctuations of local electronic states. For our photocurrent measurements, the sampling frequency is much higher than 1 Hz, so the noise in our device should be flicker (1/f) noise, which is analogous to previous works about the 2D materials-based devices⁶. In the field of 2D

materials-based devices, the noise current can be replaced by equation S20 to make it simpler for the calculation of detectivity:

5. Supplementary information, page 9, line 28; page10, line 1:

Then implement equation S20 into S18, the detectivity can be calculated by equation S21, here we refer this detectivity as the calculated detectivity (D_C^*):

6. Supplementary information, page 10, line 3-4:

Based on equation S21, it's noted that the working bandwidth is removed, so in most works, the bandwidth can be ignored to the extract D_C^* data

7. Supplementary information, page 10, line 6-13:

The dark current can lead to overestimated detectivity comparing with the noise density spectra, to accurately characterize the device performances, here we have also extracted the measured detectivity D_M^* data based on equation S18. The drain bias is at 1.0 V, the bandwidth $\Delta f = 1000\text{Hz}$, the corresponding noise current density is $4.55 \times 10^{-11} \text{ AHz}^{1/2}$. As shown in Figure S10, the measured detectivity D_M^* is about 1-2 magnitude lower than the calculated detectivity D_C^* , the highest D_M^* is 1.15×10^{10} Jones at $1.06 \mu\text{m}$, and the highest D_M^* is 3.01×10^9 Jones for $3.0 \mu\text{m}$, which is still excellent for practical applications.

8. Supplementary information, page 40, Figure S24:

Figure S24 (A) Noise spectra density of the tellurium device. The drain bias is fixed at 0 V (red curve) and 1.0 V (blue curve), and the gate bias is 0 V. The noise is dominated by $1/f$ noise. (B)-(C) Dark current under drain bias from -1.0 V to 1.0 V at room temperature and 77 K, respectively, the gate bias is 0 V. It's also proved that the dark current is lower than the noise current, which leads to the overestimated calculated detectivity (D_C^*) comparing with the measured detectivity (D_M^*).

#3.2.2. Why the noise spectra is very noisy, for example, ranging from $1e-26$ to $1e-21$ at 10 Hz?

Answer: Thanks for the reviewer's valuable comments. The noisy noise spectra may be due to disturbances from the poor connections between the device and the equipment. Besides, based on the photoresponse speed tests, the rise and fall time is at the level of $\sim 50 \mu s$ (Figure S9 in the revised Supplementary information), it's necessary to extend the noise spectra to higher frequencies to make our experimental results more convincing. So, we have re-measured the noise spectra for the same device in the revised manuscript and discussed the origin of noise based on the new spectra. The revised parts in the Supplementary information are listed below:

1. Supplementary information, page 40, Figure S24:

Figure S24 (A) Noise spectra density of the tellurium device. The drain bias is fixed at 0 V (red curve) and 1.0 V (blue curve), and the gate bias is 0 V. The noise is dominated by $1/f$ noise. (B)-(C) Dark current under drain bias from -1.0 V to 1.0 V at room temperature and 77 K, respectively, the gate bias is 0 V. It's also proved that the dark current is lower than the noise current, which leads to the overestimated calculated detectivity (D_C^*) comparing with the measured detectivity (D_M^*).

#3.2.3. Even with the unknown plateau and large uncertainty in the noise spectra, there is still a clear trend of noise spectra density in Figure S36. By linearly fitting the curve, one could extract the noise spectra density S_n^2 (with unit of A^2/Hz) with values of $8e-23$, $8e-24$ at 1 Hz and 10 Hz, respectively. Based on these values, the detectivity could be calculated precisely by using

$$D^* = R^* \sqrt{A/S_n} \quad (C1)$$

Where R, A and S_n are responsivity, area in unit cm^2 and noise spectra density in unit $\text{A/Hz}^{1/2}$. For responsivity value of $3.54\text{e}2 \text{ A/W}$ and area of 15 square micrometres, the detectivity should be $1.53\text{e}10$ Jones at 1Hz and $4.85\text{e}10$ Jones at 10 Hz. Both of these values are one order of magnitude smaller than the reported value of $2.83\text{e}11$ Jones in the paper.

The authors were one-step close to reach the precise equation of detectivity. However, they finally chose other equations (equation S18, S19) and did not specify where these equations are from. In comparison with equation C1, they actually replaced noise spectra density (in unit of A^2/Hz) with $2qI_{\text{dark}}$ in their equation S18. Therefore, it is not surprised that they have never specified the essential frequency when reporting detectivity.

I notice in 2D photodetection field, many people use approximate equations instead of the standard one. However, these approximate equations may have special preconditions, which in other cases may not stand.

Answer: Thanks for the reviewer's important comments. To calculate detectivity, the most accurate method is based on equation C1 as the reviewers have mentioned, here we define the detectivity extracted from C1 to be measured detectivity (D_M^*).

$$D_M^* = \frac{\sqrt{A\Delta f}}{NEP} = \frac{R\sqrt{A\Delta f}}{I_n} \quad (A1)$$

I_n is the noise current, here the responsivity is normalized to the device area (A) and working bandwidth Δf , so the detectivity can be calculated with the implementation of $I_n^{-1.4}$. For most cases, the noise in 2D materials-based FET devices is dominated by 1/f noise, so the noise current can be replaced by the equation A2 to estimate the detectivity in a simpler way, which is the calculation method implemented in most works about 2D materials-based devices:

$$i_{N,e} = \sqrt{2qI_{\text{dark}}\Delta f}. \quad (A2)$$

Here q is the electron charge, and I_{dark} is the dark current of the device without laser illumination.

For our photocurrent measurements, the noise is dominated by 1/f noise from the

remeasured noise density spectra (Supplementary information, Figure S21A), so the equation A2 could be applicable for our case. Based on equation A1 and A2, the detectivity is defined by equation A3, here we refer this detectivity to be calculated detectivity (D_C^*):

$$D_C^* = \frac{R\sqrt{A\Delta f}}{\sqrt{2qI_{dark}\Delta f}} = \frac{R\sqrt{A}}{\sqrt{2qI_{dark}}} \quad (A3)$$

Based on equation A3, we can find that to extract calculated detectivity (D_C^*), the bandwidth Δf can be ignored, so we have not specified the essential frequency when reporting such detectivity. The calculated detectivity (D_C^*) can be used to characterize the device performances in the field of 2D materials. In addition, as most works have used equation A3 to characterize detectivity, to make it more convenient to compare device performances in Figure S24 and Table S2, we still use the calculated detectivity (D_C^*) in the manuscript.

It's also worth noting that for most cases, the 1/f noise can be higher than the dark current measured for the device, so the calculated detectivity (D_C^*) would be overestimated¹. This is the case where the reviewer has stated in his comments. To make our performance characterization more convincing, we have used equation C1 to calculate the measured detectivity (D_M^*) based on the newly measured noise density spectra in the revised Supplementary information Figure S12, and discussed in detail about the overestimated D_C^* in the revised manuscript as the reviewer has concerned. Both D_M^* and D_C^* results are checked carefully in the revised manuscript.

References:

1. Lin, T., Wang, J. Strategies toward high-performance solution-processed lateral photodetectors. *Adv. Mater.* 1901473 (2019).
2. Xie, C., Yan, F. Flexible photodetectors based on novel functional materials. *Adv. Mater.* 13, 1701822 (2017).
3. Xie, C., Mak, C., Tao, X., Yan, F. Photodetectors based on two-dimensional layered materials beyond graphene. *Adv. Mater.* 27, 1603886 (2017).
4. Wang, J. et al. Recent progress on localized field enhanced two-dimensional

material photodetectors from ultraviolet-visible to infrared. *Small* **13**, 1700895 (2017).

The revised parts in the manuscript and Supplementary Information are listed below:

1. Page 7, line 17-20:

we have summarized the major parameters of photoresponsivity (R), calculated detectivity (D_C^*) and external quantum efficiency (EQE) of the photodetector at room temperature by using the pseudo-color mapping,

2. Page 7, line 21-22:

Here we focus on the D_C^* to make it more convenient to compare the device performances with previous works.

3. Page 8, line 16-19:

The carrier lifetime is extracted from transient photoluminescence decay curves of Te, as shown in Figure S11, based on carrier lifetime and transit time, the photogain (G) can be calculated, which is at the level of $\sim 10^3$.

4. Page 8, line 19-25:

It's still worth noting that the calculated detectivity (D_C^*) can always overestimate the actual detectivity of 2D materials-based devices, as the dark current is lower than the actual noise current. As a result, we have also analyzed the measured detectivity (D_M^*) in the Supplementary Information Figure S12, the frequency is chosen to be 1 kHz based on the response speed measurements, the drain bias is at 1.0 V and the gate bias is 0 V. The highest D_M^* is 1.15×10^{10} Jones at $1.06 \mu\text{m}$, and the highest D_M^* is 3.01×10^9 Jones for $3.0 \mu\text{m}$.

5. Page 8, line 30; Page 9, line 1-2:

The R , D_C^* and EQE as function of the wavelength and laser power of the incident light, are summarized in Figure 2H-2J, respectively (scatter diagrams are included in Figure S8D-F, S13K-M)

6. Supplementary information, page 9, line 11-13:

based on photoresponse and noise current results, here we define the detectivity extracted from this method to be the measured detectivity (D_M^*):

7. Supplementary information, page 9, line 18-25:

The noise spectra density of the device is measured as shown in Figure S24A, the noise at 0 V drain bias is lower than that of at 1.0 V drain bias, and the noise is dominated by flicker (1/f) noise which is originated from fluctuations of local electronic states. For our photocurrent measurements, the sampling frequency is much higher than 1 Hz, so the noise in our device should be flicker (1/f) noise, which is analogous to previous works about the 2D materials-based devices⁶. In the field of 2D materials-based devices, the noise current can be replaced by equation S20 to make it simpler for the calculation of detectivity:

8. Supplementary information, page 9, line 1; page 10, line 1:

Then implement equation S20 into S18, the detectivity can be calculated by equation S21, here we refer this detectivity as the calculated detectivity (D_C^*):

9. Supplementary information, page 10, line 3-4:

Based on equation S21, it's noted that the working bandwidth is removed, so in most works, the bandwidth can be ignored to extract D_C^* data

10. Supplementary information, page 10, line 6-13:

Figure S24B. The dark current can lead to overestimated detectivity comparing with the noise density spectra, to accurately characterize the device performances, here we have also extracted the measured detectivity D_M^* data based on equation S18. The drain bias is at 1.0 V, the bandwidth $\Delta f = 1000\text{Hz}$, the corresponding noise current density is $4.55 \times 10^{-11} \text{ AHz}^{1/2}$. As shown in Figure S10, the measured detectivity D_M^* is about 1-2 magnitude lower than the calculated detectivity D_C^* , the highest D_M^* is 1.15×10^{10} Jones at $1.06 \mu\text{m}$, and the highest D_M^* is 3.01×10^9 Jones for $3.0 \mu\text{m}$, which is still excellent for practical applications.

11. Supplementary information, page 10, line 14-16:

External quantum efficiency (EQE) can be calculated through equation S22, which measures the ratio of photogenerated carriers number over incident photon numbers in one second.

12. Supplementary information, page 10, line 20-28; page 11, line 1-3:

Photogain characterizes the ratio between the photo carrier's life time (τ_l) and transit

time (τ_T) through the channel, which can be calculated from equation S23 and S24.

$$\tau_T = \frac{L^2}{\mu V_{DS}}, \quad (S23)$$

$$G = \frac{\tau_l}{\tau_T}, \quad (S24)$$

where L is the channel length, μ is the mobility and V_{DS} is the drain bias. From transient photoluminescence decay curve in Figure S11, the carrier's life time $\tau_l = \sim 900 \text{ ns}$. For our high quality Te, the high mobility and narrow channel length is beneficial for fast transit time $\tau_T = 3.838 \times 10^{-10} \text{ s}$, and the long carrier's life time τ_l is a result of shallow trap at few defects due to band tail states in Te. The photogain is at a high level of ~ 2300 , which is not a result of photogating as we have discussed in the manuscript.

13. Supplementary Information, page 28, Figure S12:

Figure S12 Measured detectivity (D_M^*) based on the noise density spectra at room temperature (300 K). the drain bias is at 1.0 V, and the bandwidth is $\Delta f = 1000 \text{ Hz}$. The noise current density is $4.55 \times 10^{-11} \text{ AHz}^{1/2}$. For most works in the field of 2D materials-based devices, the detectivity can be overestimated by implementing dark current into calculations.

Besides the above three major concerns, I have other concerns as follows.

#4. For the anisotropic electrical measurement, why did the author choose X and Y channel with different lengths as shown in Figure S10? In principle, they should be at the same lengths so their properties could be compared.

Answer: Thanks for the reviewer's valuable comments. In the previous manuscript,

the device in Figure S10 exhibits different channel lengths along X and Y direction, to characterize the anisotropic electronic properties, the current is normalized by the channel lengths for both directions. To make the anisotropic electrical measurement more convincing, we have re-fabricated a new device with same lengths along X and Y channel (Figure S1E, revised Supplementary information), the channel lengths are 10 μm for both directions, and measured the electrical properties along X and Y direction. The new transfer curves are shown in Fig. 1C(i), drain current under zero gate bias is shown in the inset. The on/off ratio and mobility along two perpendicular directions are shown in Fig. 1C(ii), and Fig 1C(iii) shows the anisotropic ratio of on/off ratio and mobility. The electronic performances along two directions are related with the anisotropic structure of Te flake.

The revised parts in the manuscript and supplementary information are listed below:

1. Page 5, line 23-24:

with the electrical conductivity anisotropic ratio (along x axis and y axis) of ~ 2.24 under gate bias of 0 V

2. Page 5, line 26-27:

the field effect mobility anisotropic ratio is ~ 1.69 (Figure 1C(iii))

3. Page 24, Figure 1C:

4. Page 24, line 9-10:

the current along x axis is ~ 2.24 times of the current along y axis.

5. Supplementary information, page 17, Figure S1E:

(E) Optical images of the Te FET device for anisotropic electrical performance measurements. The channel width is $\sim 10 \mu\text{m}$. Scale bar, $10 \mu\text{m}$.

#5. Mistakes in SI, for example:

#5.1. In Note 2, line 4, the author indicated 'note 4', but it should be note 3.

#5.2. Reference 10 in SI should be about $\text{Bi}_2\text{O}_2\text{Se}$ instead of GeP.

#5.3. Reference 37 in SI has unrecognized symbols.

Answer: Thanks for pointing out our mistakes. We have checked the supplementary

information thoroughly and corrected some mistakes. For #5.2, the reference about Bi₂O₂Se should be 17, and the reference 10 is about GeP. The revised parts in the Supplementary information are listed below:

1. Supplementary information, page 4, line 5-6:

From the transfer curve, the calculated field effect mobility is $\mu = \sim 900 \text{ cm}^2\text{V}^{-1}\text{s}^{-1}$ (details included in Supplementary Information, notes 3),

2. Supplementary information, page 42, Figure S26:

3. Supplementary information, page 45:

Table S1 Field effect mobility of different 2D materials

Material	Thickness	Mobility(cm ² V ⁻¹ s ⁻¹)	References
Tellurium	27.5 nm	938	This work
GeP	4.3 nm	200	10
Tellurium	16 nm	700	11
MoS ₂	5 nm	28	12
ReS ₂	28 layers	79.1	13
WS ₂	3.5-5 nm	60	14
Black phosphorus	5 nm	286	15
Black phosphorus	10 nm	984	16
Bi ₂ O ₂ Se	5 nm	200	17
Tellurium	12.3 nm	419	21

4. Supplementary information, page 51, line 21-22:

37. Li, L. *et al.* Highly in plane anisotropic 2D GeAs₂ for polarization sensitive photodetection. *Adv. Mater.* **30**, 1804541 (2018).

Reviewer 3:

Review of “Stable high resolution mid infrared polarization imaging based on quasi-2D tellurium at room temperature”

The manuscript entitled “Stable high resolution mid infrared polarization imaging based on quasi-2D tellurium at room temperature” describes the use of quasi-2D Te (27nm thick) for high contrast and polarized mid-IR imaging. The manuscript has an excellent detail and analysis of the properties of the photodetectors and the device performance is of interest.

The authors state “Besides, we are aware of no report about mid infrared polarization imaging based on 2D materials at all” while there are no reports detailing imaging, there are reports of 2D material mid infrared polarization photodetectors (<https://doi.org/10.1021/acs.nanolett.6b01977>, <https://www.nature.com/articles/s41467-017-01978-3>) which should be cited within the manuscript.

Answer: Thanks for the reviewer’s valuable suggestion. We have added more citations (Ref. 24-25 in the revised manuscript) to make our statement more convincing.

The revised parts in the manuscript are listed below:

1. Page 3, line 23-25:

Besides, there are reports of 2D material mid infrared (MIR) polarization photodetectors^{24, 25}, but the works about high contrast polarization imaging based on 2D materials have not yet been realized,

2. Page 8, line 1:

(Supplementary Information, Table S2, Figure S27)^{24-26, 35-43}.

3. Page 20, line 28-29; Page 21, line 1-2

24. Guo, Q. et al. Black phosphorus mid-infrared photodetectors with high gain. *Nano Lett.* 16, 4648-4655 (2016).

25. Chen, X. et al. Widely tunable black phosphorus mid-infrared photodetector. *Nat. Commun.* 8, 1672 (2017).

Based on this I believe the novelty of the paper lies in the ability to A) form 2D Te nanosheets that are air stable and fabricate a photodetector from them; and B) use these nanosheets specifically for photodetector based imaging. In this context, specifically case B) I believe the manuscript demonstrates suitable potential impact on the field, detailed analysis and methodology, to be suitable for publication after the following revisions:

1. The authors state that the quasi-2D Te nanosheets were grown by a solution phase method as described in their reference 25. Looking at the thickness of the nanosheet (Figure S4) this is approximately 27 nm. In reference 25 the authors describe the fabrication route as capable of producing 2D nanosheets down to 3 nm thickness. Why have the authors chosen to use 27 nm thick Te in this study? Was any comparison done to thinner sheets? The novelty on the paper rests on the lack of MIR polarization imaging from 2D materials therefore the use of such a thick Te nanosheet needs to be addressed.

Answer: Thanks for the reviewer's valuable comments. Based on reference 25, the thickness of Te can be precisely controlled during the growth procedure. The Te flakes are provided by Prof. Wu (the corresponding author of reference 25, and the corresponding author in our work), and according to his experience, the provided Te flakes show highest crystal quality. So, we have not measured the photodetecting performance of the thinner flakes. Moreover, with thicker thickness, the bandgap of Te becomes narrower, so the detectable wavelength can be further extended to longer wavelength, which is significant for NIR and MIR photodetecting and imaging. In addition, larger photocurrent can be achieved by using thicker materials due to strong absorption, strong light-matter interactions in the confined 2D plane can also attribute to strong optoelectronic performances. But the sample thickness should not be too thick, otherwise the overlarge dark current will lead to poor photo-detectivity and other performance.

We have revised the manuscript carefully to address the reason for using thick Te samples, the revised parts in the manuscript are listed below:

1. Page 5, line 1-7:

The thickness of Te flake measured by atomic force microscope (AFM) is ~27.5 nm (Figure S1D), and its thicker thickness is beneficial to reach stronger absorption, leading to higher photocurrent, but the thickness should not be too thick, otherwise the overlarge dark current will lead to poor photo-detectivity and other performance. Besides, the thicker Te flake possesses narrower bandgap⁸, which is beneficial to extend the detectable bandwidth to longer infrared wavelength.

2. Page 8, line 1-3:

The thick Te flake (Figure S1D) also contributes to such high performances, as the transmitted light to the substrate is limited, leading to stronger absorption characters.

3. Supplementary information, page 17, line 13:

The thickness of the Te crystal is ~27.5 nm.

2. Comparing the material to other 2D MIR photodetectors such as Black Phosphorous, (which are sub 3nm thick), how does the device performance of the 27 nm Te compare? Can this performance difference be attributed to the extra thickness of material increasing the amount of photons absorbed?

Answer: Thanks for the reviewer's good comments. We have summarized photodetecting performances for many 2D materials based photodetectors. (Table S2, Supplementary information). In addition, we agree with the reviewer. The extra thickness of Te flake can contribute to the enhanced photon absorption, leading to the high performance. Moreover, the Te flake is very stable at ambient conditions, making it promising for MIR polarization applications.

3. Table S1-S3 - it is hard to compare mobility values with vastly different materials thicknesses - can the authors add a column showing 2D material thickness for the measured mobility as this would provide context to the measured values - particularly reference 17 and 21.

Answer: Thanks for the reviewer's significant comments. We have added a column showing 2D material thickness to make it clearer for mobility comparison. The

revised parts in the supplementary information are listed below:

1. Supplementary information, page 42, Figure S26:

2. Supplementary information, page 45:

Table S1 Field effect mobility of different 2D materials

Material	Thickness	Mobility(cm ² V ⁻¹ s ⁻¹)	References
Tellurium	27.5 nm	938	This work
GeP	4.3 nm	200	10
Tellurium	16 nm	700	11
MoS ₂	5 nm	28	12
ReS ₂	28 layers	79.1	13
WS ₂	3.5-5 nm	60	14
Black phosphorus	5 nm	286	15
Black phosphorus	10 nm	984	16
Bi ₂ O ₂ Se	5 nm	200	17
Tellurium	12.3 nm	419	21

4. Figure S14 - do the band structure calculations assume a quasi-2D material? Can 26 nm thick be treated as bulk for electronic purposes?

Answer: Thanks for the reviewer's valuable comments. The band structure is calculated for bulk Te with a quasi-2D structure, the helical chain is along k_y direction

(Fig. S5 in the revised Supplementary information). Yes, the thickness of 26 nm is treated as bulk for electronic purposes.

5. Can the authors comment on how many devices they characterized? If there are more than 1 are these all the same thickness? If there was only 1 - can the authors comment on how repeatable the data is?

Answer: Thanks for the reviewer's valuable comments. We have fabricated three different devices, and all the measurements are carried out in the three different devices. The results as shown in Figure 1B, 1D, 1E, Figure 2 and Figure 3, are measured in the device shown in Figure 1A. For the performances from Figure 2 and Figure 3, we have chosen electrode 2 and 4, so the channel length is 6 μm and the channel width is 2.5 μm . Figure 1C is measured in the device shown in Fig. S1E, with two pairs of electrodes located along two perpendicular directions, the channel lengths are 10 μm for both directions. Figure 4 is measured by another device. All these devices show similar thickness in the range of 26 nm to 27.5 nm, so the experimental results are comparable and repeatable.

Also, minor text/formatting changes are recommended.

- Figure 3. the degree symbol in D, E, and F is superscript formatted - it shouldn't be
- Figure 4, The scale bar in A (that applies to C D and E) is difficult to read, this should be enlarged or added to C, D, and E.
- Figure S4 - needs a scale bar or axis; can the authors do roughness analysis/provide a profile along the long axis of the flake

Answer: Thanks for the reviewer's valuable suggestions. We have checked the manuscript thoroughly and corrected these changes. Scale bars are added for all mapping figures in the manuscript and supplementary information. For Figure S4, we have checked the AFM results and added a profile along the long axis of the flake, roughness analysis is also provided (revised AFM results are shown in Figure S1D in the revised Supplementary information). The revised parts in the manuscript are listed below:

1. Page 28, Figure 3:

2. Page 29, Figure 4:

3. Page 29, line 10:

Scale bar in (C)-(E), 1 mm.

4. Supplementary information, page 17, Figure S1:

5. Supplementary information, page 17, line 10-13:

(D) AFM results of synthesized tellurium nanoflake. (i) The AFM image of the tellurium crystal, scale bar, $10 \mu\text{m}$, (ii) shows the height distribution of tellurium along the red and blue dashed line in (i), respectively. The thickness of the Te crystal is $\sim 27.5 \text{ nm}$.

Reviewers' comments:

Reviewer #1 (Remarks to the Author):

The authors have adequately responded to some of the previous comments, however, there still exists several claims in the paper which are either incorrect/unnecessary or need to be further supported.

The authors comment on the polarization extinction ratio of their Te detectors being high. Compared to existing devices, including those made from low dimensional materials, a value of 8 is not particularly high – some papers reporting much higher – I recommend that the authors provide a more representative comparison. The authors claim commercial polarizers also have a ratio of around ~ 10 . From my understanding this ratio is at least 10x this in the infrared region. I suggest the authors correct their statement on commercial polarizers.

The authors claim the inherent fabrication advantages of Te based detectors for polarization detectors. I still cannot see how this system provides any fabrication advantages to using a standard IR detector with a polarizer both of which are well developed systems and provide significantly better contrast due a higher polarization resolution. In addition, a solution synthesis process is used for growing Te flakes. It seems that processing such flakes into a polarization-aligned FPA would represent a significantly more difficult fabrication process (even allowing for dramatic improvements in this area) than the fabrication of a wire-grid polarizer which is an existing mass-produced item. I would suggest the authors tone-down their claim as to the advantages of the system in its current form.

The above comment relates to claims of high contrast and resolution. Based on the polarization ratio of 8 and the 1mm scalebar in figure 4 I don't think the authors can claim either of these characteristics have been demonstrated. Further, it is not clear (even with dramatic improvements in Te growth technology) how Te based detectors could provide a better pixel density than conventional IR detectors. I suggest removing statements that high-resolution or high-contrast has been achieved.

Minor comments:

Typically, external quantum efficiency is not a metric which is used in systems which exhibit gain. The authors may want to remove the EQE plots. Especially considering the responsivity has been reported anyway – hence the EQE adds no additional information and may confuse some readers.

Figure 1E, was this 60% absorption value collected on a 27nm flake on a Si/SiO₂ wafer? A quick simulation, using Te refractive index values from the literature shows, that the absorption is at most 35% (this is even assuming a perfect $\frac{1}{4}$ wave SiO₂ thickness). If this plot was obtained on a thicker flake, this information needs to be clearly included.

I notice significant changes have been made to authorship without significant changes to the data/conclusions in the main text.

Reviewer #2 (Remarks to the Author):

I understand that the authors have made a lot efforts to improve the paper. After carefully reviewing the authors' reply, I am still not convinced by the main arguments. My main concerns are about the physical mechanism of photo-gain in the paper and the detectivity calculation. I believe these two parts are the main scientific contents in the paper and sadly they are confusing and misleading from my point of view. This is important, for example, the high responsivity of 300 A/W in the paper is against quantum physics unless there is photogain. However, the authors

failed to explain it after they ruled out my explanation of photogating. Another concern is about the novelty and the main claim of the paper. The authors believed it is 'realizing high contrast polarization imaging applications'. However, I am very pessimistic about the future application in imaging due to the impossible growth of large-area single crystals. I will address these concerns point by point below.

#1. Novelty

The authors replied that the novelty lies on the 'realizing high contrast polarization applications based on our device...for the first time'. The authors give example of calculating DoLP data by their detectors. However, the truth is that the imaging is realized by scanning and rotating a single device with single crystal orientation. Usually for other 2D materials, this demonstration is enough as their corresponding multi-pixel camera does not rely on crystal orientation. Here, in the case of Te detector, the realizing imaging system for calculating DoLP needs all the pixels have same orientation, which is very challenging. Although the authors supplied several possible methods of growing single crystal of Te, the challenge still stands. Given the other method of metal grids with different orientations, I am very pessimistic about the concept presented in the paper.

#2, perspective of tellurium as photodetector

The authors mentioned 'III-V compounds and many 3D materials...suffer from large noise at room temperature, thus requiring large cooling facilities to enhance detectivity'. However, I strongly doubt this claim as the method to grow single-crystalline III-V compounds are very mature. GaAs can have very high mobility. They usually made to detect light with much larger wavelength than light in the paper, and thus they are specially made to target these wavelengths with cooling condition. I believe if made by targeting at short-wavelength infrared light, III-V compounds could behave much better than nearly all 2D materials, even at room temperature. This is beyond the discussion of this paper.

The authors also mentioned other ways to make large area Te, however, it is very challenging first of all. Also if we consider that the application in this paper needs single-crystalline structure, it is nearly impossible.

#2.1 Defects

The authors claim 'few defects nature based on HAADF-STEM'. However, from my point of view, it is very hard to conclude this by such measure. Usually, STEM can only observe a tiny area of the sample and does not give statistic of defects concentration. In principle, if there is no defects, Te should show absolute insulating behavior at room temperature instead of the behavior as shown in Fig. 1 and 2.

#2.2 Gain and photogating.

The authors did control experiment by measuring transfer curves with and without illumination in Fig. S9. However, they failed to show the original data of transfer curves. Instead, they show photocurrent vs. gates. This confuses me. Let's assume all the transfer data exactly show the minimum points at the same gate values. This cannot rule out the photogating behavior as the transfer curve is a collective behavior of all the electrons flowing inside the device. And these dark electrons may overwhelm the trapped carrier density, which plays a role in photogating. This also relates to the trapping time scale of the photocarrier and the DC measurement dwell time. As the Authors may measure it by Keithley, which typically has long DC dwell time and may further cloud the observation of neutrality point's movement.

The authors rule out photogating effect by showing the response time of 60 micro-second. They also mentioned that it is ultrafast. Firstly this speed is not ultrafast if compared with response of graphene and III-V compounds (shorter than 1 ns). So this speed exactly demonstrates the photogating if this speed is material-limit. Secondly, the authors claimed 354 A/W responsivity. This is against the quantum physics if there is no gain. If there is no photogain, as the authors claimed, there should be another explanation for the unusual quantum efficiency. The authors' explanation of 'large carrier life time and short transit time' simply does not stand from my point of view. If we assume all the photons excite photoelectrons, and then all the electrons and holes move out from the channel, the efficiency should be 100%, no matter how fast the charges move. Therefore, the authors need to find explanation for the unusual quantum efficiency if they disagree

with me.

#3.2.3 Detectivity

The authors have used the exact equation to calculate the detectivities and admitted that it is 1 to 2 orders of magnitude smaller than the values by their method. However, they insist to use their values in the main text. They also cited several papers and claim it is 'implemented in most works about 2D materials-based devices'. My opinion is that there should be only one true value in every case, which is exactly the beauty of science. I am not sure if 'most' work from 2D materials-based devices have used the simplified equation, but I do think that the noise equivalent power is exactly the core of detectors. By saying noise, it refers all noise. I hope people in 2D world were right, but I am afraid, we have to respect the truth and completely stop using the simplified equation in current paper.

Reviewer #3 (Remarks to the Author):

The authors have taken time to thoroughly address the reviewers comments, significantly strengthening the manuscript. The expansion of the manuscript enables the reader to understand the context and potentially repeat the work if desired.

One key point of novelty within the paper remains the high mobility reported, $938 \text{ cm}^2 \text{ V}^{-1} \text{ s}^{-1}$, which is almost as high as black phosphorus, $984 \text{ cm}^2 \text{ V}^{-1} \text{ s}^{-1}$. While this improvement is dramatic compared to the previously best reported value of Te, $700 \text{ cm}^2 \text{ V}^{-1} \text{ s}^{-1}$, there is an apparent trend of increasing mobility with increasing flake thickness (i.e moving from the 2D domain into the bulk domain), which the authors address in their response to reviewers letter (pg 60, L1,2). This trend with increased mobility also holds true for black phosphorous. The expansion of table S1 allows readers to draw their own conclusions on these results in the context of materials performance - but logically the question remains, if thicker Te flakes are used would the mobility continue to increase giving improved performance, and in this case is 2D an important distinction.

The introduction, pg 4 L94-96, still states that 2D-Te has a mobility "which is far larger than other 2D materials", which the authors have acknowledged is not the case with black phosphorous having a larger mobility.

These points are relatively minor, and I believe given the expansive extra detail added to the manuscript that it can be considered suitable for publication in Nature Communications.

Response Letter

Reviewer #1

The authors have adequately responded to some of the previous comments, however, there still exists several claims in the paper which are either incorrect/unnecessary or need to be further supported.

Response: We sincerely thank the reviewer for the great efforts providing valuable suggestions.

1. The authors comment on the polarization extinction ratio of their Te detectors being high. Compared to existing devices, including those made from low dimensional materials, a value of 8 is not particularly high – some papers reporting much higher – I recommend that the authors provide a more representative comparison. The authors claim commercial polarizers also have a ratio of around ~10. From my understanding this ratio is at least 10x in the infrared region. I suggest the authors correct their statement on commercial polarizers.

Answer: We sincerely thank the reviewer for the valuable suggestions. We agree with the reviewer that the anisotropic ratio of Te is not particularly high, compared to the reported results for other 2-D materials such as black phosphorus (*ACS Nano* 11, 11724-11731 (2017); *Adv. Mater. Interfaces* 5, 1800960 (2018))^{1,2}. In our work, the anisotropic ratio of Te is ~8 at 2.3 μm illumination and higher than 10 for the longer incident wavelengths, which is sufficient for polarization imaging measurement based on the IR polarization imaging experimental results. We have corrected the claims about the description of the anisotropic ratio of Te in the revised manuscript. (Page 10, line 9-15; Page 10, line 27-29)

Moreover, the Te-based photodetector shows good stability under the atmospheric environment for a long working time, which is much better than

un-encapsulated black phosphorous and black phosphorous-arsenic. To address the reviewer's comment, we have provided a more representative benchmark comparison with other 2D materials in the revised supplementary information. Please see below the revised text and the comparison table.

We also thank the reviewer for pointing out the inaccurate statements about the commercial polarizers. Considering that 2D materials have potential to show another routine for future photodetecting applications, which is different from commercial polarizers, we have just focused on the polarization applications of the 2D Te. In the revised manuscript, we have also corrected and deleted these inaccurate statements.

References:

1. Amani, M. *et al.* Mid-wave infrared photoconductors based on black phosphorus-arsenic alloys. *ACS Nano* 11, 11724-11731 (2017).
2. Li, X. *et al.* Polarization-dependent photocurrent of black phosphorus/rhenium disulfide heterojunctions. *Adv. Mater. Interfaces* 5, 1800960 (2018).

The revised parts in the revised manuscript are listed below:

1. Page 10, line 10-15:

Importantly, a higher photoresponsivity anisotropic ratio of ~ 7.58 is realized under $2.3 \mu\text{m}$ illumination for Te-based photodetector, which is sufficient to achieve polarized imaging applications without the assistance of extrinsic modulation. The anisotropic ratio is also compared with the widely researched 2D materials such as black phosphorous, WTe_2 , ReS_2 , ReSe_2 , etc. (Table S3, Figure S27)^{9, 17, 21, 23, 26, 38, 43, 46-50}, which is showing satisfactory polarization properties of Te.

2. Page 10, line 28-30:

Based on Figure S21, the polarization extinction ratio of Te-based device under $2.3 \mu\text{m}$ illumination is reaching ~ 8 , which is acceptable in the field of 2D materials.

3. Page 22, line 17-20:

49. Amani, M. *et al.* Mid-wave infrared photoconductors based on black phosphorus-arsenic alloys. *ACS Nano* **11**, 11724-11731 (2017).

50. Li, X. *et al.* Polarization-dependent photocurrent of black phosphorus/rhenium disulfide heterojunctions. *Adv. Mater. Interfaces* **5**, 1800960 (2018).

4. Supplementary information, page 47-48, Table S3:

Black phosphorus-arsenic	Photoresponse	>100 @ 3.5 μm	38
	Absorption	~10 @ 3.5 μm	
Black phosphorus/ReS ₂ heterostructure	Photoresponse	31	39
Black phosphorus	Photoresponse	3.5 @ 1.2 μm	40
	Reflectance	~2 (energies above the bandgap)	
	Transmittance	~2 @ 2.3 eV	

5. Supplementary information, page 51, line 23-28:

38. Amani, M. *et al.* Mid-wave infrared photoconductors based on black phosphorus-arsenic alloys. *ACS Nano* **11**, 11724-11731 (2017).

39. Li, X. *et al.* Polarization-dependent photocurrent of black phosphorus/rhenium disulfide heterojunctions. *Adv. Mater. Interfaces* **5**, 1800960 (2018).

40. Yuan, H. *et al.* Polarization-sensitive broadband photodetector using a black phosphorus vertical p-n junction. *Nat. Nanotechnol.* **10**, 707-713 (2015).

2. The authors claim the inherent fabrication advantages of Te based detectors for polarization detectors. I still cannot see how this system provides any fabrication advantages to using a standard IR detector with a polarizer both of which are well developed systems and provide significantly better contrast due a higher polarization resolution. In addition, a solution synthesis process is used for growing Te flakes. It seems that processing such flakes into a polarization-aligned FPA would represent a significantly more difficult fabrication process (even allowing for dramatic improvements in this area) than the fabrication of a wire-grid polarizer which is an existing mass-produced item. I would suggest the authors tone-down their claim as to the advantages of the system in its current form.

Answer: We are grateful for the review's valuable comments. We would like to emphasize that the novelty of our work lies in that it is the first demonstration of 2D materials-based photodetectors to realize the stable mid-IR polarization imaging with enhanced contrast under strong scattering environment at room temperature, which is just the first step toward future integration of polarization-aligned FPA. Moreover, as we addressed in the previous response, making the full of the advantages of 2D Te provides another routine for the development of mid-IR polarization photodetecting applications, which is still in the infancy with some challenges including material growth, design and fabrication. That being said, we agree with the reviewer that the inherent fabrication advantages of the 2D Te-based photodetector were overemphasized over the traditional polarization infrared photodetectors (using a standard IR detector with a polarizer). Also, it requires more efforts to implement cost-effective the 2D Te based IR detectors through an economically viable manufacturing process that may be drawing to a close to the state-of-the-art. To address the comments, we have made the following revisions to tone-down our claim as to the advantages of the system in its current form.

The revised parts in the manuscript are listed below:

1. Page 2, line 2-10:

Polarized mid-infrared (MIR) imaging systems have developed quickly in recent years, for next-generation imaging technology, it generally requires miniaturization, high level of integration and flexibility, high pixel density, good workability at room temperature and severe environments, and so on. The emerging two-dimensional (2D) materials have provided another routine to make the full of their potential in polarized imaging systems, due to the ease of integrating 2D materials on complex structures, their native in-plane anisotropy crystal structure for high polarization photosensitivity, and their strong quantum confinement effects for excellent photodetection performances at room temperature.

2. Page 3, line 5-7:

At present, the trend of modernizing polarized infrared imaging applications increasingly requires convenience, miniaturization, good workability at room temperature and severe environments, and larger pixel density for future devices⁵⁻⁷.

3. Page 10, line 10-15:

Importantly, a higher photoresponsivity anisotropic ratio of ~ 7.58 is realized under $2.3 \mu\text{m}$ illumination for Te-based photodetector, which is sufficient to achieve polarized imaging applications without the assistance of extrinsic modulation. The anisotropic ratio is also compared with the widely researched 2D materials such as black phosphorous, WTe_2 , ReS_2 , ReSe_2 , etc. (Table S3, Figure S27)^{9, 17, 21, 23, 26, 38, 43, 46-50}, which is showing satisfactory polarization properties of Te.

4. Page 11, line 27-29:

It suggests that the Te-based device needs fewer pixels to realize the same resolution for *DoLP*, making it easier to further promote the imaging resolution.

5. Page 12, line 13-30; page 13, line 1:

At present, most *DoFP* structures require a layer of polarization filter and a layer of photo-sensing array⁵, making it requires four single-point devices with different

polarization filtering directions to detect polarized signals along four different directions for the calculation of one pixel in *DoLP*. Theoretically, wire grid polarizers with matured fabrication technology can also reach a high polarization extinction ratio as only light polarized parallel to the grid can pass through the polarizer. As for 2D materials including Te, their intrinsic polarized photosensitivity can provide another route to realize polarization functionalities. For example, the intrinsic polarized photosensitivity of Te promotes polarization imaging without polarization filters. The photocurrent in one single point device along different polarization orientations can be collected by rotating the Te device, for the calculation of *DoLP* to enhance the imaging contrast. However, it's worth noting that the Te-based devices and many other 2D materials-based polarization devices are still at the infancy with some existing challenges at present. Currently, the fabrication of large-scale device arrays is a great challenge in the field of 2D materials. With the development of matured methods for producing large films, the Te photodetector array could be one promising option to realize the practical polarization imaging. This work serves as the first step towards the development of polarization imaging arrays based on 2D materials, which is of great significance.

6. Page 13, line 8-10:

2D Te is also applicable for flexible devices with polarization sensitivity, promising the high level of integration and flexibility.

3. The above comment relates to claims of high contrast and resolution. Based on the polarization ratio of 8 and the 1mm scalebar in figure 4 I don't think the authors can claim either of these characteristics have been demonstrated. Further, it is not clear (even with dramatic improvements in Te growth technology) how Te based detectors could provide a better pixel density than conventional IR detectors. I suggest removing statements that high-resolution or high-contrast has been achieved.

Response: We thank the reviewer very much for the valuable comments, and agree with the reviewer. Considering that our work is the first step toward the application of polarization imaging in field of 2D materials-based photodetectors, more efforts are needed to reach the goal of high contrast and resolution. To make our claims more concrete, we have corrected the “high-contrast” to be “contrast-enhanced” and removed some overemphasized statements in the revised manuscript including “high resolution”. By virtue of the intrinsic anisotropic photoresponse of Te, the imaging contrast of the Te-based photodetector is enhanced under scattering environments. Thanks again for the reviewer pointing out the mistake statements.

The revised parts in the manuscript are listed below:

1. Page 1: Title

Stable **contrast-enhanced** mid-infrared polarization imaging based on quasi-2D tellurium at room temperature

2. Page 2, line 10-11:

However, polarized infrared imaging **with enhanced contrast** based on 2D materials **has yet to be realized** to confirm such feasibility.

3. Page 2, line 11-14:

Here we report the systematic investigation and experimental demonstration of polarized infrared imaging for a designed target obscured by scattering media with enhanced contrast using anisotropic quasi-2D tellurium (Te) photodetector.

4. Page 2, line 19-21:

Significantly, the Te photodetector has a large anisotropic ratio of ~ 8 for $2.3 \mu\text{m}$ illumination, ensuring **contrast-enhanced** polarized imaging in a scattering environment.

5. Page 3, line 22-23:

but the works about polarization imaging with enhanced contrast based on 2D materials have not yet been realized

6. Page 4, line 13-14:

which is sufficient for potential contrast-enhanced polarized imaging.

7. Page 4, line 19-20:

the quasi-2D tellurium-based photodetector reported here can realize contrast-enhanced imaging for the target.

8. Page 13, line 19-20:

and used the photodetector to successfully realize contrast-enhanced polarized infrared imaging.

9. Page 13, line 24-26:

Based on the anisotropic photoresponses, contrast-enhanced imaging is preserved for quasi-2D Te-based photodetector under scattering environments.

10. Page 29, Figure 4:

4. Minor comments:

Typically, external quantum efficiency is not a metric which is used in systems which exhibit gain. The authors may want to remove the EQE plots. Especially considering

the responsivity has been reported anyway – hence the EQE adds no additional information and may confuse some readers.

Response: Thanks for the reviewer’s suggestions. We agree with the reviewer that EQE is not a typical metric for photodetector with large photoconductive gain. To address this comment, we have removed the EQE plots in the revised manuscript. The revised parts in the manuscript are listed below:

1. Page 7, line 16-30; page 8, line 1-15:

To characterize the device performances under different illumination powers and wavelengths, we have summarized the major parameters of photoresponsivity (R) and measured detectivity (D_M^*) of the photodetector at room temperature by using the pseudo-color mapping, as shown in Figure 2E-2F (scatter diagrams are included in Figure S7K-N). The drain bias is fixed at 1.0 V, and the gate bias is fixed at 0 V, the frequency of the noise current is chosen to be 1 kHz, the corresponding noise current density is $4.55 \times 10^{-11} \text{ AHz}^{1/2}$ at room temperature, and the calculation details are described in the Supplementary Information notes 5. The lower illumination power always leads to higher R and D_M^* , and the evolution of R and D_M^* under fixed laser power is in accordance with absorption results. Due to effective absorption of Te at $3.0 \mu\text{m}$, high R of $\sim 3.54 \times 10^2 \text{ A/W}$ and D_M^* of $\sim 3.01 \times 10^9 \text{ Jones}$ are realized under the illumination at the wavelength of $3.0 \mu\text{m}$, which is competitive among various 2D materials-based photodetectors (Supplementary Information, Table S2, Figure S26)^{24-26, 35-43}. The thick Te flake (Figure S1D) also contributes to such high performance due to its stronger absorption character, as the transmitted light to the substrate is limited. The high performance is also related to a large photoconductive gain, which is assisted by the fast carrier transition time due to the high mobility of Te and the short channel length of $\sim 6.0 \mu\text{m}$. In addition, the high crystal quality with few defects can contribute to the photoconductive gain through shallow states of photo-induced carriers at the band tail^{44, 45}, which is a fast process to provide a large gain. The existence of photogain is revealed from the EQE values in Figure S8A-C.

Besides, we have also measured the transfer curves under $1.06\ \mu\text{m}$ illumination and $1.0\ \text{V}$ drain bias at room temperature, as shown in Figure 2G panel (i), the net photocurrent is summarized in the panel (ii), the maximum net photocurrent is realized at the neutral point with $\sim 5.9\ \text{V}$ gate bias, also suggesting that the photoresponses of the device are not dominated by the trap-induced photogating effect. To further show the origin of photogain, we have measured the photocurrent distribution in the Te channel (Figure 2G panel (iii)), the photocurrent is generated at the channel/electrode interface instead of the whole channel, excluding the photogating as a dominant factor for the high photogain.

2. Page 8, line 19-24:

The carrier lifetime is extracted from transient photoluminescence decay curves of Te, as shown in Figure S10. Based on carrier lifetime and transit time, a photogain (G) at the level of $\sim 2.3 \times 10^3$ can be calculated, indicating the high performance in the Te-based device. Besides, to make it more convenient to compare the device performances with previous works, the calculated detectivity (D_C^*) is also summarized in Figure S11.

3. Page 9, line 1-7:

The dependences of R and D_M^* on the wavelength and laser power of the incident light, are summarized in Figure 2E, F, I, and J, respectively (scatter diagrams are included in Figure S12K-N, external quantum efficiency (EQE) is shown in Figure S8D-F). Here the results show similar trends to those measured at $300\ \text{K}$ (Figure 2E-2F). At $77\ \text{K}$, a higher R can reach up to $\sim 8.36 \times 10^2\ \text{A/W}$ and D_M^* is at a higher level of $\sim 9.04 \times 10^9$ Jones, under the illumination wavelength of $3.0\ \mu\text{m}$.

4. Page 26, Figure 2:

5. Page 26, line 6-8:

(E)-(F) Pseudo-color mapping figures for R and D_M^* , as a function of laser power and wavelength at room temperature (300 K).

6. Page 26, line 10; page 27, line 1-10:

(G) Photoresponses under different gate biases. The drain bias is 1.0 V, and the incident wavelength is 3.0 μm . Panel (i) shows the transfer curves under different incident powers and the dark state, and panel (ii) shows the net photocurrents under different illumination powers. The maximum photocurrent is achieved at the neutral point of ~ 5.9 V gate bias for all the incident powers. (H) The photocurrent distribution in the Te channel under 0.637 μm and 0.83 μm illumination at 1.0 V drain bias and 0 V gate bias, with the photocurrent generated at the channel/electrode interface. The inset shows the image of the Te device with laser spot size of ~ 3 μm ,

scale bar, 10 μm . (I)-(J) Pseudo-color mapping figures for R and D_M^* , as a function of laser power and wavelength at 77 K.

7. Supplementary information, page 23, Figure S7:

Figure S7. Unpolarized broadband optoelectronic responses for tellurium nanoflake device at room temperature (300 K). (A)-(J) Photocurrent for 0.52 μm , 0.637 μm , 0.785 μm , 0.83 μm , 0.94 μm , 0.106 μm , 0.131 μm , 0.155 μm , 2.3 μm , and 3.0 μm illumination, respectively, as a function of laser power and drain bias, the gate bias is 0 V. (K)-(L) R and D_M^* as a function of laser power for different wavelengths, respectively, the drain bias is fixed at 1.0 V, the gate bias is 0 V. (M)-(N) R and D_M^* as a function of wavelength under different illumination laser powers at room temperature.

8. Supplementary information, page 24, Figure S8:

Figure S8. (A)-(C) *EQE* results at room temperature. (A) Pseudo-color mapping image of *EQE*. (B) *EQE* as a function of illumination laser power for different wavelengths at room temperature. (C) *EQE* as a function of wavelength under different laser powers at room temperature. (D)-(F) *EQE* results at low temperature. (D) Pseudo-color mapping image of *EQE*. (E) *EQE* as a function of illumination laser power for different wavelengths at 77 K temperature. (F) *EQE* as a function of wavelength under different laser powers at 77 K temperature. The performance under 77 K is higher.

9. Supplementary information, page 27, Figure S11:

Figure S11. The calculated detectivity (D_C^*) results. (A)-(C) D_C^* at room temperature. (A) Pseudo-color mapping results of D_C^* . (B) D_C^* as a function of laser power for different wavelengths at room temperature. (C) D_C^* as a function of wavelength under different laser powers at room temperature. (D)-(F) D_C^* at 77 K

temperature. (A) Pseudo-color mapping results of D_C^* . (B) D_C^* as a function of laser power for different wavelengths at 77 K temperature. (C) D_C^* as a function of wavelength under different laser powers at 77 K temperature. D_C^* is about 1-2 magnitude higher than D_M^* due to the lower dark current than the actual noise current.

10. Supplementary information, page 28, Figure S12:

Figure S12. Unpolarized broadband optoelectronic responses for tellurium nanoflake device at 77 K. (A)-(J) Photocurrent under the illumination of 0.52 μm , 0.637 μm , 0.785 μm , 0.83 μm , 0.94 μm , 1.06 μm , 1.31 μm , 1.55 μm , 2.3 μm , and 3.0 μm , respectively. The gate bias is fixed at 0 V. (K)-(L) R and D_M^* as a function of laser power for different wavelengths, respectively, the drain bias is fixed at 1.0 V, the gate bias is 0 V. (M)-(N) R and D_M^* as function of wavelength under different illumination laser powers at 77 K temperature.

5. Figure 1E, was this 60% absorption value collected on a 27nm flake on a Si/SiO₂ wafer? A quick simulation, using Te refractive index values from the literature shows, that the absorption is at most 35% (this is even assuming a perfect $\frac{1}{4}$ wave SiO₂

thickness). If this plot was obtained on a thicker flake, this information needs to be clearly included.

Response: We sincerely appreciate the reviewer for the valuable comments to point out our description of the absorption measurements. The absorption measurement was performed for a 27-nm-thick Te flake on a Si/SiO₂ wafer, based on method 4 in the manuscript, and the measured value should be twice the absorption of Te. We have revised the manuscript accordingly to address this comment and clarify the description for the absorption measurements:

1. Page 6, line 18-19:

Based on method 4, the measured value should be twice the absorption of Te.

2. Page 15, line 10-11:

It is also noted that the light is absorbed by Te twice along the light path, so the measured absorption value should be twice the absorption rate of 2D Te.

6. I notice significant changes have been made to authorship without significant changes to the data/conclusions in the main text.

Response: Thanks for the reviewer's comments. In the 1st revised manuscript, we have carried out a significant amount of additional experiments to fully address the reviewers' comments and improve the quality of our work. These additional experiments include:

- absorption at longer wavelengths (Fig. 1E)
- noise spectral density (Fig. S23)
- transient photoluminescence (Fig. S10)
- re-measuring the AFM height distribution in two perpendicular directions (Fig.

S1D)

- measuring the photoresponses under different gate bias (Figure 2G)

These experiments were carried out with the help of newly added authors (Fang Zhong, Fang Wang). The description for each author's contribution has also been updated in the main text.

Reviewer #2

I understand that the authors have made a lot efforts to improve the paper. After carefully reviewing the authors' reply, I am still not convinced by the main arguments. My main concerns are about the physical mechanism of photo-gain in the paper and the detectivity calculation. I believe these two parts are the main scientific contents in the paper and sadly they are confusing and misleading from my point of view. This is important, for example, the high responsivity of 300 A/W in the paper is against quantum physics unless there is photogain. However, the authors failed to explain it after they ruled out my explanation of photogating. Another concern is about the novelty and the main claim of the paper. The authors believed it is 'realizing high contrast polarization imaging applications'. However, I am very pessimistic about the future application in imaging due to the impossible growth of large-area single crystals. I will address these concerns point by point below.

Response: We are very grateful to the reviewer for the inspiring and constructive comments. We have fully and rigorously responded to the whole comments from the reviewer.

For the discussion about the physical mechanism of our measured photo-gain in the 1st response letter, we should expect to see a significant shift of the neutral point in the device's transfer curve between cases with and without illuminations, if the deep trap-induced photo-gating behavior dominates the photocurrent¹. To address this

comment, we have measured the dependence of the photocurrent on the gate bias, and the results (Fig. 2G) show that the neutral point in the transfer curve is not shifted when we switch on and turn off the illumination, suggesting that the observed photo-gain of our device is not dominated by the deep trap-induced photo-gating mechanism^{2, 3}. Besides, we have also measured the photocurrent distribution in the channel of 2D Te-based device, and the obvious photoresponse should be distributed through the whole channel for the device under the domination of deep trap-induced photo-gating. But the photoresponse is localized near the channel-electrode interface (Fig. 2H) based on the measurement results, with no photoresponse at the center of the channel, indicating that the photo-gain of our device is not dominated by the deep trap-induced photo-gating⁴.

Actually, we admit that the high photoresponsivity is assisted by photogain in the device. It has been reported that the photogain of 2D materials based photodetectors can also originate from photoconductive gain due to the band-tail states in 2D semiconductors, which is a fast process to gives rise to a large gain for the high responsivity^{5, 6}. In addition, Amani, M. *et al.*² (*ACS Nano* 12, 7253-7263 (2018)) have measured the transfer curves of Te-based device with and without illumination, and reported the similar finding to our results.

For the detectivity calculation, we appreciate the reviewer's insightful comment. Indeed, many reports about 2D materials-based photodetectors have calculated the detectivity based on the dark current⁷⁻¹³, which will lead to an overestimated detectivity. To address this comment, we have measured the noise spectral density at both room temperature and 77 K to calculate the detectivity of the device more accurately.

We sincerely thank again the reviewer for his/her great efforts providing insightful and constructive comments to help improve the quality of our manuscript. We have done our best to address the reviewer's every comment in the following point-by-point response, including clearer discussions about the photogain mechanism

and detectivity calculations. As for the novelty of our work, we have tried our best to explain the background, challenges, potentials in the field of 2D materials-based photodetectors, and our intention in this work, to show the novelty. We hope that we can respond to the reviewers' concern with more detail, which are listed for the following comments.

References:

1. Island, J. O. et al. Gate controlled photocurrent generation mechanisms in high-gain In₂Se₃ phototransistors. *Nano Lett.* **15**, 7853-7858 (2015).
2. Amani, M. et al. Solution-synthesized high-mobility tellurium nanoflakes for short-wave infrared photodetectors. *ACS Nano* **12**, 7253-7263 (2018).
3. Guo, Q. et al. Black phosphorus mid-infrared photodetectors with high gain. *Nano Lett.* **16**, 7253-7263 (2016).
4. Long, M. et al. Room temperature high-detectivity mid-infrared photodetectors based on black arsenic phosphorus. *Sci. Adv.* **3**, e1700589 (2017).
5. Furchi, M. M., Polyushkin, D. K., Pospischil, A., Mueller, T. Mechanisms of photoconductivity in atomically thin MoS₂. *Nano Lett.* **14**, 6165-6170 (2014).
6. Fu, Q. et al. Ultrasensitive 2D Bi₂O₂Se phototransistors on silicon substrate. *Adv. Mater.* **31**, 1804945 (2018).
7. Gong, F. et al. High-sensitivity floating-gate phototransistors based on WS₂ and MoS₂. *Adv. Funct. Mater.* **26**, 6084-6090 (2016).
8. Mao, J. et al. Ultrafast, broadband photodetector based on MoSe₂/Silicon heterojunction with vertically standing layered structure using graphene as transparent electrode. *Adv. Sci.* **3**, 1600018 (2016).
9. Wang, P. et al. Arrayed van der Waals broadband detectors for dual-band detection.

Adv. Mater. **29**, 1604439 (2017).

10. Chao Xie and Feng Yan. Flexible photodetectors based on novel functional materials. *Small* **13**, 1701822 (2017).

11. Wan, X. *et al.* A self-powered high-performance graphene/silicon ultraviolet photodetector with ultra-shallow junction: breaking the limit of silicon? *npj 2D materials and applications* **1**, 4 (2017).

12. Shin, G. H. *et al.* S0-MoS₂ vertical heterojunction for a photodetector with high responsivity and low noise equivalent power. *ACS Appl. Mater. Interfaces* **11**, 7626-7634 (2019).

13. Lu, J. *et al.* 2D In₂S₃ nanoflake coupled with graphene toward high-sensitivity and fast-response bulk-silicon Schottky photodetector. *Small* **15**, 1904912 (2019).

#1, Novelty

The authors replied that the novelty lies on the 'realizing high contrast polarization applications based on our device...for the first time'. The authors give example of calculating DoLP data by their detectors. However, the truth is that the imaging is realized by scanning and rotating a single device with single crystal orientation. Usually for other 2D materials, this demonstration is enough as their corresponding multi-pixel camera does not rely on crystal orientation. Here, in the case of Te detector, the realizing imaging system for calculating DoLP needs all the pixels have same orientation, which is very challenging. Although the authors supplied several possible methods of growing single crystal of Te, the challenge still stands. Given the other method of metal grids with different orientations, I am very pessimistic about the concept presented in the paper.

Response: We sincerely appreciate the reviewer's comments regarding the novelty of our work. Indeed, the state-of-the-art polarization imaging with the

integration of metal grids based on commercial photodetectors shows advantages in many ways over that of based on the emerging 2D materials-based photodetectors, including matured growth methods. For 2D materials, they hold significant potentials in the future to realize novel functionalities and novel device structures, by integrating with photonic crystals, waveguides, meta-materials, flexible substrates, other complex structures, and making the full of the unique characters including polarization, spin-related phenomena, etc., for photodetecting.

Here, we have made the full of intrinsic polarized absorption of the 2D Te to verify the feasibility for polarized imaging from another routine. Our work has reported for the first time for 2D materials-based photodetectors to realize the stable mid-infrared polarization imaging with enhanced contrast under strong scattering environment at room temperature. We believe that the results reported here will be impactful and of interests to a broad audiences for materials and device research in the IR photodetector community, which warrants more systematic studies through future efforts for exploring the intrinsic mid-IR properties and potential mid-IR applications of 2D Te or other 2D materials, as well as the economically viable processes to fabricate and integrate them into high-performance devices.

However, although the growth is not the scope of our work, there's no denying that scalable growth is the main shortcoming toward the realization of imaging arrays, leading to that the researches on 2D materials-based photodetector arrays are still at the infancy. The experiments in our work are carried out based on a single pixel device just due to this limited growth method at present, and many follow-up works are still needed to be done in the future. It can expect the potential developments of large-scale 2D materials growths and corresponding device manufacturing processes which are presently promoted by many scientists in the field of 2D materials, to realize the practical mid-infrared polarization imaging device with the same crystal orientation for *DoLP* calculations.

We thank again for the reviewer, and we have corrected the overemphasized

advantages of 2D materials-based photodetectors, to tone-down the claim. The revised parts in the manuscript are listed below:

1. Page 2, line 2-5:

Polarized mid-infrared (MIR) imaging systems have developed quickly in recent years, for next-generation imaging technology, it generally requires miniaturization, high level of integration and flexibility, high pixel density, good workability at room temperature and severe environments, and so on.

2. Page 3, line 5-7:

At present, the trend of modernizing polarized infrared imaging applications increasingly requires convenience, miniaturization, good workability at room temperature and severe environments, and larger pixel density for future devices⁵⁻⁷.

3. Page 12, line 13-30; page 13, line 1:

At present, most *DoFP* structures require a layer of polarization filter and a layer of photo-sensing array⁵, making it requires four single-point devices with different polarization filtering directions to detect polarized signals along four different directions for the calculation of one pixel in *DoLP*. Theoretically, wire grid polarizers with matured fabrication technology can also reach a high polarization extinction ratio as only light polarized parallel to the grid can pass through the polarizer. As for 2D materials including Te, their intrinsic polarized photosensitivity can provide another route to realize polarization functionalities. For example, the intrinsic polarized photosensitivity of Te promotes polarization imaging without polarization filters. The photocurrent in one single point device along different polarization orientations can be collected by rotating the Te device, for the calculation of *DoLP* to enhance the imaging contrast. However, it's worth noting that the Te-based devices and many other 2D materials-based polarization devices are still at the infancy with some existing challenges at present. Currently, the fabrication of large-scale device arrays is a great challenge in the field of 2D materials. With the development of matured

methods for producing large films, the Te photodetector array could be one promising option to realize the practical polarization imaging. This work serves as the first step towards the development of polarization imaging arrays based on 2D materials, which is of great significance.

4. Page 13, line 8-10:

2D Te is also applicable for flexible devices with polarization sensitivity, promising the high level of integration and flexibility.

#2, perspective of tellurium as photodetector

The authors mentioned 'III-V compounds and many 3D materials...suffer from large noise at room temperature, thus requiring large cooling facilities to enhance detectivity'. However, I strongly doubt this claim as the method to grow single-crystalline III-V compounds are very mature. GaAs can have very high mobility. They usually made to detect light with much larger wavelength than light in the paper, and thus they are specially made to target these wavelengths with cooling condition. I believe if made by targeting at short-wavelength infrared light, III-V compounds could behave much better than nearly all 2D materials, even at room temperature. This is beyond the discussion of this paper.

Answer: We sincerely thank the reviewer for the comments. There may have been a misunderstanding about the description of “III-V compounds and many 3D materials...suffer from large noise at room temperature, thus requiring large cooling facilities to enhance detectivity”. In our work, we want to state that the functionalities of the photodetecting system should be designed based on the intrinsic properties of the channel material, as a result, both III-V compounds and 2D materials can be applied for different functionalities considering the requirements for different figure-of-merits, and they have different advantages and disadvantages. Although 2D materials are facing the challenge of large-scale growths, 2D materials for some

aspects promise to provide another routine to realize novel photodetecting devices in the future. This is why we have taken efforts to explore the intrinsic polarization of the 2D Te for contrast-enhanced photodetecting and imaging.

Based on our intension, the above description about III-V compounds mentioned in the 1st response letter is misunderstanding, which is not actually what we want to emphasize. Here we have just focused on the advantages and challenges faced by 2D materials at present, and the performances of the 2D Te-based photodetectors in the revised manuscript.

We sincerely thank again the reviewer. Just as the reviewer mentioned, this is beyond the discussion of this paper. But we have still carefully checked the 2nd revised manuscript to ensure this related discussion uncontroversial.

#3 The authors also mentioned other ways to make large area Te, however, it is very challenging first of all. Also if we consider that the application in this paper needs single-crystalline structure, it is nearly impossible.

Answer: Thanks for the reviewer's comments. As we have mentioned above, 2D materials promise to provide another routine to realize novel photodetecting devices in the future, by integrating with photonic crystals, waveguides, meta-materials, flexible substrates, other complex structures, and making the full of the unique characters including polarization, spin-related phenomena, etc, for photodetecting. Although 2D materials are facing a big bottleneck in synthesis of large-area 2D materials at present, it's still valuable and significant to explore the photodetecting applications based on 2D materials, as the potential of 2D materials can be further promoted after solving the problems in their large-scale growths, which have attracted many groups to take efforts.

Here, we have successfully synthesized the largest 2D Te single crystal through a substrate free approach¹, which can lead to the substrate-agnostic transfer^{2,3} and the

deterministically assemble/integration of our 2D Te with the micro-fabricated circuits to make related devices. Moreover, the related discussions in the scope of this paper are focused on the high performances of 2D Te-based photodetector, and demonstrated for the first time the 2D materials-based stable mid-infrared polarization imaging with enhanced contrast under strong scattering environment at room temperature. The growth of large-area 2D Te is beyond the scope of this work. Although the imaging is realized based on just one pixel due to the limited growth method at present, the imaging mechanism based on 2D Te in our manuscript is applicable after solving the problem of Te, and our work is the first step toward this goal.

References:

1. Wang, Y. *et al.*, Field-effect transistors made from solution-grown two-dimensional tellurene. *Nat. Electron.*, 1, 228-236 (2018).
2. Yuan, H. *et al.* Polarization-sensitive broadband photodetector using a black phosphorus vertical p-n junction. *Nat. Nanotechnol.* 10, 707-713 (2015).
3. Photonics and optoelectronics of 2D semiconductor transition metal dichalcogenides. *Nat. Photonics* 10, 216 - 226 (2016).

#2.1 Defects

The authors claim 'few defects nature based on HAADF-STEM'. However, from my point of view, it is very hard to conclude this by such measure. Usually, STEM can only observe a tiny area of the sample and does not give statistic of defects concentration. In principle, if there is no defects, Te should show absolute insulating behavior at room temperature instead of the behavior as shown in Fig. 1 and 2.

Answer: Thanks for the reviewer's comments. It's true that HAADF-STEM can only observe a small area, but by scanning the crystal within a large area, the crystal

quality can be verified, which was demonstrated in our previous report (Wang Y. *et al.*¹). In addition, the theoretical effort² suggests that the enhanced inter-chain interaction in 2D Te leads to a fast transport of defects (e.g., vacancies and interstitial atoms) across the Te chains, together with the fast intra-chain transport, which enables rapid healing of these defects at room temperature. Moreover, our preliminary characterizations¹ for many samples suggest their good crystallinity without observing line defects such as grain boundaries and just with few defects. Considering these factors, we will leave the exploration of the potential defects concentration and their influences in 2D Te on its photodetector performance for future studies. In principle, Te should be insulating if there is no defects. In experiments, the Te sample is of high quality with few defects, leading to intrinsic p-type behavior of 2D and bulk Te at room temperature as reported in our work, and also widely reported in many previous works¹⁻⁷.

References:

1. Wang, Y. *et al.*, Field-effect transistors made from solution-grown two-dimensional tellurene. *Nat. Electron.*, **1**, 228-236 (2018).
2. Liu, Y. Y., Wu, W. Z., Goddard, W. A., Tellurium: fast electrical and atomic transport along the weak interaction direction. *J. Am. Chem. Soc.*, **140**, 550-553 (2018).
3. Zhou, G. *et al.*, High-mobility helical tellurium field-effect transistors enabled by transfer-free, low-temperature direct growth. *Adv. Mater.*, **30**, 1803109 (2018).
4. Blakemore, J. S., Nomra, K. C., Intrinsic optical absorption in tellurium, *Phys. Rev.*, **127**, 1024-1029 (1962).
5. Amani, M. *et al.* Solution-synthesized high-mobility tellurium nanoflakes for short-wave infrared photodetectors. *ACS Nano* **12**, 7253-7263 (2018).
6. Wu, W. *et al.*, Tellurene: its physical properties, scalable nanomanufacturing, and

device applications. *Chem. Soc. Rev.* **47**, 7203-7212 (2018).

7. Du, Y. *et al.*, One-dimensional van der Waals material tellurium: Raman spectroscopy under strain and magneto-transport. *Nano Lett.*, **17**, 3965-3973 (2017).

#2.2 Gain and photogating.

The authors did control experiment by measuring transfer curves with and without illumination in Fig. S9. However, they failed to show the original data of transfer curves. Instead, they show photocurrent vs. gates. This confuses me. Let's assume all the transfer data exactly show the minimum points at the same gate values. This cannot rule out the photogating behavior as the transfer curve is a collective behavior of all the electrons flowing inside the device. And these dark electrons may overwhelm the trapped carrier density, which plays a role in photogating. This also relates to the trapping time scale of the photocarrier and the DC measurement dwell time. As the Authors may measure it by Keithley, which typically has long DC dwell time and may further cloud the observation of neutrality point's movement.

Answer: Thanks for the reviewer's comments. The photocurrent vs. gate bias is calculated by: $I_{ph}(V_g) = I_{illumination}(V_g) - I_{dark}(V_g)$ and the transfer curves under illumination can be extracted from the data. We have added the transfer curves under illumination in the revised manuscript to make our conclusions clearer (Figure 2G). The neutral point of the transfer curve is not shifted under on or off illumination, suggesting that the deep trap-induced photogating effect is not dominating, which is consistent with previous works¹⁻⁴. The photoresponsivity decreases with increased illumination power, which also indicates that the trap-induced photogating effect plays weaker roles in our illuminating power range⁶. To further respond to the reviewer's concern, we have also measured the photocurrent distribution in the 2D Te device, as shown in Fig. 2H. The photocurrent is mainly generated at the channel/electrode interface instead of the whole channel, and no photoresponse is

observed at the center of the channel, indicating that deep-trap induced photogating is not the dominant factor in the 2D Te device^{5,6}.

Actually, just as the reviewer has stated, the dark current can overwhelm the trapped carrier density, this is common in the field of 2D materials. When the trapped carrier density is very high, the photogating can dominate the device performance and the neutral point of the transfer curve can shift obviously¹⁻⁴. But precisely because of the very low trapped carrier density comparing with the carrier density in the whole device, the dark current always overwhelms the trapped carrier density, leading to that the photogating may not be obvious in the device and the neutral point of the transfer curve is not shift⁵⁻⁷, which is the same case of the 2D Te in our work. The trap density is very low since Te is of high quality only with few defects, which can be filled or emptied easily and fast. The transport behaviors are not dominated by photogating in our case, experiments as we have discussed above can be applied to prove our claims. The existence of photo-gain is a photoconductive gain originated from band tail states as we have stated in the revised manuscript, which is a fast process to provide a large gain for photodetecting based on some 2D materials^{8, 9}. In addition, strong photogating can be observed in many 2D materials-based photodetectors when the thickness is thinned down to a few layers, as their atom-thin nature provides an easily modulation by photogating. In our work, considering that the thickness of the Te is relatively thick, its carrier transport behaviors under illumination can be less affected by photogating.

For measurements by Keithley, we have used KEYSIGHT B1500A to measure the time-resolved photoresponse of the 2D Te-based photodetector, but wrongly written KEITHLEY 1500 in the methods part. We thank the reviewer for pointing out this spelling mistake and have corrected it in the revised manuscript. KEYSIGHT B1500A is widely used to characterize the photogating behaviors based on shifts in transfer curves⁷. Besides, we have collected the photocurrent data when the photoresponse is stable, after changing the illumination power, we have referred to the datasheets and handbook for our device and tried our best to relieve the influence of the DC dwell

time. We have also referred to many previous works to ensure the veracity of our experimental methods. Therefore, it is reasonable to conclude that the effects of the DC dwell time of our measurement tool, if any, can be safely ignored in our discussion. We thank again for the reviewers to make us think more about the DC dwell time.

References:

1. Konstantatos, G. *et al.* Hybrid graphene-quantum dot photodetectors with ultrahigh gain. *Nat. Nanotechnol.* **7**, 363-368 (2012).
2. Nguyen, V. T. *et al.* Phototransistors with negative or ambipolar photoresponse based on as-grown heterostructures of single-walled carbon nanotube and MoS₂. *Adv. Mater.* **28**, 1802572 (2018).
3. Fang, H. and Hu, W. Photogating in low dimensional photodetectors. *Adv. Sci.* **4**, 1700323 (2017).
4. Guo, Q. *et al.* Black phosphorus mid-infrared photodetectors with high gain. *Nano Lett.* **16**, 4648-4655 (2016).
5. Long, M. *et al.* Room temperature high-detectivity mid-infrared photodetectors based on black arsenic phosphorus. *Sci. Adv.* **3**, e1700589 (2017).
6. Zhou, Z. *et al.* Perpendicular optical reversal of the linear dichroism and polarized photodetection in 2D GeAs. *ACS Nano*, **12**, 12416-12423 (2018).
7. Amani, M. *et al.* Solution-synthesized high-mobility tellurium nanoflakes for short-wave infrared photodetectors. *ACS Nano* **12**, 7253-7263 (2018).
8. Furchi, M. M., Polyushkin, D. K., Pospischil, A., Mueller, T. Mechanisms of photoconductivity in atomically thin MoS₂. *Nano Lett.* **14**, 6165-6170 (2014).
9. Fu, Q. *et al.* Ultrasensitive 2D Bi₂O₂Se phototransistors on silicon substrate. *Adv. Mater.* **31**, 1804945 (2018).

#3 *The authors rule out photogating effect by showing the response time of 60 micro-second. They also mentioned that it is ultrafast. Firstly this speed is not ultrafast if compared with response of graphene and III-V compounds (shorter than 1 ns). So this speed exactly demonstrates the photogating if this speed is material-limit. Secondly, the authors claimed 354 A/W responsivity. This is against the quantum physics if there is no gain. If there is no photogain, as the authors claimed, there should be another explanation for the unusual quantum efficiency. The authors' explanation of 'large carrier life time and short transit time' simply does not stand from my point of view. If we assume all the photons excite photoelectrons, and then all the electrons and holes move out from the channel, the efficiency should be 100%, no matter how fast the charges move. Therefore, the authors need to find explanation for the unusual quantum efficiency if they disagree with me.*

Answer: Thanks for the reviewer's comments. The response time is influenced by many factors, including the materials quality (defects density, mobility, electron mass), strain, scattering at substrate interface, channel size, built-in p-n junctions, doping etc., as a result, the response time ranges from 10^{-12} s to 10^0 s level for different materials and device structures, including both 2D materials such as graphene, and III-V compounds. It's also hard to determine whether the speed is material-limit for many works. For graphene and III-V compounds, it's easier to reach an ultrafast response time generally. For example, it has been revealed by the studies that the faster response speed of graphene is usually caused by the ultra-narrow channel length and ultrahigh mobility of graphene¹⁻², or the formation of built-in p-n junctions³. Some previous works have also reported that III-V nanowires with lateral collection of carriers can lead to fast speed (at the level of several ps to tens of μ s), which is related to the modulation of the surface charging states⁴⁻⁶, including doping⁴, and effective separation of electrons and holes in the formed depletion layer⁷.

Generally, according to many works about 2D materials-based photodetectors

dominated by trap-induced photo-gating, their response speeds are usually relatively slow (at the level of 10^{-3} to 10^0 s)⁸⁻¹². It's difficult to determine the photogain mechanism by only considering the response time of ~ 60 μ s for our Te device. But we have provided more experimental results to explore the photogain mechanism. Firstly, the neutral point of the transfer curve is not shifted under both dark and illuminated conditions. Secondly, the photocurrent is mainly generated at the channel/electrode interface instead of the whole channel, suggesting the deep trap-induced photogating effect is not dominating in our device¹³⁻¹⁴.

To further discuss the photogain mechanism, we have already noted in our manuscript carefully that the photogain is originated through shallow states of photo-induced carriers at band tails, which is a result of photoconductive effect and different from photogating related with defect-induced deep traps¹⁵⁻¹⁷. Generally, the carrier lifetime and transit time are widely used to observe the existence and evolution of the photoconductive gain, according to previous work^{15, 16, 18, 19}, so we have calculated the photoconductive gain based on the carrier lifetime and transit time to be $\sim 2.3 \times 10^3$, which can also explain the existence of large gain. The EQE results higher than 100% in our device is related to the large gain, otherwise the EQE should be below 100% in theory. In addition, Guo, Q. *et al.*'s work also noted that "*We note that photoconductive gain can still occur without considering the localized states. However, it usually requires a high mobility channel, large source-drain bias or short channel length to ensure that the transit time is small enough compared to the photo-carrier lifetime without trapping effect*"¹⁹, this is also in accordance with our case as the Te crystal shows large mobility, the channel length is relatively short, which can also contribute to the high photoconductive gain in our device.

In our manuscript, there may still exist some inaccurate statements, we have taken the reviewer's comments into consideration and revised them carefully to make the whole manuscript clearer, and the revised parts are listed below:

References:

1. Mueller, T., Xia, F. and Avouris, P. Graphene photodetectors for high-speed optical communications. *Nat. Photon.* **4**, 297-301 (2010).
2. Liu, C. H., Chang, Y. C., Norris, T. B., Zhong, Z. H. Graphene photodetectors with ultra-broadband and high responsivity at room temperature. *Nat. Nanotechnol.* **9**, 273-278 (2014)
3. Xia, F. *et al.* Ultrafast graphene photodetector. *Nature Nanotechnol.* **4**, 839-843 (2009).
4. Xia, H. *et al.* Distinct photocurrent response of individual GaAs nanowires induced by n-type doping. *ACS Nano* **6**, 6005-6013 (2012).
5. Lapiere, R. R. *et al.* A review of III-V nanowire infrared photodetectors and sensors. *J. Phys. D: Appl. Phys.* **50**, 123001 (2017).
6. Li, D. *et al.* Ultra-fast photodetectors based on high-mobility indium gallium antimonide nanowires. *Nat. Commun.*, **10**, 1664 (2019).
7. Tan, H. *et al.* Single-crystalline InGaAs nanowires for room-temperature high-performance near-infrared photodetectors. *Nano-Micro Lett.* **8**, 29-35 (2016).
8. Li, L. *et al.* 2D GeP: an unexploited low-symmetry semiconductor with strong in-plane anisotropy. *Adv. Mater.* **30**, 1706771 (2018).
9. Liu, C.-H. *et al.* Graphene photodetectors with ultra-broadband and high responsivity at room temperature. *Nat. Nanotechnol.* **9**, 273-278(2014).
10. Xu, H. *et al.* High responsivity and gate tunable graphene-MoS₂ hybrid phototransistor. *Small* **10**, 2300-2306 (2014).
11. Kufer, D. and Konstantatos, G. Highly sensitive, encapsulated MoS₂ photodetector with gate controllable gain and speed. *Nano Lett.* **15**, 7307-7313 (2015).

12. Fang, H. and Hu, W. Photogating in low dimensional photodetectors. *Adv. Sci.* **4**, 1700323 (2017).
13. Long, M. *et al.* Room temperature high-detectivity mid-infrared photodetectors based on black arsenic phosphorus. *Sci. Adv.* **3**, e1700589 (2017).
14. Zhou, Z. *et al.* Perpendicular optical reversal of the linear dichroism and polarized photodetection in 2D GeAs. *ACS Nano*, **12**, 12416-12423 (2018).
15. Fu, Q. *et al.* Ultrasensitive 2D Bi₂O₂Se phototransistors on silicon substrate. *Adv. Mater.* **31**, 1804945 (2018).
16. Jiang, J. *et al.* Defect engineering for modulating the trap states in 2D photodetectors. *Adv. Mater.* **30**, 1804332 (2018).
17. Konstantatos, G. *et al.* Hybrid graphene-quantum dot phototransistors with ultrahigh gain. *Nat. Nanotechnol.* **7**, 363-368 (2012).
18. Huang, M. *et al.* Broadband black-phosphorus photodetectors with high responsivity. *Adv. Mater.* **28**, 3481-3485 (2016).
19. Guo, Q. *et al.* Black phosphorus mid-infrared photodetectors with high gain. *Nano Lett.* **16**, 7253-7263 (2016).

1. Page 8, line 1-15:

The high performance is also related to a large photoconductive gain, which is assisted by the fast carrier transition time due to the high mobility of Te and the short channel length of $\sim 6.0 \mu\text{m}$. In addition, the high crystal quality with few defects can contribute to the photoconductive gain through shallow states of photo-induced carriers at the band tail^{44, 45}, which is a fast process to provide a large gain. The existence of photogain is revealed from the EQE values in Figure S8A-C. Besides, we have also measured the transfer curves under $1.06 \mu\text{m}$ illumination and 1.0 V drain

bias at room temperature, as shown in Figure 2G panel (i), the net photocurrent is summarized in the panel (ii), the maximum net photocurrent is realized at the neutral point with ~ 5.9 V gate bias, also suggesting that the photoresponses of the device are not dominated by the trap-induced photogating effect. To further show the origin of photogain, we have measured the photocurrent distribution in the Te channel (Figure 2G panel (iii)), the photocurrent is generated at the channel/electrode interface instead of the whole channel, excluding the photogating as a dominant factor for the high photogain.

2. Page 8, line 19-24:

The carrier lifetime is extracted from transient photoluminescence decay curves of Te, as shown in Figure S10. Based on carrier lifetime and transit time, the photogain (G) at the level of $\sim 2.3 \times 10^3$ can be calculated, indicating the high performance in the Te-based device. Besides, to make it more convenient to compare the device performances with previous works, the calculated detectivity (D_C^*) is also summarized in Figure S11.

#3.2.3 Detectivity

The authors have used the exact equation to calculate the detectivities and admitted that it is 1 to 2 orders of magnitude smaller than the values by their method. However, they insist to use their values in the main text. They also cited several papers and claim it is 'implemented in most works about 2D materials-based devices'. My opinion is that there should be only one true value in every case, which is exactly the beauty of science. I am not sure if 'most' work from 2D materials-based devices have used the simplified equation, but I do think that the noise equivalent power is exactly the core of detectors. By saying noise, it refers all noise. I hope people in 2D world were right, but I am afraid, we have to respect the truth and completely stop using the simplified equation in current paper.

Answer: Thanks for the reviewer's valuable comments. We agree with the reviewer for the comment about detectivity. In the 1st response letter, we have measured the detectivity D_M^* , but put in the 1st revised supplementary information. Here, we have used the measured detectivity D_M^* in the 2nd revised manuscript to make the discussions more accurate. For low-temperature case, we have also re-measured the noise density spectra to calculate D_M^* under 77 K temperature. The revised parts in the manuscript are listed below:

1. Page 2, line 14-17:

Broadband (from 0.52 μm to 3.0 μm) sensitive photoresponse is realized, with high photoresponsivity of $\sim 3.54 \times 10^2$ A/W and detectivity of $\sim 3.01 \times 10^9$ Jones in the MIR range (the wavelength of 3.0 μm) at room temperature.

2. Page 4, line 5-11:

The wide spectral photoresponse of the Te photodetector is confirmed from 0.52 μm to 3.0 μm , and the most sensitive photoresponsivity (R) under 1.06 μm illumination is $\sim 1.36 \times 10^3$ A/W with corresponding measured detectivity (D_M^*) of $\sim 1.15 \times 10^{10}$ Jones. The photoresponsivity also remains higher than $\sim 3.53 \times 10^2$ A/W and the measured detectivity higher than $\sim 3.01 \times 10^9$ Jones, when illuminated by the incident light with a wavelength of 3.0 μm .

3. Page 7, line 16-30; page 8, line 1-15:

To characterize the device performances under different illumination powers and wavelengths, we have summarized the major parameters of photoresponsivity (R) and measured detectivity (D_M^*) of the photodetector at room temperature by using the pseudo-color mapping, as shown in Figure 2E, F, I, and J, respectively (scatter diagrams are included in Figure S7K-N). The drain bias is fixed at 1.0 V, and the gate bias is fixed at 0 V, the frequency of the noise current is chosen to be 1 kHz, the corresponding noise current density is 4.55×10^{-11} A/Hz^{1/2} at room temperature, and the calculation details are described in the Supplementary Information notes 5.

The lower illumination power always leads to higher R and D_M^* , and the evolution of R and D_M^* under fixed laser power is in accordance with absorption results. Due to effective absorption of Te at $3.0\ \mu\text{m}$, high R of $\sim 3.54 \times 10^2\ \text{A/W}$ and D_M^* of $\sim 3.01 \times 10^9$ Jones are realized under the illumination at the wavelength of $3.0\ \mu\text{m}$, which is competitive among various 2D materials-based photodetectors (Supplementary Information, Table S2, Figure S26)^{24-26, 35-43}. The thick Te flake (Figure S1D) also contributes to such high performance due to its stronger absorption character, as the transmitted light to the substrate is limited. The high performance is also related to a large photoconductive gain, which is assisted by the fast carrier transition time due to the high mobility of Te and the short channel length of $\sim 6.0\ \mu\text{m}$. In addition, the high crystal quality with few defects can contribute to the photoconductive gain through shallow states of photo-induced carriers at the band tail^{44,45}, which is a fast process to provide a large gain. The existence of photogain is revealed from the EQE values in Figure S8A-C. Besides, we have also measured the transfer curves under $1.06\ \mu\text{m}$ illumination and $1.0\ \text{V}$ drain bias at room temperature, as shown in Figure 2G panel (i), the net photocurrent is summarized in the panel (ii), the maximum net photocurrent is realized at the neutral point with $\sim 5.9\ \text{V}$ gate bias, also suggesting that the photoresponses of the device are not dominated by the trap-induced photogating effect. To further show the origin of photogain, we have measured the photocurrent distribution in the Te channel with another device (Figure 2H), the photocurrent is generated at the channel/electrode interface instead of the whole channel, excluding the photogating as a dominant factor for the high photogain.

4. Page 8, line 19-24:

The carrier lifetime is extracted from transient photoluminescence decay curves of Te, as shown in Figure S10. Based on carrier lifetime and transit time, the photogain (G) at the level of $\sim 2.3 \times 10^3$ can be calculated, indicating the high performance in the Te-based device. Besides, to make it more convenient to compare the device performances with previous works, the calculated detectivity (D_C^*) is also summarized in Figure S11.

5. Page 9, line 1-7:

The dependence of R and D_M^* on the wavelength and laser power of the incident light, are summarized in Figure 2E, F, I, and J, respectively (scatter diagrams are included in Figure S12K-N, external quantum efficiency (EQE) is shown in Figure S8D-F). Here the results show the similar trends to those measured at 300 K (Figure 2E-2F). At 77 K, a higher R can reach up to $\sim 8.36 \times 10^2$ A/W and D_M^* is at a higher level of $\sim 9.04 \times 10^9$ Jones, under the illumination wavelength of $3.0 \mu\text{m}$.

6. Page 26, Figure 2:

7. Page 26, line 6-8:

(E)-(F) Pseudo-color mapping figures for R and D_M^* , as a function of laser power and wavelength at room temperature (300 K).

8. Page 26, line 10; page 27, line 1-7:

(G) Photoresponses under different gate biases. The drain bias is 1.0 V, and the incident wavelength is 3.0 μm . Panel (i) shows the transfer curves under different incident powers and the dark state, and panel (ii) shows the net photocurrents under different illumination powers. The maximum photocurrent is achieved at the neutral point of ~ 5.9 V gate bias for all the incident powers. Panel (iii) shows the photocurrent distribution in the Te channel, with the photocurrent generated at the channel/electrode interface. (H) The photocurrent distribution in the Te channel under 0.637 μm and 0.83 μm illumination at 1.0 V drain bias and 0 V gate bias, with the photocurrent generated at the channel/electrode interface. The inset shows the image of the Te device with laser spot size of ~ 3 μm , scale bar, 10 μm . (I)-(J) Pseudo-color mapping figures for R and D_M^* , as a function of laser power and wavelength at 77 K.

9. Supplementary information, page 9, line 18-28; page 10, line 1-16:

The noise spectral density of the device is measured as shown in Figure S23A and B at room temperature and 77 K, respectively, the noise at 0 V drain bias is lower than that of at 1.0 V drain bias, and the noise is dominated by flicker (I/f) noise which is originated from fluctuations of local electronic states. For our photocurrent measurements, the sampling frequency is much higher than 1 Hz, so the noise in our device should be flicker (I/f) noise, which is analogous to previous works about the 2D materials-based devices⁶. To extract the D_M^* value, the drain bias is at 1.0 V, the bandwidth $\Delta f = 1000\text{Hz}$, the corresponding noise current density is $4.55 \times 10^{-11} \text{ A}/\sqrt{\text{Hz}}$ and $3.58 \times 10^{-11} \text{ A}/\sqrt{\text{Hz}}$ at room temperature and 77 K, respectively.

In the field of 2D materials-based devices, some works have implemented the

dark current to calculate the detectivity; in this algorithm, the noise current is replaced by equation S20:

$$i_N = \sqrt{2qI_{dark}\Delta f}, \quad (S20)$$

where q is the electron charge, and I_{dark} is the dark current of the device without laser illumination ⁵. Then implement equation S20 into S18, the detectivity can be calculated by equation S21, here we refer this detectivity as the calculated detectivity (D_C^*):

$$D_C^* = \sqrt{\frac{A}{2qI_{dark}}} R. \quad (S21)$$

Based on equation S21, it's noted that the working bandwidth is removed, so in most works, the bandwidth can be ignored to extract D_C^* data. To make it more convenient to compare the device performances with previous works, the calculated detectivity (D_C^*) is also summarized in Figure S11 for Te-based device. The dark current is measured many times before laser illumination, the drain bias is from -1.0 V to 1.0 V, as shown in Figure S23C and D at room temperature and 77 K, respectively. The dark current can lead to overestimated detectivity comparing with the noise density spectra, as shown in Figure S11, the measured detectivity D_M^* is about 1-2 magnitude lower than the calculated detectivity D_C^* . Here the D_M^* value is more accurate to characterize the device performances.

10. Supplementary information, page 23, Figure S7:

Figure S7. Unpolarized broadband optoelectronic responses for tellurium nanoflake device at room temperature (300 K). (A)-(J) Photocurrent for 0.52 μm , 0.637 μm , 0.785 μm , 0.83 μm , 0.94 μm , 1.06 μm , 1.31 μm , 1.55 μm , 2.3 μm , and 3.0 μm illumination, respectively, as a function of laser power and drain bias, the gate bias is 0 V. (K)-(L) R and D_M^* as a function of laser power for different wavelengths, respectively, the drain bias is fixed at 1.0 V, the gate bias is 0 V. (M)-(N) R and D_M^* as a function of wavelength under different illumination laser powers at room temperature.

11. Supplementary information, page 24, Figure S8:

Figure S8. (A)-(C) *EQE* results at room temperature. (A) Pseudo-color mapping image of *EQE*. (B) *EQE* as a function of illumination laser power for different wavelengths at room temperature. (C) *EQE* as a function of wavelength under different laser powers at room temperature. (D)-(F) *EQE* results at low temperature. (D) Pseudo-color mapping image of *EQE*. (E) *EQE* as a function of illumination laser power for different wavelengths at 77 K temperature. (F) *EQE* as a function of wavelength under different laser powers at 77 K temperature. The performance under 77 K is higher.

12. Supplementary information, page 27, Figure S11:

Figure S11. The calculated detectivity (D_C^*) results. (A)-(C) D_C^* at room temperature. (A) Pseudo-color mapping results of D_C^* . (B) D_C^* as a function of laser power for different wavelengths at room temperature. (C) D_C^* as a function of wavelength under different laser powers at room temperature. (D)-(F) D_C^* at 77 K temperature. (A) Pseudo-color mapping results of D_C^* . (B) D_C^* as a function of laser power for different wavelengths at 77 K temperature. (C) D_C^* as a function of wavelength under different laser powers at 77 K temperature. D_C^* is about 1-2 magnitude higher than D_M^* due to the lower dark current than the actual noise current.

13. Supplementary information, page 28, Figure S12:

Figure S12. Unpolarized broadband optoelectronic responses for tellurium nanoflake device at 77 K. (A)-(J) Photocurrent under illumination of 0.52 μm , 0.637 μm , 0.785 μm , 0.83 μm , 0.94 μm , 1.06 μm , 1.31 μm , 1.55 μm , 2.3 μm , and 3.0 μm , respectively. The gate bias is fixed at 0 V. (K)-(L) R and D_M^* as a function of laser power for different wavelengths, respectively, the drain bias is fixed at 1.0 V, the gate bias is 0 V. (M)-(N) R and D_M^* as function of wavelength under different illumination laser powers at 77 K temperature.

14. Supplementary information, page 39, Figure S23:

Figure S23. (A)-(B) Noise spectral density of the tellurium device at room temperature and 77 K, respectively. The drain bias is fixed at 0 V (red curve) and 1.0 V (blue curve), and the gate bias is 0 V. The noise is dominated by $1/f$ noise. (C)-(D) Dark current with the drain bias from -1.0 V to 1.0 V at room temperature and 77 K, respectively, the gate bias is 0 V. It's also proved that the dark current is lower than the noise current, which leads to the overestimated calculated detectivity (D_C^*) comparing with the measured detectivity (D_M^*).

Reviewer #3

The authors have taken time to thoroughly address the reviewers' comments, significantly strengthening the manuscript. The expansion of the manuscript enables the reader to understand the context and potentially repeat the work if desired.

One key point of novelty within the paper remains the high mobility reported, $938 \text{ cm}^2 \text{ V}^{-1} \text{ s}^{-1}$, which is almost as high as black phosphorus, $984 \text{ cm}^2 \text{ V}^{-1} \text{ s}^{-1}$. While this improvement is dramatic compared to the previously best reported value of Te, $700 \text{ cm}^2 \text{ V}^{-1} \text{ s}^{-1}$, there is an apparent trend of increasing mobility with increasing flake thickness (i.e moving from the 2D domain into the bulk domain), which the authors address in their response to reviewers letter (pg 60, L1,2). This trend with increased mobility also holds true for black phosphorus. The expansion of table S1 allows readers to draw their own conclusions on these results in the context of materials performance - but logically the question remains, if thicker Te flakes are used would the mobility continue to increase giving improved performance, and in this case is 2D an important distinction.

Answer: We are grateful for the reviewer's comments and appreciation for our work. Yes, we agree with the reviewer. Actually, with the increase of thickness, the mobility of Te will be increased to saturation that a certain thickness is achieved¹. In addition, the dark current in thicker samples can also be higher, limiting the detectivity. As a result, all these effects are competing and there should be a trade-off in thickness for photodetection. We have discussed the thickness dependent properties of 2D Te detailedly in the previous work¹, and we are focusing on the polarized infrared imaging under scattering with enhanced contrast based on the intrinsic anisotropy of 2D Te.

Reference:

1. Wang, Y. *et al.*, Field-effect transistors made from solution-grown two-dimensional tellurene. *Nat. Electron.*, **1**, 228-236 (2018).

The introduction, pg 4 L94-96, still states that 2D-Te has a mobility "which is far larger than other 2D materials", which the authors have acknowledged is not the case with black phosphorous having a larger mobility.

Answer: Thanks for the reviewer pointing out our inaccurate statements. Sorry for the overemphasized claim about the Te mobility. In the revised manuscript, we have corrected our statements carefully.

The revised parts in the manuscript are listed below:

1. Page 4, line 3-5:

carrier mobility measured by field effect transistor (FET)-based devices can reach up to $\sim 9 \times 10^2 \text{ cm}^2 \text{V}^{-1} \text{s}^{-1}$ at room temperature, which is at the forefront in the field of 2D materials.

2. Supplementary information, page 41, Figure S25:

These points are relatively minor, and I believe given the expansive extra detail added to the manuscript that it can be considered suitable for publication in Nature Communications.

Answer: We are very grateful for the reviewer's efforts and comments again.

Reviewers' comments:

Reviewer #1 (Remarks to the Author):

The authors have addressed most of my earlier comments to better reflect the actual impact of the study and clarify some points.

One point has not been adequately addressed:

The response to the device absorption comment was confusing. The simple simulations I mentioned in the previous review incorporates the effects of multiple passes (I believe this is what the authors are alluding to?). I suggest the authors run their own fully rigorous simulations for their specific setup configuration to understand what I mean. As it currently stands, the reported absorption of the device is quite high (twice the limit for a 27nm Te flake on an SiO₂/Si substrate). Most experts in the field will notice this and suspect something is wrong with the characterization (including the other impressive metrics presented in the paper).

What was the SiO₂ thickness of your substrate?

A less important point is the use of the phrase "contrast-enhanced" in the title and other points of the text. This phrasing is confusing, and I would suggest removing mention of contrast altogether for the benefit of readers. As the authors have agreed in the reviewer comments, impressive contrast is not a convincing finding from this study.

Reviewer #2 (Remarks to the Author):

I believe that the authors have corrected several major mistakes in physics. Now the explanations, such as photogain and detectivity, comply with basic physics. With all these correct conclusions, we could look back and re-examine the manuscript. If compared with paper reported in ACS Nano 2018, 12, 7, 7253-7263, this paper reports relatively similar detectivity (in scale of 1E-9 jones), which leaves the main claim as polarization imaging. However, as I mentioned in my last (second)-round comment #1, the future application of this concept requests that all the pixels have the same orientation, which is nearly impossible for the given method. In addition, the material may not beat III-V compounds in DoLP, as III-V compounds integrated by metal grid should work better. My opinion regarding to the potential application here echoes well with the first reviewer, who also doubted the claim.

By saying these words, I mean that, compared with contribution of ACS Nano paper, the main contribution of this work is a further demonstration of polarization-dependent photoresponse by using polarization-dependent imaging system. However, this does not mean their work is not thorough and complete. On the contrary, the authors have carried out sufficient experiments to support the discussion. Their explanation, after several revisions, complies with basic physics finally.

Here are several mistakes:

#1. The photogain origins from the so-called band-tail states (shallow states) manuscript page 8, line 8. I believe these states come from the defects inside the crystal, if not from interfaces. This should be further clarified as it is contrary to 'few defects' as emphasized in the paper.

#2. In line 17, page 10 of SI, the statement of "iN is noise current" seems wrong. If I was not mistaken, it should be noise spectral density.

Reviewer #3 (Remarks to the Author):

The authors have significantly expanded the discussion and context of their work through the reviewing processes. I believe that with this added context and discussion the publication is suitable for Nature Communications.

Response Letter

Reviewer #1

The authors have addressed most of my earlier comments to better reflect the actual impact of the study and clarify some points.

Response: We are grateful for the reviewer's comments and the acknowledgement of our efforts to properly address the reviewer's comments.

1. One point has not been adequately addressed:

The response to the device absorption comment was confusing. The simple simulations I mentioned in the previous review incorporates the effects of multiple passes (I believe this is what the authors are alluding to?). I suggest the authors run their own fully rigorous simulations for their specific setup configuration to understand what I mean. As it currently stands, the reported absorption of the device is quite high (twice the limit for a 27nm Te flake on an SiO_2/Si substrate). Most experts in the field will notice this and suspect something is wrong with the characterization (including the other impressive metrics presented in the paper).

Response: We sincerely thank for the reviewer's valuable comments. In the previous response letter, we misunderstood the review comment and thought that the reviewer may refer to the absorption of Te with the light passing through only once, leading to the confusing response that "the measured value should be twice the absorption of Te". We are sorry that we did not fully understand the reviewer's intention well, and for our confusing response with insufficient details. In the revision, we have carried out additional simulation and experimental measurements after carefully inspecting our measurement setup. We hope that these new results and discussions could help clear

up the reviewer's concern.

Firstly, we have carried out the simulation of the absorption. The optical absorption coefficient α can be calculated based on the complex refractive index:

$$\alpha = \frac{2Ek}{hc},$$

E is the photon energy, h is the Planck's constant, c is the light speed, and k is the imaginary part of the complex refractive index. Since the imaginary part of the complex refractive index k as a function of photon energy is known based on previous works, we can calculate the absorption coefficient quickly. Based on the absorption coefficient, we assume the thickness of Te is ~27 nm, and the substrate is perfect 1/4 wave thickness, then the maximum absorption of ~33% is achieved at ~1.25 μm wavelength, as shown below.

Figure A1. Simulation results for the absorption characters of Te. The absorption ratio is calculated with the thickness of ~27 nm.

Based on the calculation results, we found that the measured absorption spectra show an irrationally high level of the absorption ratio, which may be due to some mistakes during our measurements for the total reflection spectra or the reflection spectra of the silicon wafer or the reflection spectra of the sample on the silicon wafer.

In this revision, we have carried out additional experiments to remeasure the absorption spectra, and revised the related discussions. We thank again for the reviewer's pointing out this mistake.

The revised parts in the revised manuscript and supplementary information are listed below:

1. Page 6, line 23-26:

The absorption characters from NIR to MIR range is observed under nonpolarized illumination (red curve), with a maximum absorption of ~28% located near 1.25 μm and absorption edge extended to more than 3.0 μm .

2. Page 25, Figure 1:

3. Page 26, line 5-6:

and the anisotropic ratio at 1.25 μm and 2.3 μm are ~1.8 and ~8.0, respectively.

4. Supplementary information page 22, Figure S6:

Figure S6. Anisotropic ratio for absorption spectra at several selected wavelengths. The anisotropic ratio is enhanced to be more than 8 from 2.3 μm to 3.0 μm , which is beneficial for polarized photodetection applications.

2. What was the SiO_2 thickness of your substrate?

Response: We sincerely appreciate the reviewer comment for us to clarify this information. The thickness of SiO_2 layer is ~ 300 nm.

3. A less important point is the use of the phrase “contrast-enhanced” in the title and other points of the text. This phrasing is confusing, and I would suggest removing mention of contrast altogether for the benefit of readers. As the authors have agreed in the reviewer comments, impressive contrast is not a convincing finding from this study.

Response: We are grateful to the reviewer for the valuable suggestion. We agree with the reviewer. Our work is the first step toward practical applications of polarization imaging in the field of 2D materials, and more efforts are required to reach a higher standard for contrast enhancement in the future. To address this comment, we have removed the statements about “contrast-enhanced” and checked the manuscript thoroughly to make our statements more concrete and accurate.

The revised parts in the manuscript are listed below:

1. Title:

Stable mid-infrared polarization imaging based on quasi-2D tellurium at room temperature

2. Page 2, line 10-11:

However, polarized infrared imaging under scattering based on 2D materials has yet to be realized to confirm such feasibility.

3. Page 2, line 19-22:

Significantly, the Te photodetector has a large anisotropic ratio of ~ 8 for $2.3 \mu\text{m}$ illumination, ensuring polarized imaging in a scattering environment, with the degree of linear polarization (DoLP) over 0.8 at the wavelength of \$2.3 \mu\text{m}\$.

4. Page 3, line 21-24:

Besides, there are reports of 2D material mid-infrared (MIR) polarization photodetectors^{24,25}, but the works about polarization imaging under scattering based on 2D materials have not yet been realized, addressing the feasibility of 2D materials-based photodetectors to further achieve contrast-enhanced polarized infrared imaging.

5. Page 4, line 11-14:

Due to the unique asymmetric crystal structure, the polarized photoresponse is also demonstrated, with the photoresponsivity anisotropic ratio of ~ 8 under the incident light with a wavelength of $2.3 \mu\text{m}$, which is sufficient for potential polarized imaging.

6. Page 4, line 19-21:

however, the quasi-2D Te-based photodetector reported here can realize imaging for the target with the degree of linear polarization (DoLP) over 0.8 at the wavelength of

2.3 μm .

7. Page 4, line 23-24:

Significantly, the feasibility of implementing Te for polarized imaging proved in this work is at the first stage,

8. Page 11, line 13-14:

the incident light is severely scattered to obstruct imaging based on non-polarizing facilities^{26, 52}.

9. Page 12, line 16:

the larger absorption anisotropic ratio can realize clearer polarization imaging,

10. Page 13, line 29-30:

quasi-2D tellurium photodetectors and used the photodetector to successfully realize polarized infrared imaging under scattering.

11. Page 14, line 4-6:

Based on the anisotropic photoresponses, polarized imaging capability is preserved for quasi-2D Te-based photodetector under scattering environments, with the DoLP over 0.8 at the wavelength of 2.3 μm .

Reviewer #2

1. I believe that the authors have corrected several major mistakes in physics. Now the explanations, such as photogain and detectivity, comply with basic physics. With all these correct conclusions, we could look back and re-examine the manuscript. If compared with paper reported in ACS Nano 2018, 12, 7, 7253-7263, this paper reports relatively similar detectivity (in scale of $1\text{E-}9$ jones), which leaves the main claim as polarization imaging. However, as I mentioned in my last (second)-round comment #1, the future application of this concept requests that all the pixels have the same orientation, which is nearly impossible for the given method.

Response: We sincerely thank the reviewer for the great efforts providing valuable suggestions, which help us significantly improve the quality of our manuscript. We also thank the reviewer for acknowledging our efforts for properly addressing the comments. We agree with the reviewer and admit that fabricating polarization imaging arrays with the same crystal orientations for all the pixels is challenging at present, which warrants more systematic investigation and development in future studies.

Despite still at the infancy stage regarding technology development, 2D materials hold significant potential for enabling novel device structures and functionalities in future photonics, through integrating with photonic crystals, waveguides, meta-materials, flexible substrates, or other complex structures. At this moment, many research groups are assiduously taking efforts to solve the growth problem of 2D materials and more scientific efforts are ongoing developing the advanced device fabrication based on micro-or nano-scale 2D materials, drawing to a close to the state-of-the-art to promise the practical applications in the future, but this is beyond the discussion and scope of this paper. Here, we report for the first time to experimentally realize the polarized IR imaging of the target under scattering environments at room temperature in 2D materials-based photodetecting field, and it will be impactful and of interest to a broad audience in materials and device research

in the IR photodetector community.

2. In addition, the material may not beat III-V compounds in DoLP, as III-V compounds integrated by metal grid should work better. My opinion regarding to the potential application here echoes well with the first reviewer, who also doubted the claim. By saying these words, I mean that, compared with contribution of ACS Nano paper, the main contribution of this work is a further demonstration of polarization-dependent photoresponse by using polarization-dependent imaging system. However, this does not mean their work is not thorough and complete. On the contrary, the authors have carried out sufficient experiments to support the discussion. Their explanation, after several revisions, complies with basic physics finally.

Response: We are grateful for the reviewer's comments and appreciation for the contribution of our work. As we have mentioned above, polarized imaging applications based on 2D materials still need to solve existing problems including matured growth method, higher anisotropic ratio for higher DoLP, etc. Meanwhile, the III-V compounds-based polarization imaging technologies are well developed and have been commercially applied in real life. Given the numerous intriguing properties of 2D materials and their potential for enabling novel, more capable photonic applications, we believe more systematic investigation and development are required in future studies to further improve the figure-of-merits for 2D materials, establish scalable processing schemes that are compatible with the state-of-the-art, and eventually realize the full potential of 2D materials for future photonics and optoelectronics applications.

Here are several mistakes:

#1. The photogain origins from the so-called band-tail states (shallow states) manuscript page 8, line 8. I believe these states come from the defects inside the crystal, if not from interfaces. This should be further clarified as it is contrary to 'few

defects' as emphasized in the paper.

Response: We sincerely thank the reviewer for this valuable comment. The band-tail states can be related to many factors, such as defects, impurities, doping, strain, mobility etc., which can be located at the surface and inside the crystal¹⁻³. As for Te, we agree with the reviewer's statement that the band-tail states should come from the defects inside the crystal, and the trapping and detrapping are fast at these defect centers, which are referred as shallow states⁴. Although the defect density can be low in the Te crystals⁵, the defects can still affect the photogain in our device through such band-tail states. To make our statements more accurate, we have clarified the discussion about band-tail states in the revised manuscript.

References:

1. Ni, Z. et al. Size-dependent structures and optical absorption of Boron-hyperdoped silicon nanocrystals. *Adv. Optical Mater.* 4, 700-707 (2016).
2. Milot, R. L. et al. The effects of doping density and temperature on the optoelectronic properties of formamidinium tin triiodide thin films. *Adv. Mater.* 30, 1804506 (2018).
3. Konstantatos, G. et al. Hybrid graphene-quantum dot phototransistors with ultrahigh gain. *Nat. Nanotechnol.* 7, 363-368 (2012).
4. Furchi, M. M., Polyushkin, D. K., Pospischil, A., Mueller, T. Mechanisms of photoconductivity in atomically thin MoS₂. *Nano Lett.* 14, 6165-6170 (2014).
5. Wang, Y. et al. Field-effect transistors made from solution-grown two-dimensional tellurene. *Nat. Electron.* 1, 228-236 (2018).

The revised parts in the manuscript are listed below:

1. Page 8, line 8-10:

In addition, the defects located inside the Te crystal can contribute to the photoconductive gain through shallow states of photo-induced carriers at the band tail^{44, 45},

2. Supplementary information page 11, line 6-7:

and the long carrier's life time τ_l is a result of shallow trap at defects inside the Te crystal due to band tail states.

#2. In line 17, page 10 of SI, the statement of “ i_N is noise current” seems wrong. If I was not mistaken, it should be noise spectral density.

Response: We sincerely thank for pointing out our oversight. We have referred to the theory about noise spectral density, and the measured detectivity should be $D^* = \frac{\sqrt{A\Delta f}}{NEP} = \frac{R\sqrt{A\Delta f}}{i_{noise}} = \frac{R\sqrt{A}}{i_{noise}/\sqrt{\Delta f}}$, A is the effective device area, NEP is the noise equivalent power, R is the photoresponsivity, i_{noise} is the noise current, Δf is the integration time. In our supplementary information, i_N should be “noise current density” $\frac{i_{noise}}{\sqrt{\Delta f}}$ (the unit is A/\sqrt{Hz}), which is related to noise current. We have corrected this in the revised supplementary information, as listed below:

1. Supplementary information page 9, line 14-18:

$$D_M^* = \frac{\sqrt{A}}{i_N} R = \frac{\sqrt{A}}{i_{noise}/\sqrt{\Delta f}} R, \quad (S18)$$

where A is the effective device area, Δf is the integration time, i_N is noise current density, i_{noise} is the noise current and R is the responsivity. The noise current i_{noise} is related to noise equivalent power (NEP):

$$NEP = \frac{i_{noise}}{R}. \quad (S19)$$

2. Supplementary information page 10, line 2-3:

in this algorithm, the noise current density is replaced by equation S20:

Reviewer #3

The authors have significantly expanded the discussion and context of their work through the reviewing processes. I believe that with this added context and discussion the publication is suitable for Nature Communications.

Answer: We sincerely thank the reviewer again for their precious time and efforts in reviewing our manuscript.

REVIEWERS' COMMENTS:

Reviewer #1 (Remarks to the Author):

The Authors have answered my remaining comments, and whilst it is alarming that they mismeasured the absorption characteristic by $\sim 100\%$ in the original submission, the revised plot is within what is physically possible.

Reviewer #2 (Remarks to the Author):

The physics in this manuscript complies with laws in physics. The claim is reasonable now. No further comments from me.

Response Letter

Reviewer #1 (Remarks to the Author):

The Authors have answered my remaining comments, and whilst it is alarming that they mismeasured the absorption characteristic by ~100% in the original submission, the revised plot is within what is physically possible.

Response: We are grateful for the reviewer's comments and the acknowledgement of our efforts to properly address the reviewer's comments.

Reviewer #2 (Remarks to the Author):

The physics in this manuscript complies with laws in physics. The claim is reasonable now. No further comments from me.

Response: We sincerely thank the reviewer for the great efforts providing valuable suggestions, which help us significantly improve the quality of our manuscript. We also thank the reviewer for acknowledging our efforts for properly addressing the comments.